_Article_

# Humanized COVID-19 decoy antibody effectively blocks viral entry and prevents SARS-CoV-2 infection

Kuo-Yen Huang[1], Ming-Shiu Lin[1], Ting-Chun Kuo[2], Ci-Ling Chen[1], Chung-Chih Lin[1], Yu-Chi Chou[3], Tai-Ling Chao[4], Yu-Hao Pang[4], Han-Chieh Kao[4], Rih-Sheng Huang[5], Steven Lin[5,6], Sui-Yuan Chang[4,7],* ID & Pan-Chyr Yang[1,2,8],** ID

## Abstract

To circumvent the devastating pandemic caused by severe acute respiratory syndrome coronavirus 2 (SARS-CoV-2) infection, a humanized decoy antibody (ACE2-Fc fusion protein) was designed to target the interaction between viral spike protein and its cellular receptor, angiotensin-converting enzyme 2 (ACE2). First, we demonstrated that ACE2-Fc could specifically abrogate virus replication by blocking the entry of SARS-CoV-2 spike-expressing pseudotyped virus into both ACE2-expressing lung cells and lung organoids. The impairment of viral entry was not affected by virus variants, since efficient inhibition was also observed in six SARS-CoV-2 clinical strains, including the D614G variants which have been shown to exhibit increased infectivity. The preservation of peptidase activity also enables ACE2-Fc to reduce the angiotensin II-mediated cytokine cascade. Furthermore, this Fc domain of ACE2-Fc was shown to activate NK cell degranulation after co-incubation with Spike-expressing H1975 cells. These promising characteristics potentiate the therapeutic prospects of ACE2-Fc as an effective treatment for COVID-19.

**Keywords** ACE2-Fc; COVID-19; decoy antibody; SARS-CoV-2; virus infection
**Subject Categories** Immunology; Microbiology, Virology & Host Pathogen Interaction

## Introduction

In December 2019, a new coronavirus, reported in Wuhan City, Hubei Province, China, and later named SARS-CoV-2, was classified as a new member of the beta-coronavirus family (Guan *et al*, 2020; Huang *et al*, 2020). SARS-CoV-2 causes atypical pneumonia (COVID-19) and apparently spreads more efficiently than SARS-CoV did in 2003 and MERS-CoV did in 2015 (de Wit *et al*, 2016). Currently, there were more than 47 million confirmed cases, with 1.2 million deaths till November 2020 globally (https://www.who.int/).

SARS-CoV infection is mediated by the transmembrane glycoprotein Spike, which recognizes and targets angiotensin-converting enzyme 2 (ACE2) for viral entry (Zhou *et al*, 2020). The viral Spike protein can be divided into two functionally distinct subunits, a receptor-binding subunit S1 and a membrane-fusion subunit S2. The S1 subunit recognizes and binds to ACE2 on host cells, while the fusogenic peptide on the S2 subunit facilitates fusion between the viral and host membrane to facilitate the release of the viral genome into the host cells (Tortorici & Veesler, 2019). Based on the viral sequence alignment (Lu *et al*, 2020) as well as the Cryo-EM structure of SARS-CoV-2 Spike protein (Wrapp *et al*, 2020), high similarities between the Spike protein of SARS-CoV and SARS-CoV-2 have been described (Monteil *et al*, 2020). Recently, it was reported that the receptor-binding domain (RBD) of SARS-CoV-2 S1, similar to that of SARS-CoV, interacts with the peptidase domain (PD) of ACE2 (Li *et al*, 2005; Wrapp *et al*, 2020).

ACE2 is expressed on the cell membrane of most organs and tissues, including the lungs, heart, kidney, brain, intestine, and endothelial cells (Kabbani & Olds, 2020). As a component involved in the signaling pathway of the renin–angiotensin system (RAS), which functions as a homeostatic regulator of vascular function, ACE2 plays an important role in the maturation of angiotensin (Ang), which controls vasoconstriction, blood pressure (Patel *et al*, 2016), and the inflammatory cytokine cascade mediated by TNF-α and IL-6 (Hirano & Murakami, 2020). ACE first metabolizes Ang I to Ang II, whose C-terminal domain is further cleaved by ACE2 to generate angiotensin 1–7 (Ang 1–7). Ang 1–7, having opposing

1   Institute of Biomedical Sciences, Academia Sinica, Taipei, Taiwan
2   Department of Internal Medicine, National Taiwan University Hospital and National Taiwan University College of Medicine, Taipei, Taiwan
3   Biomedical Translation Research Center (BioTReC), Academia Sinica, Taipei, Taiwan
4   Department of Clinical Laboratory Sciences and Medical Biotechnology, National Taiwan University College of Medicine, Taipei, Taiwan
5   Institute of Biological Chemistry, Academia Sinica, Taipei, Taiwan
6   Institute of Biochemical Sciences, National Taiwan University, Taipei, Taiwan
7   Department of Laboratory Medicine, National Taiwan University Hospital, Taipei, Taiwan
8   Genomics Research Center, Academia Sinica, Taipei, Taiwan
    *Corresponding author. Tel: +886 2 312 3456; E-mail: sychang@ntu.edu.tw
    **Corresponding author: Tel: +886 2 23562905; E-mail: pcyang@ntu.edu.tw

functions to Ang II, has been shown to exert antioxidant and anti-inflammatory effects upon lung and heart injury (Patel *et al*, 2016). The concentration of Ang II is delicately regulated. Increased levels of Ang II are believed to upregulate ACE2 activity, which subsequently leads to decreased Ang II and increased Ang 1–7 levels. Treatment with recombinant human ACE2 (rhACE2) and B38-CAP, a bacteria-derived ACE2-like enzyme, has been reported to suppress Ang II-induced hypertension, cardiac hypertrophy, and fibrosis in the mouse model (Liu *et al*, 2018; Minato *et al*, 2020). In addition, ACE2 was shown to improve sepsis-induced acute lung injury using cecal ligation and perforation (CLP) and endotoxin challenge (Imai *et al*, 2005). In addition, impaired ACE2 expression was observed in mice receiving an intraperitoneal injection of SARS-CoV Spike protein, suggesting that the Spike proteins might worsen lung injury by hijacking ACE2 function and reducing its expression levels (Kuba *et al*, 2005). Thus, ACE2 plays a key role in protecting organs from injury other than serving as the entry receptor for SARS-CoV-2 (Hirano & Murakami, 2020).

Four potential strategies have been developed to combat SARS-CoV-2 entry, including Spike protein-based vaccine, inhibition of transmembrane protease serine 2 (TMPRSS2) activity, therapeutic antibodies blocking the interactions between ACE2 and the Spike protein, and soluble ACE2 (Kruse, 2020; Zhang *et al*, 2020). While it will take time to evaluate the safety and efficacy of the vaccine candidates, it was recently demonstrated that human recombinant soluble ACE2 can significantly block SARS-CoV-2 entry into host cells (Monteil *et al*, 2020). The antibody Fc domain, in addition to its ability to trigger antibody-dependent cellular cytotoxicity (ADCC) and complement-dependent cytotoxicity (CDC), endows the fusion protein with a longer half-life (Czajkowsky *et al*, 2012). Whether the ACE2-Fc fusion protein can retain the peptidase activity of ACE2 and endow the fusion proteins with ADCC and CDC activity while blocking SARS-CoV-2 entry remains to be investigated. Here, we aim to examine the protective role of this shorter ACE2-Fc (18–615 A.A.) fragment on SARS-CoV-2 infection.

## Results

### Production and functional assay of the ACE2-Fc decoy antibody

Here, we designed three SARS-CoV-2 Spike protein constructs, including ectodomain residues 1–1,273 (full length), 1–674 (S1), and 319–591 (receptor-binding domain, RBD), all fused with the Fc region of human IgG1 or a flag tag at the C-terminus for further

Western blot analysis or enzyme-linked immunosorbent assay (ELISA) (Fig 1A and B). The ectodomain of human ACE2, residues 18–615, was also constructed with a fused Fc region of human IgG1 at the C-terminus and IL-2 secretion signaling peptide at the N-terminus to facilitate secretion of ACE2-Fc out of the cells (Fig EV1A). This soluble ACE2-Fc chimeric region could be specifically recognized by the anti-ACE2 antibody (Fig EV1B). The ACE2-Fc and Spike 1–674-Fc can individually form a stable homodimer (Fig 1C and D). The ACE2-Fc and Spike 1–674-Fc protein are likely to be heavily N-glycosylated since size reduction was observed in SDS–PAGE after PNGase F (Peptide: N-glycosidase F) treatment (Fig 1E and F). ELISA was subsequently performed to examine whether this purified ACE2-Fc antibody could bind to the SARS-CoV-2 Spike. ACE2-Fc was further modified with monomer D-biotin at its C-terminus by an Avi-tag affinity process (Fig EV1C). As shown in Fig EV2A and B, ACE2-Fc-Biotin, as the functional receptor, could interact with the Spike 1–674 and the Spike 319–591 truncated proteins. The ACE2-Fc-Biotin/Spike S1 interaction could be disrupted by a 20-fold excess of unlabeled ACE2-Fc or the Spike S1 subunit in a dose-dependent manner (Figs 2A and EV2C). In addition, ACE2-Fc could bind to the cell surface of human lung adenocarcinoma H1975 cells expressing full-length Spike protein in a dose-dependent manner (Fig 2B). The colocalization of the FITC-conjugated ACE2-Fc and anti-Spike antibody by confocal microscopy further confirm the specific recognition of the Spike proteins by the ACE2-Fc (Fig 2C). Using a fluorescent peptide substrate (Mca-Tyr-Val-Ala-Asp-Ala-Pro-Lys(Dnp)-OH) (Enari *et al*, 1996), we demonstrated that the purified ACE2-Fc retained peptidase activity as compared with the human normal IgG and buffer controls (Fig 2D). The effects of ACE2-Fc on Ang II-mediated inflammatory cascade were subsequently investigated. After the co-incubation of Ang II with ACE2-Fc, we observed that ACE2-Fc could significantly suppress Ang II-induced TNF-α production (Fig 2E) and phosphorylation of ADAM17 (a disintegrin and metalloprotease 17) (Fig 2F). Taken together, we demonstrated that the ACE2-Fc decoy antibody preserves the peptidase activity of ACE2-Fc to reduce the Ang II-mediated cytokine cascade while exhibiting specific interaction with the SARS-CoV-2 Spike proteins.

### ACE2-Fc inhibits SARS-CoV-2 Spike-mediated cell–cell fusion and syncytia formation

To determine whether the decoy antibody is able to inhibit SARS-CoV-2 fusion with the target cells, we cotransfected SARS-CoV-2 Spike protein and EGFP into the HEK293T cells as the effector cells (293T-S) and used the ACE2-stable-expressing HEK293T and H1975

---

**Figure 1. Production of the decoy antibody (chimeric ACE2-Fc).**

A   Schematic diagram of SARS-CoV-2 Spike protein and the Spike constructs, Spike 1–1,273 (full length), 1–674 (S1), and 319–591 (RBD-SD1) used in this study. The arrows indicate the cleavage sites of furin and TMPRSS2.

B   Western blot analysis of SARS-CoV-2 Spike 1–1,273, 1–674, and 319–591 expression in HEK293T cells. The black arrows indicate the location of the induced target proteins.

C   Detection of decoy antibody ACE2-Fc and Spike S1-Fc chimeric homodimer formation in nonreducing Coomassie Brilliant Blue staining.

D   Recognition of the ACE2-Fc and Spike S1-Fc by anti-human IgG Fc antibody in nonreducing sodium dodecyl sulfate–polyacrylamide gel electrophoresis (SDS–PAGE).

E, F   Deglycosylation of ACE2-Fc and Spike 1-674-Fc by PNGase F. PNGase F digested ACE2-Fc (500 ng) and Spike 1-674-Fc (500 ng) were subjected to Coomassie Brilliant Blue staining (E) and Western blot analysis by anti-human IgG Fc antibody (F).

Data information: NTD: N-terminal domain; RBD: receptor-binding domain; SD: connector domain; TM: transmembrane domain; CT: cytoplasmic tail; FP: fusion peptide. IB, immunoblotted with the indicated antibodies. GAPDH served as a loading control. Experiments were performed at least three times with similar results.

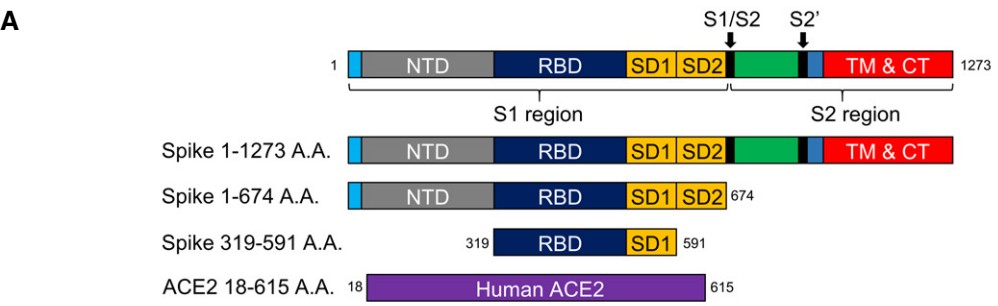

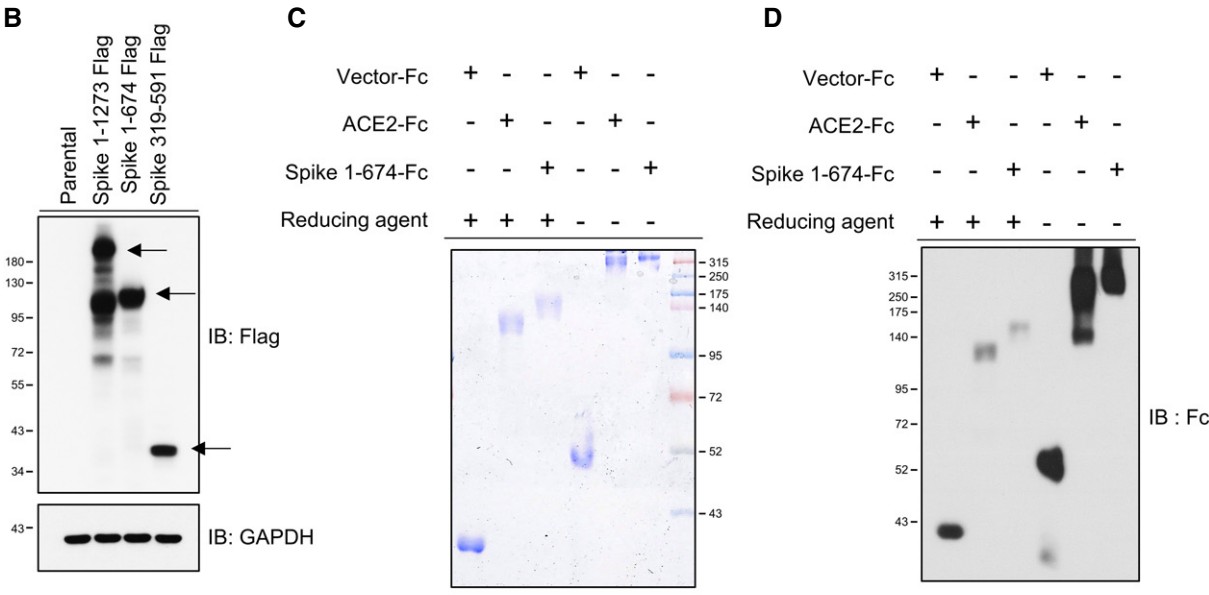

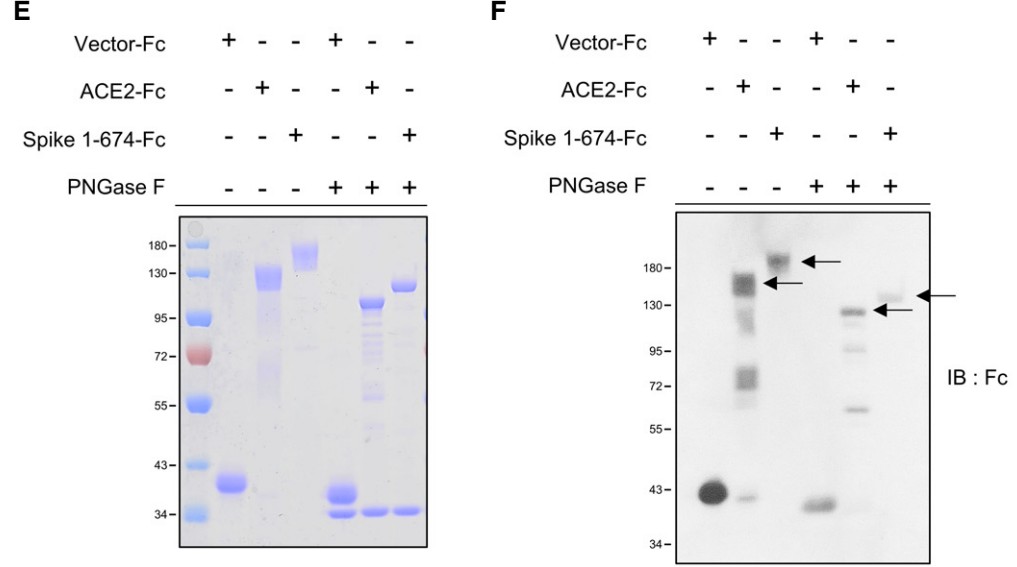

**Figure 1.**

cells as the target cells (293T-ACE2 and H1975-ACE2) (Fig 3B). The target cells without ACE2 overexpression were used as controls. The effector cells (293T-S) were preincubated with ACE2-Fc or IgG at 37°C for 1 h before mixing with the target cells or control cells for another 4 h (cell–cell fusion assay) or 24 h (syncytia formation assay) (Fig 3A). The area of EGFP was counted, and the inhibitory effect of ACE2-Fc on SARS-CoV-2 Spike-mediated cell–cell fusion and syncytia formation was quantified ($n = 6$). As shown in Fig 3C and D, ACE2-Fc significantly impaired SARS-CoV-2 Spike-mediated cell–cell fusion and syncytia formation compared to the normal human IgG control in both the HEK293T and the H1975 cell systems. In addition, a dose-dependent inhibition of cell–cell fusion or syncytia was observed at increasing doses of ACE2-Fc in H1975 cell system ($P$-values between each dose of ACE2-Fc were shown in Appendix Table S1). These results demonstrated that ACE2-Fc could block SARS-CoV-2 infection via abrogating virus-mediated cell–cell fusion and syncytium formation.

## Cytotoxicity and stability of ACE2-Fc

To examine the potential cytotoxicity of ACE2-Fc on normal cells, two different normal human bronchial epithelial (NBE) cells were treated with various concentrations of ACE2-Fc or IgG for 3 days before the cell viability assay. As shown in Fig EV3A and B, no cell toxicity was observed in these two normal cells at the concentration up to 400 μg/ml of ACE2-Fc. The stability of ACE2-Fc in serum was subsequently determined. We incubated 2 μg/ml ACE2-Fc in 50% normal human serum at 37°C for 0, 1, 2, and up to 10 days. The stability of ACE2-Fc was determined by assaying its binding ability to the Spike proteins in the ELISA assay. As shown in Fig EV3C, not a significant reduction of ACE2-Fc/Spike binding was observed up to 10 days. These results suggest that ACE2-Fc has no toxicity to epithelial cells and may be stable in serum for 10 days, which may facilitate its future clinical application.

## ACE2-Fc blocks entry of pseudotyped lentivirus into ACE2-expressing cells and lung organoids

The ability of ACE2-Fc to block SARS-CoV-2 entry was first examined by using the Spike-expressing pseudotyped lentivirus whose backbone G protein of vesicular stomatitis virus (VSV-G) was replaced with the SARS-CoV-2 Spike protein. The pseudotyped virus was preincubated with either ACE2-Fc or human IgG1 for 1 h at 37°C before adding to the ACE2-expressing HEK293T cells (293T-ACE2) or parental HEK293T cells for another hour. The entry of Spike-expressing pseudotyped virus into host cells was specifically mediated by ACE2 expression (Fig 4A). The dose-dependent block-age of viral entry by ACE2-Fc was not only observed in HEK293T cells but also in another ACE2-expressing H1975 cell (H1975-ACE2) (Fig 4A) ($P$-values between each dose of ACE2-Fc were shown in Appendix Table S1). Similar neutralization effects were also observed at a 10-fold higher virus input (Fig EV4).

Since the lung is the primary site for SARS-CoV-2 infection, we established an airway organoid model following the methods described in a recent article (Sachs *et al*, 2019) to investigate the neutralization ability of ACE2-Fc against SARS-CoV-2 entry. The derived airway differentiation organoids were successfully established and composed of several airway epithelial cells with specific markers, including basal (P63), secretory (club cell marker secretoglobin family 1A member 1 (SCGB1A1) and secretory cell marker mucin 5AC (MUC5AC)), and multiciliated cells (cilia marker acetylated α-tubulin) (Fig 4B). Of note, unlike the A549 or the parental human normal bronchial epithelial cells (HBEpc), these airway organoids expressed a high-level of ACE2 in addition to TMPRSS2 (Fig 4B and C). We then examined the neutralization ability of ACE2-Fc for Spike-expressing pseudotyped virus in the airway organoid model. As shown in Fig 4D, the airway organoids are susceptible to virus entry and the ACE2-Fc could significantly block virus entry at the concentration of 100 μg/ml. Our findings indicated that ACE2-Fc can inhibit entry of SARS-CoV-2 Spike-expressing pseudotyped virus entry into ACE2-expressing cells, including the airway organoids.

## ACE2-Fc blocks SARS-CoV-2 entry and replication

The inhibitory effects of ACE2-Fc on viral entry were further confirmed using real SARS-CoV-2 isolates from patients suffering from COVID-19 infection at National Taiwan University Hospital. As expected, the preincubation of SARS-CoV-2 with ACE2-Fc blocked plaque formation in Vero E6 cells in a dose-dependent manner (Figs 5A and EV5) ($P$-values between each dose of ACE2-Fc were shown in Appendix Table S1). The half-maximal effective

---

**Figure 2. Functional characterization of the decoy antibody.**

A   Competitive ELISA. ACE2-Fc can compete with ACE2-Fc-Biotin for Spike 1-674 binding in a dose-dependent manner. The competition effects of ACE2-Fc on ACE2-Spike protein binding were normalized to PBS control. Error bars represent the standard deviation (SD), $n = 2$. Statistical analysis was performed by unpaired two-tail $t$-test. **$P < 0.01$, ***$P < 0.001$.

B   Recognition of full-length Spike on H1975 (left panel as negative control) or H1975-Spike (right panel) cells by ACE2-Fc using flow cytometry analysis. Isotype control: 40 μg/ml mouse IgG-FITC.

C   Confocal microscopy of H1975-Spike-overexpressing cells. Spikes on H1975 cells were stained with anti-Spike antibody and Alexa Fluor® 594-conjugated secondary antibody on ice for 1 h. After that, ACE2-Fc-FITC was incubated for another 1 h. Scale bars equal to 20 μm. DAPI was used as a nuclear counterstain.

D   Preservation of ACE2-Fc peptidase activity. The peptidase activity of ACE2-Fc was measured by cleavage of the fluorescent peptide substrates.

E   Inhibition of angiotensin II (Ang II)-induced TNF-α production by ACE2-Fc. Ang II was preincubated with indicated amounts of ACE2-Fc or IgG for 30 min. After that, the mixtures were added to RAW264.7 cells for 12 h. The concentration of TNF-α in the culture medium was determined by ELISA. Error bars represent the standard deviation (SD), $n = 3$. Statistical analysis was performed by unpaired two-tail $t$-test. *$P < 0.05$. The dotted line represents the mean value of TNF-α in control group.

F   Inhibition of Ang II-induced ADAM17 (a disintegrin and metalloprotease 17) phosphorylation by ACE2-Fc. The protein extracts from (E) were immunoblotted with the indicated antibodies (left panel). β-actin served as the loading control. The signal intensity was normalized to cells only (right panel). Data are representative of three independent experiments, and the values are expressed as the mean ± SD (right panel). Error bars represent the standard deviation (SD), $n = 2$. Statistical analysis was performed by unpaired two-tail $t$-test. *$P < 0.05$. IgG represents the human normal IgG control.

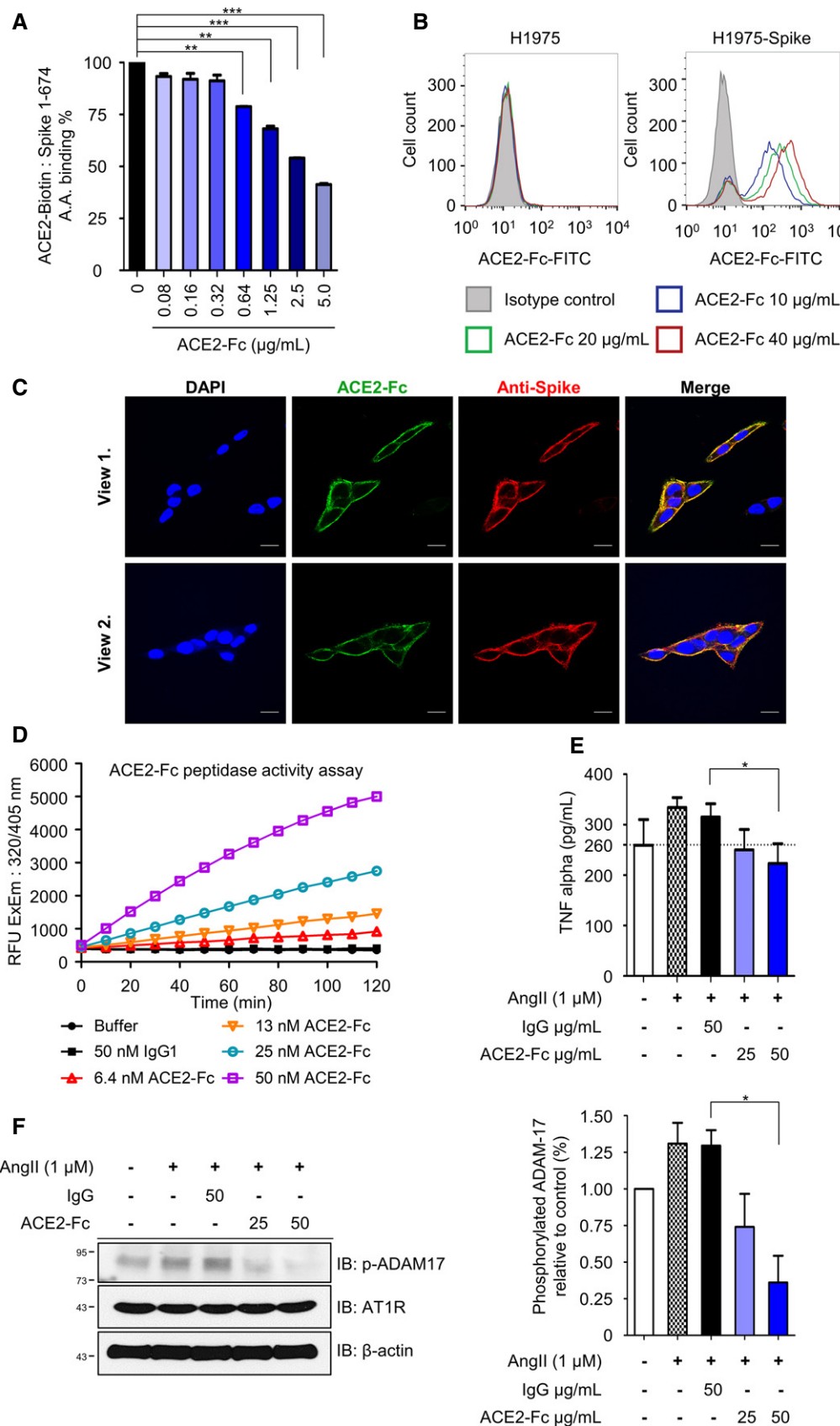

**Figure 2.**

concentrations (EC50) value for ACE2-Fc were 23.8 ± 5.94 µg/ml. The inhibitory effect was subsequently verified by the yield reduction assay. Pretreatment of SARS-CoV-2 with ACE2-Fc dose-dependently reduced the SARS-CoV-2 RNA copies in the culture supernatant (Fig 5B) and the SARS-CoV-2 nucleoprotein expression in the infected cells (Fig 5C). In addition, we extended the incubation period of virus-ACE2-Fc from 1 to 48 h to examine whether resistant viruses would emerge (Fig 5D). Notably, comparable inhibitory effects on viral protein expression and supernatant viral RNA were observed when the ACE2-Fc was present in the culture medium for 48 h as compared to the pretreatment group (Fig 5E and F).

To determine whether ACE2-Fc also exhibit inhibitory effects on other circulating SARS-CoV-2 strains, five other clinical SARS-CoV-2 strains, NTU3, NTU13, NTU14, NTU25, and NTU27, were included for analysis, and their genetic characteristics were summarized in Table EV1. Notably, NTU3, NTU14, and NTU25 strains harbor the D614G mutation, which has been reported to increase the viral infectivity (Korber et al, 2020; Yurkovetskiy et al, 2020). As shown in Fig 6, ACE2-Fc exhibited a potent ability to block SARS-CoV-2 protein expression (Fig 6A) and viral RNA in the supernatants and infected cells (Fig 6B and C). In this study, we further showed that our decoy antibody could block entry of various SARS-CoV-2 strains and no resistant virus could be selected after prolonged co-incubation of ACE2-Fc and SARS-CoV-2.

### ACE2-Fc induced degranulation of natural killer (NK) cells

Proteins fused to the Fc domain enable these molecules to interact with Fc receptors, which are critical for the induction of immune responses (Czajkowsky et al, 2012). Amon these, antibody-dependent cellular cytotoxicity (ADCC) is an adaptive immune response, mainly mediated by the natural killer (NK) cells. After the specific interactions between antibodies or Fc fusion proteins with specific antigens on the cell surface of infected cells, cross-linking of the Fc domain will activate the CD16 (FcγRIII) receptor to trigger degranulation (CD107a on the cell membrane) and cytokine production (IFN-γ and TNF-α) of NK cells (McDonald et al, 2015; Sun et al, 2019). A recent report showed that the RBD-specific antibodies from an individual infected with SARS-CoV in 2003 could induce nearly 10% ADCC against SARS-CoV-2 (Pinto et al, 2020). Therefore, experiments were conducted to examine whether ACE2-Fc could induce primary human NK cell activation. H1975 cells, transduced with full-length Spike by the lentiviral vector (H1975-Spike), were used as target cells (Fig 7A). The NK cell degranulation assay was performed to determine the CD107a, IFN-γ, and TNF-α expression levels after

the co-incubation of the NK cells with H1975-Spike cells in the presence of ACE2-Fc or recombinant ACE2 (1–740 A.A. without an Fc tag). Induction of the expression levels of these three degranulation markers was observed when serially diluted ACE2-Fc was added to the coculture of NK and H1975-Spike cells (Fig 7B–D). In contrast, the degranulation of NK cells was not observed in the presence of recombinant ACE2. Taken together, our results suggest that ACE2-Fc could not only block SARS-CoV-2 infection but also induce NK cell activation, which may help to remove the infected cells in vivo.

## Discussion

Here, we provided evidence that a decoy antibody (ACE2-Fc) significantly reduced SARS-CoV-2 infection, while retained the peptidase activity and the ability to suppress the inflammatory responses induced by angiotensin II (Fig 8). Chimeric ACE2-Fc antibody has been shown to have a longer elimination phase half-life of 174.2 h as compared to the 1.8 h for the recombinant ACE2 (rACE2) by the pharmacokinetic study in mice (Liu et al, 2018). In this study, we further confirmed that ACE2-Fc retained nearly 100% of its Spike-binding ability after 10 days of incubation at 37°C (Fig EV3C).

The serum concentration of Ang II is increased and acts as the key component of pro-inflammatory cytokines when SARS-CoV hijacks cellular ACE2 (Kuba et al, 2005). After Ang II binds to its receptor AT1R, it can activate ADAM17 and NF-κB (Esteban et al, 2004). ADAM17 phosphorylation can further facilitate the maturation of IL-6 and TNF-α from their precursor forms (Hirano & Murakami, 2020). Therefore, SARS-CoV-2 infection might cause cytokine storms in COVID-19 patients, which is likely to be prevented by the restoration of ACE2 peptidase activity. In our study, we demonstrated that expression of a shorter ACE2 fragment (18–615 A.A.) can specifically and efficiently block SARS-CoV-1 infection, while still preserving its peptidase activity (Fig 2D–F). Based on the previous findings in mouse ACE2 (Wysocki et al, 2019), two shorten mouse ACE2 variants (1–605 and 1–619 A.A.) were shown to exhibit higher ACE2 enzyme activity than that of the full-length ACE2 (1–740 A.A.). In addition, ACE2 enzyme activity could be detected in urine from ACE2 knockout mice after intravenous infusion of the mouse variant (1–619 A.A.), but not ACE2 (1–740 A.A.). Infusion of this short variant (ACE2 1–619 A.A.) also recovered the ACE2 activity of the kidney in the Ace2-deficient mice. These evidences supported that the shortened version of the mouse ACE2-Fc (1–619 A.A.) is more stable in plasma and retains higher enzyme activity to convert Ang II to Ang 1–7. The decoy antibody ACE2-Fc designed in this study is a shorten version of ACE2 (18–615), similar

**Figure 3. Inhibition of Spike-induced cell–cell fusion and syncytia formation by ACE2-Fc.**

A  The schematic diagram for cell–cell fusion and syncytia formation.

B  HEK293 and H1975 cells were transduced with full-length ACE2 by lentivirus. Protein extracts were immunoblotted with the indicated antibodies. O/E represents overexpression.

C  GFP and full-length Spike cotransfected HEK293T cells were used as effector cells (293-S). HEK293/ACE2 (HEK293T-overexpressing ACE2) (left panel) or H1975/ACE2 (H1975-overexpressing ACE2) (right panel) cells were used as target cells. The 293-S cells were preincubated with normal human IgG or ACE2-Fc at 37°C for 1 h. After that, the mixtures were added to target cells for 4 h (cell–cell fusion) or 24 h (syncytia formation). The fluorescent areas were measured by inverted fluorescence microscopy and Metamorph software (Metamorphosis). Scale bars equal to 50 µm. The white arrows indicate the cell–cell fusion or syncytia formation.

D  The inhibition of cell–cell fusion and syncytia formation by ACE2-Fc in HEK293/ACE2 and H1975/ACE2 was determined with the formula described in the Materials and Methods section. Error bars represent the standard deviation (SD), n = 6. Statistical analysis was performed by unpaired two-tail t-test. **P < 0.01, ***P < 0.001.

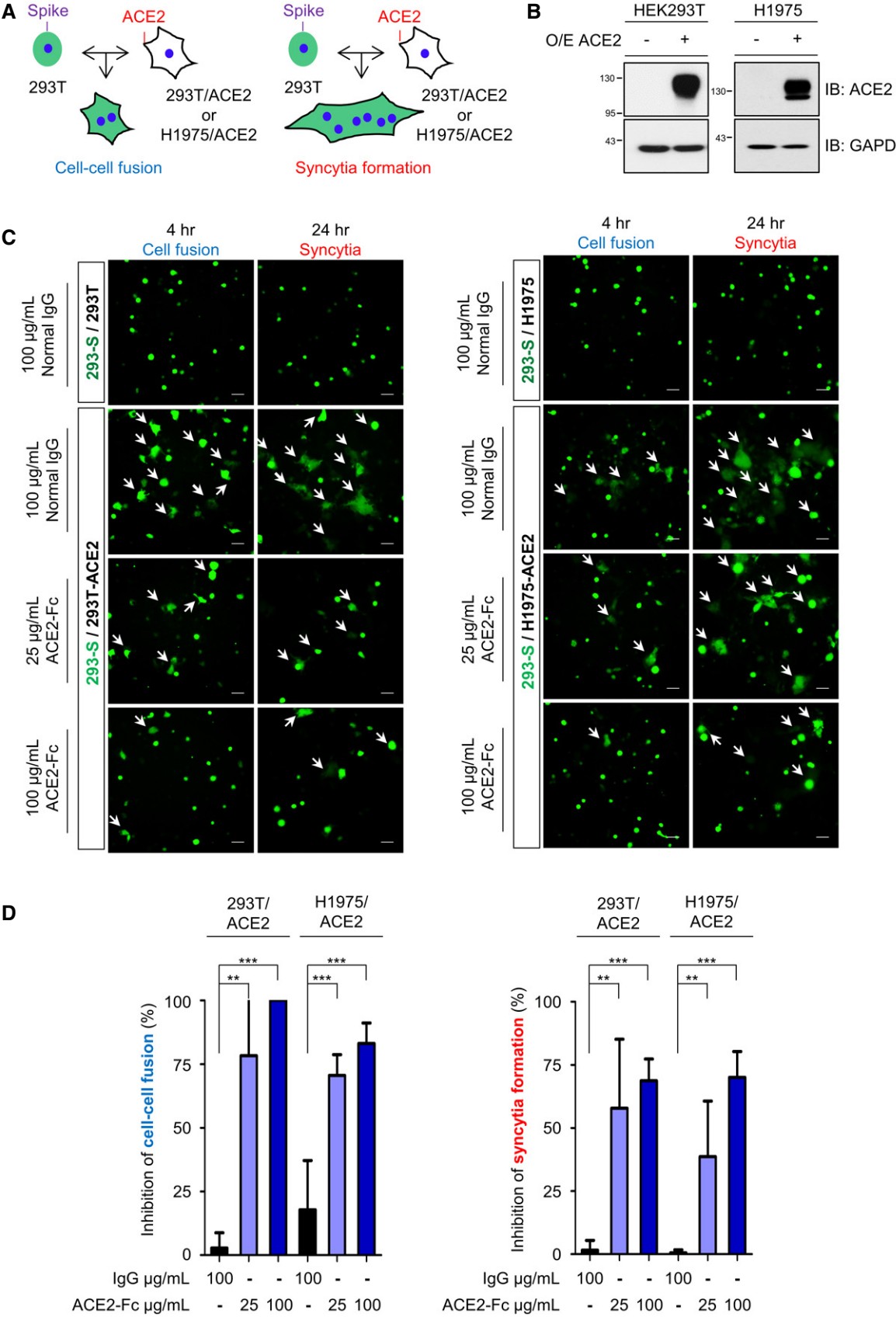

**Figure 3.**

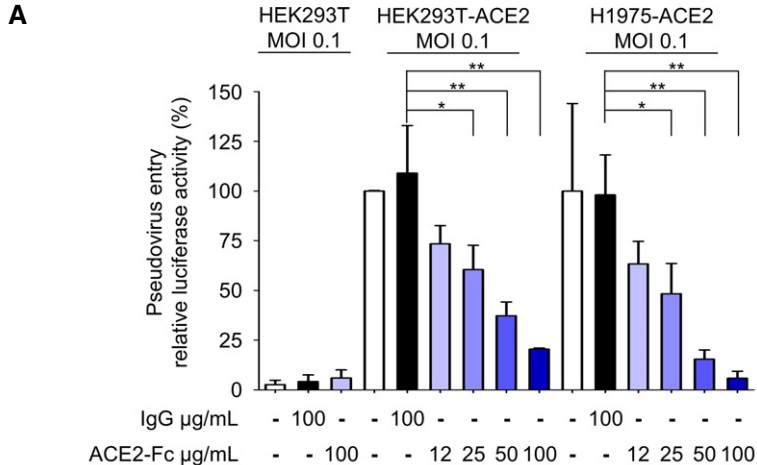

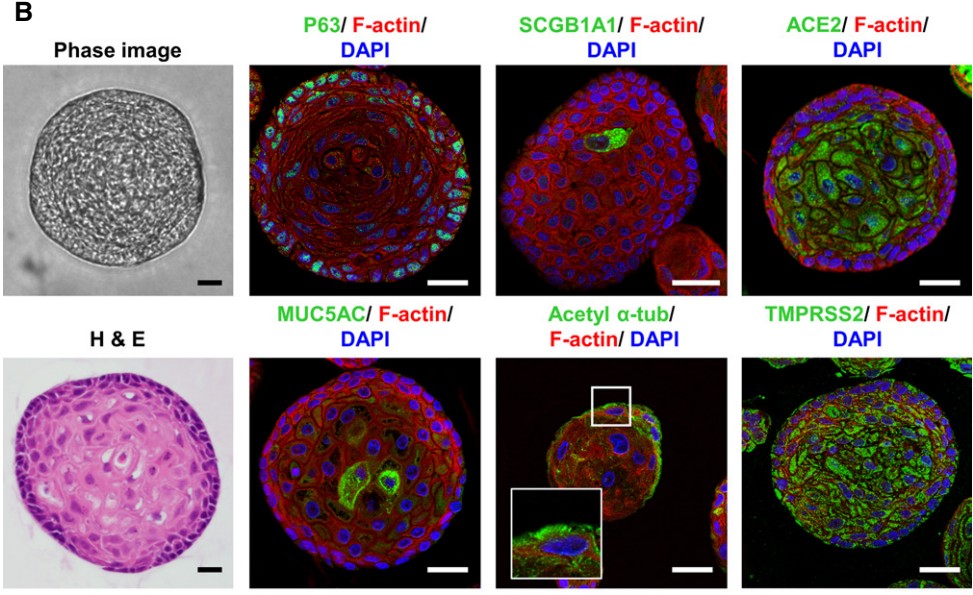

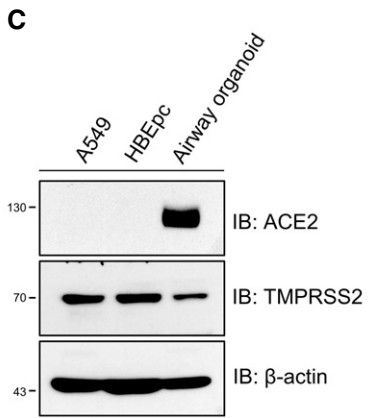

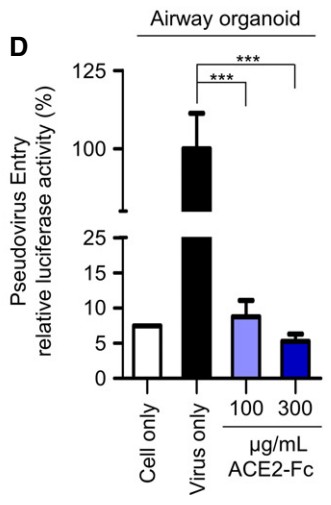

**Figure 4.**

**Figure 4.  Blockage of Spike-expressing pseudovirus entry into ACE2-expressing cells by ACE2-Fc.**

A     ACE2-Fc blocked the entry of Spike-expressing pseudotyped lentivirus into HEK293T-ACE2 and H1975-ACE2 cells. The relative luciferase activities, normalized to the only virus group, represent the efficiency of virus entry. MOI: Multiplicity of infection.

B–D   Airway organoid model system. (B) Confocal microscopy images of the airway organoids. The green fluorescence represents the specific staining of indicated monoclonal antibodies and Alexa Fluor 488-conjugated secondary antibodies. DAPI was used as a nuclear counterstain. Hematoxylin and eosin (H&E) stain: hematoxylin stained the nuclei in blue color; eosin stained the cytoplasm in pink color. Scale bars equal to 20 μm. (C) Lung cancer A549 cells, human normal bronchial epithelial cells (HBEpc), and HBEpc-differentiated cells (airway organoids) were harvested for immunoblotting with the indicated antibodies. (D) Blockage of pseudovirus entry into airway organoids by ACE2-Fc. Mixtures of pseudotyped lentivirus with or without ACE2-Fc were cocultured with airway organoids for 72 h. The virus entry was determined by quantifying the luciferase activity in the cell lysates.

Data information: Error bars represent the standard deviation (SD), n = 3. Statistical analysis was performed by unpaired two-tail t-test. *P < 0.05, **P < 0.01, ***P < 0.001. Experiments were performed at least three times with similar results.

to that in the mouse ACE2 study (Wysocki *et al*, 2019). Our study results clearly demonstrated that in addition to block the entry and subsequent infection of host cells by the SARS-CoV-2 pseudoviruses as well as the SARS-CoV-2 clinical isolates, the ACE2-Fc (18–615 A.A.) fusion protein is very stable in plasma up to 10 days and also preserve its peptidase activity to exert its regulator effects on immune responses (Figs 6 and 7).

The renin–angiotensin system (RAS) is a hormone system that regulates vascular function, including the regulation of blood pressure, natriuresis, and blood volume control (Tikellis & Thomas, 2012). As a key regulator in the RAS, ACE2 acts by converting Ang II into Ang 1–7 (Gheblawi *et al*, 2020). Ang II can also downregulate ACE2 expression through AT1R when the RAS is overactivated in cardiovascular disease through the ERK1/2 and p38 MAPK signaling pathways (Koka *et al*, 2008). Three strategies have been evolved to reduce Ang II levels upon RAS overactivation, including ACE inhibitors (ACEis), angiotensin receptor blockers (ARBs), and mineralocorticoid receptor blockers (Gheblawi *et al*, 2020). Most of these compounds increase the protein and mRNA levels of ACE2 or peptidase activity in animal models (Kai & Kai, 2020). Being known as the receptor for SARS-CoV-2, we speculate that ARBs- or ACEis-induced ACE2 upregulation may facilitate SARS-CoV-2 infection. In addition, the administration of recombinant ACE2 or ACE2-Fc could augment ACE2 enzymatic activity and reduce systemic inflammation. A lower plasma Ang II level after taking recombinant ACE2 may be safe in healthy volunteers and patients with acute respiratory distress syndrome or pulmonary arterial hypertension as demonstrated in the previous pilot studies (Haschke *et al*, 2013; Khan *et al*, 2017; Hemnes *et al*, 2018). Therefore, for RAS disorder patients with COVID-19 treatment, administration of ACE2-Fc in circulation may block SARS-CoV-2 infection even in those subjects with induced ACE2 after receipt of ARBs or ACEis treatment. Nevertheless, it should be noted that similar to ARBs and ACEis, administration of ACE2-Fc may be harmful to the developing fetus during pregnancy since it could lower blood pressure by converting Ang II to Ang 1–7.

The ADCC process is initiated by the interaction of the antibody with the virus antigen of infected cells, which subsequently leads to the cross-linking of the CD16 receptors on NK cells. The degranulation and cytokine production is triggered upon NK cell activation, which then results in the elimination of infected cells (Sun *et al*, 2019). Both ADCC and virus neutralization are important in controlling disease progression in patients with HIV or influenza virus H1N1 infection (Jegaskanda *et al*, 2013; Wren *et al*, 2013). When the neutralizing antibody is not sufficient to completely block viral infection, a secondary infection may eventually occur and lead to severe clinical consequences. In our study, we demonstrated that ACE2-Fc could significantly neutralize SARS-CoV-2 infection and replication by either pretreatment or full-time treatment *in vitro* (Fig 5). In addition, we showed that the decoy antibody (ACE2-Fc), but not ACE2 (1–740 A.A. without a fused Fc domain), could significantly activate degranulation of NK cells from three independent donors (Fig 7). These results implicate that the ACE2-Fc can not only neutralize viral entry but also activate NK cells to remove the SARS-CoV-2 infected cells.

The SARS-CoV-2 Spike protein encodes 22 potential N-linked oligosaccharides per protomer, which might play a role in epitope masking and possibly immune evasion (Watanabe *et al*, 2020). Notably, the presence of the A475V, L452R, V483A, and F490L mutations in the RBD domain of SARS-CoV-2 was shown to decrease the neutralizing activity of antibody and might impede the development of therapeutic antibodies (Li *et al*, 2020). In a recent report (Korber *et al*, 2020), the SARS-CoV-2 bearing the D614G

**Figure 5.  Blockage of SARS-CoV-2 entry into host cells by ACE2-Fc.**

A     Inhibition of SARS-CoV-2 infection by ACE2-Fc in the plaque assay. Mixtures of ACE2-Fc and SARS-CoV-2 were incubated for 1 h before adding to Vero E6 cells for another 1 h at 37°C. The ACE2-Fc and SARS-CoV-2 premixtures were removed, and the cells were washed once with PBS and overlaid with methylcellulose with 2% FBS for 5–7 days before being stained with crystal violet. Those results showed increase of plaque formation was regarded as no inhibition of plaque formation. TPCK-treated trypsin: N-tosyl-L-phenylalanine chloromethyl ketone-treated trypsin.

B, C   Yield reduction assay was performed to determine the inhibitory effects of ACE2-Fc on virus titers and protein expression. The culture medium and cell extracts were harvested 24 h postinfection for real-time PCR (B) and Western blot (C). The NP/PCNA represents the relative NP expression as compared to that of PCNA, which served as a loading control. The NP/PCNA numbers below the panel are the ratios of NP/PCNA normalized to that of the human IgG control group. PCNA: Proliferating cell nuclear antigen.

D     Schematic illustration of the ACE2-Fc pretreatment and full-time experiment procedure, delineating the stages where the ACE2-Fc was present during the experiment.

E     The cell lysates from the pretreatment and full-time experiments were harvested and immunoblotted with the indicated antibodies.

F     The SARS-CoV-2 titer in the pretreatment and full-time experiments was analyzed by real-time PCR.

Data information: Error bars represent the standard deviation (SD), n = 3. Statistical analysis was performed by unpaired two-tail t-test. *P < 0.05, **P < 0.01.

**A**

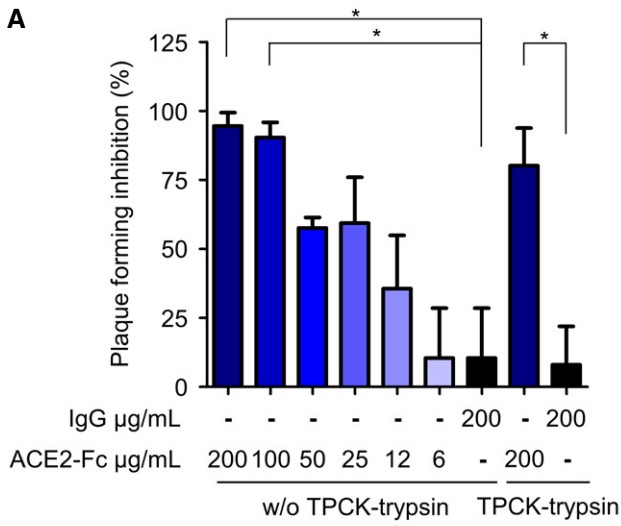

**B**

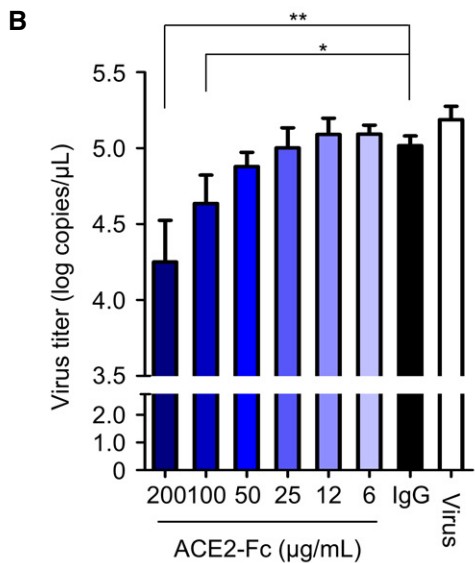

**C**

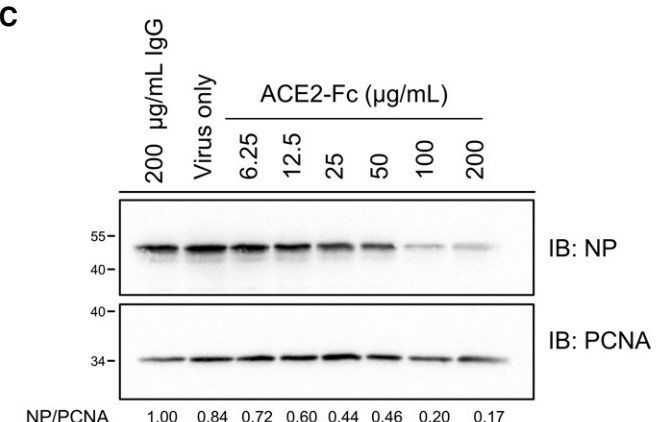

**D**

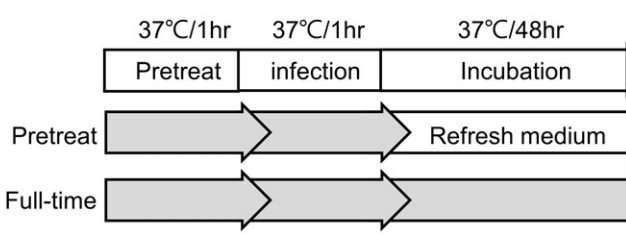

**E**

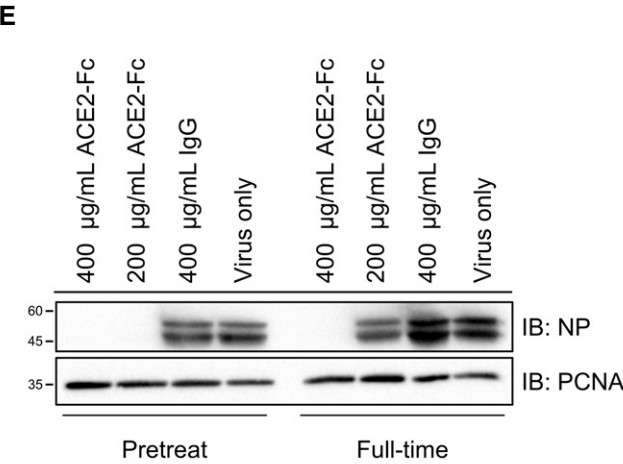

**F**

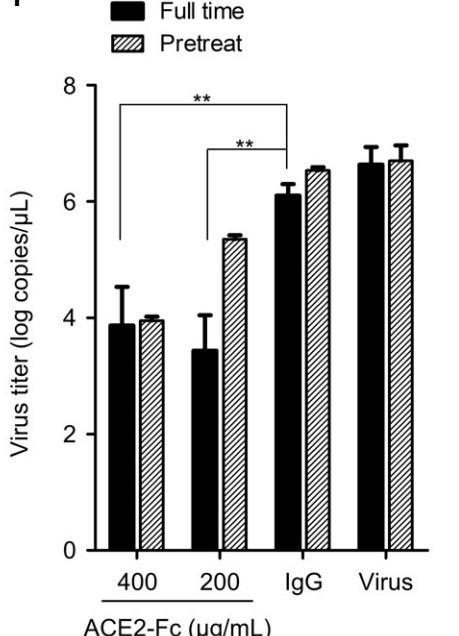

Figure 5.

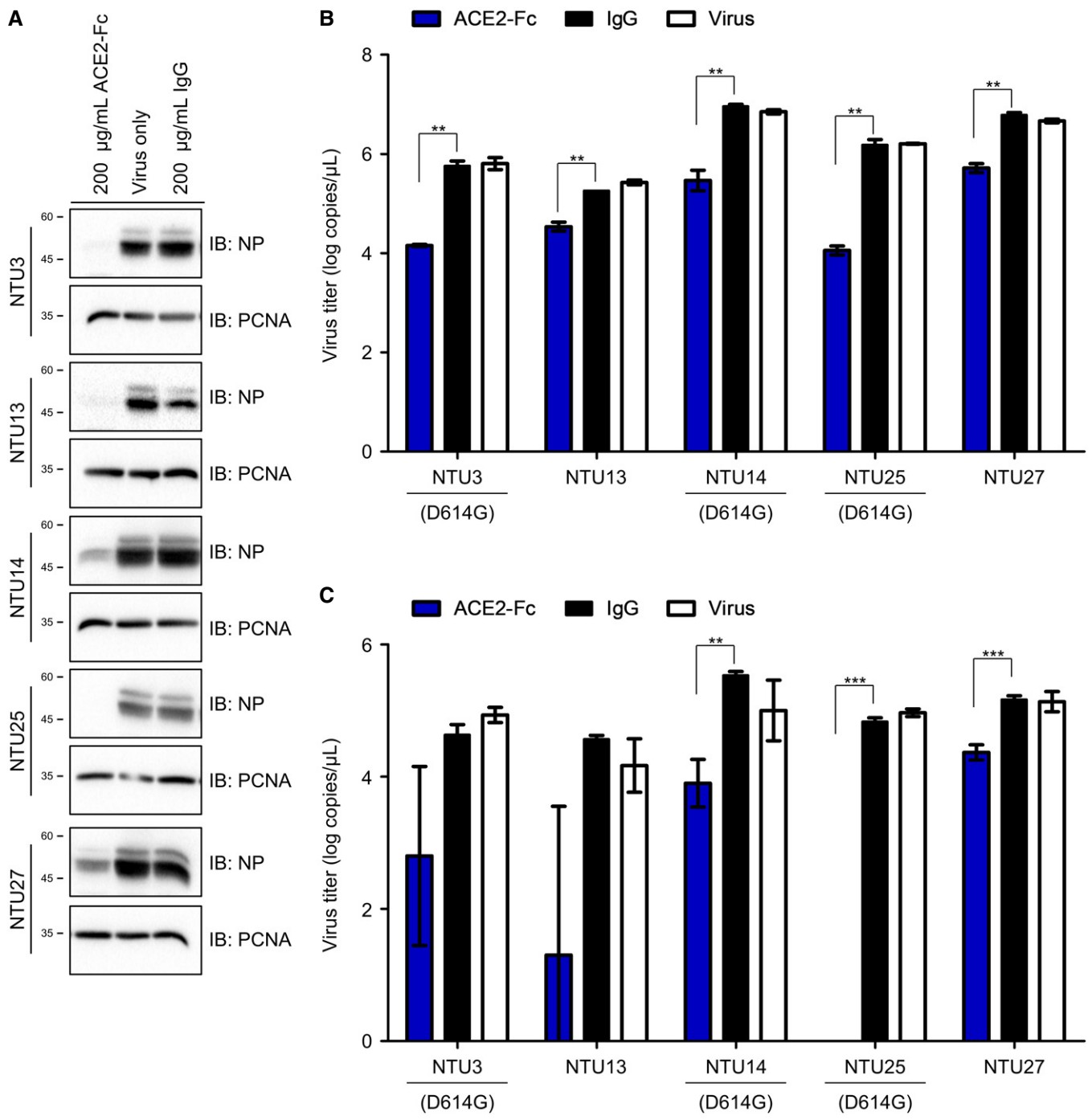

**Figure 6.  Neutralization activity of ACE2-Fc on different SARS-CoV-2 strains.**

Yield reduction assay was performed to determine the inhibitory effects of ACE2-Fc on the entry of 5 different SARS-CoV-2 strains into Vero E6 cells.

A     The (A) NP proteins in the cell lysates were determined by Western blot analysis.

B, C   The virus RNA in the (B) culture medium and (C) cell lysates was quantified by real-time RT–PCR. The genetic information and sequence reference numbers for the 5 SARS-CoV-2 strains were summarized in Table EV1. Error bars represent the standard deviation (SD), $n = 3$. Statistical analysis was performed by unpaired two-tail $t$-test. **$P < 0.01$, ***$P < 0.001$. Experiments were performed at least three times with similar results.

mutation in Spike was adapted during the human-to-human transmission and has become the pandemic strain. The SARS-CoV-2 Spike with D614G pseudotyped virus was shown to exhibit enhanced infectivity compared to their counter strain with D614 at Spike (Korber *et al*, 2020). Three of six of our isolated SARS-CoV-2 strains harbored D614G mutation (Table EV1). Although infection

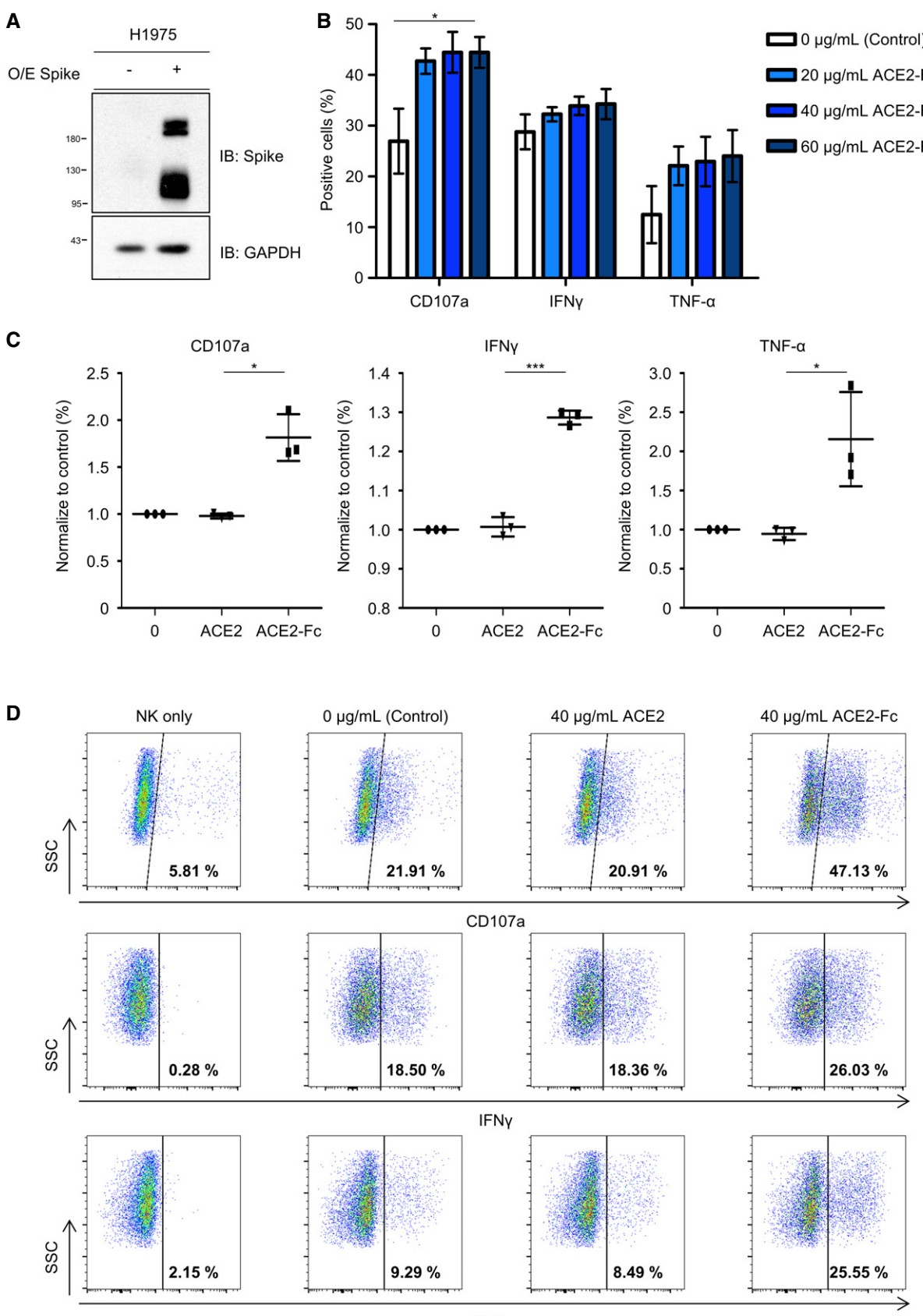

Figure 7.

Figure 7.   Effects of ACE2-Fc on NK cell degranulation.

IL-2-activated NK cells were incubated with H1975-Spike cells at a 1:1 cell ratio in the presence of ACE2-Fc or ACE2. The activation of NK cell was determined by the CD107a, IFN-γ, and TNF-α expression levels.

A   H1975 cells were transduced with full-length Spike by Lentiviral vector. Protein extracts were immunoblotted with the indicated antibodies. O/E represents overexpress.

B   The effect of ACE2-Fc activation on degranulative capacity of NK cells. The experiments were performed with the primary human NK cells derived from three independent donors. Error bars represent the standard deviation (SD), $n$ = 3. Statistical analysis was performed by unpaired two-tail $t$-test. *$P$ < 0.05 (each concentration of ACE2-Fc vs. control).

C   The CD107a, IFN-γ, and TNF-α expression levels were determined by flow cytometry after the NK cells and H1975-Spike cells co-incubation in the presence or absence of ACE2-Fc or ACE2 control. Error bars represent the standard deviation (SD), $n$ = 3. Statistical analysis was performed by unpaired two-tail $t$-test. *$P$ < 0.05. The percentage of positive cells in both groups was normalized to the NK/H1975-Spike only group.

D   The differential expression of degranulation markers after the treatment of ACE2-Fc or ACE2 by the flow cytometry. The numbers in each plot indicate the percentages of positive cells.

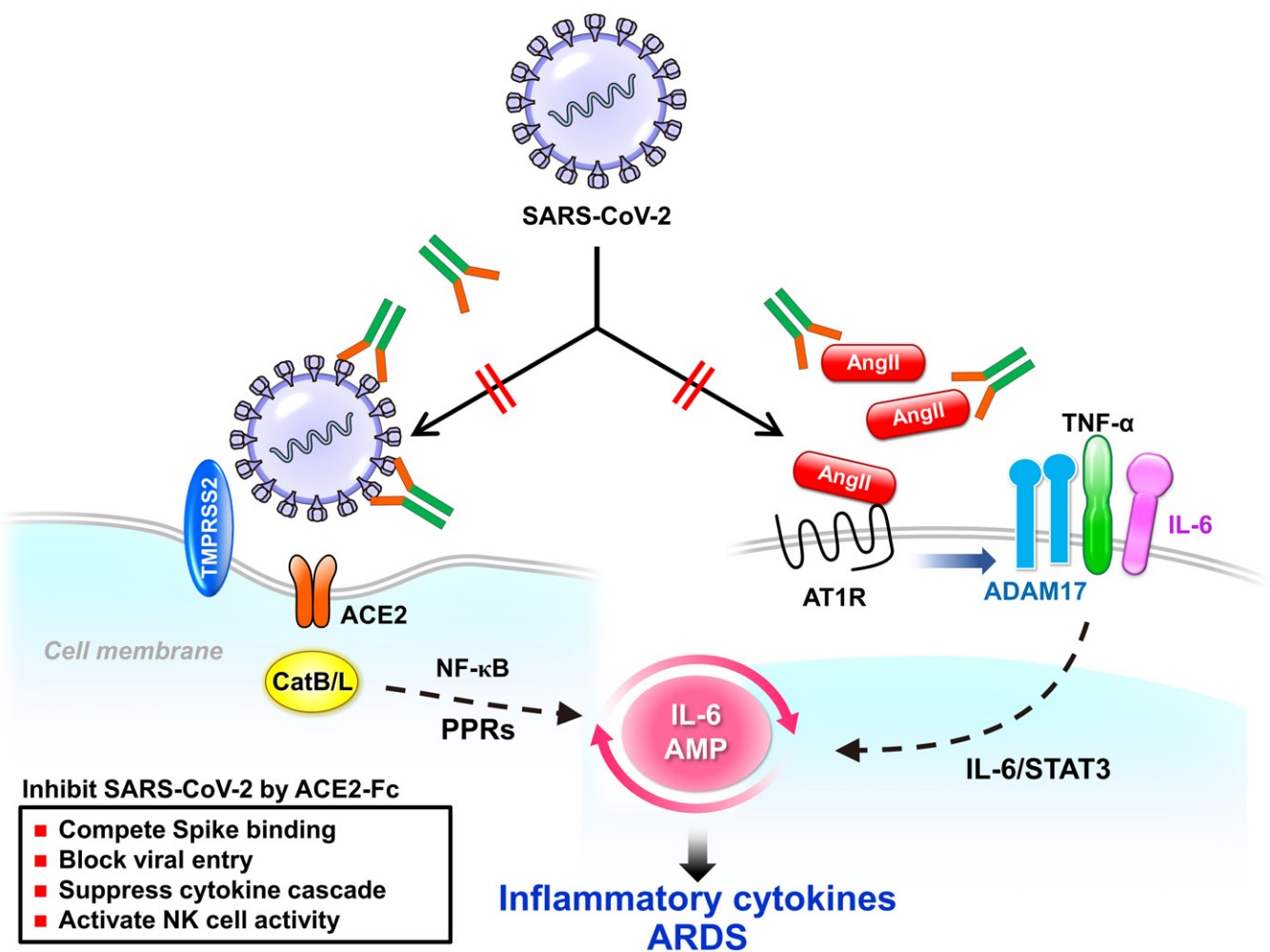

**Figure 8.   Proposed model for the role of the decoy antibody ACE2-Fc in SARS-CoV-2 entry and infection.**

The decoy antibody (ACE2-Fc) not only reduces SARS-CoV-2 infection but also decreases TNF-α secretion and ADAM-17 phosphorylation mediated by angiotensin II. Ang II: angiotensin II; ARDS: acute respiratory distress syndrome; CatB/L: cathepsin B and L; PPRs: pattern recognition receptors; AMP: amplifier; STAT3: Signal transducer and activator of transcription 3; ATIR: angiotensin II type I receptor; ADAM17: a disintegrin and metalloprotease 17; TMPRSS2: transmembrane Serine Protease 2.

of SARS-CoV-2 with D614G was shown to correlate with higher infectivity and viral load in study patients, our results demonstrated that ACE2-Fc could significantly reduce the entry of SARS-CoV-2 with D614G mutation (Fig 6).

Taken together, we demonstrated that ACE2-Fc (18–615 A.A.) has three important characteristics: neutralization of SARS-CoV-2 infection, conversion of Ang II to Ang 1–7, and activation of NK cells. Based on the efficient blocking of SARS-CoV-2 entry, including

the D614G variant strains, in the human epithelial cells and the airway organoids, we believe that the prophylactic use of ACE2-Fc could prevent healthcare workers or people at high risk from SARS-CoV-2 infection. Nevertheless, the potential benefits of Ang II conversion and NK cell activation activities of ACE2-Fc in the COVID-19 disease process require further evaluation in the animal model. Our *in vitro* results suggest that ACE2-Fc has the potential to develop as an effective therapeutic against SARS-CoV-2 infection.

# Materials and Methods

## Generation of the fusion protein

The 18-615 A.A. of ACE2 or 1–1,273, 1–674, and 319–591 A.A. of the SARS-CoV-2 Spike with humanized codons were PCR-amplified and cloned into pCDNA 3.1(-) plasmids with the Fc region of human IgG1 using *Nhe* I and *Sal* I restriction enzymes. The Expi293F system (Thermo Fisher Scientific) was applied to generate recombinant proteins in the culture medium. These soluble recombinant proteins were purified by Protein G Sepharose (Merck). The concentration of recombinant protein was measured at 280 nm by NanoDrop, and the purity was determined by polyacrylamide gel electrophoresis.

## Cell lines

HEK293T and Vero E6 cells were purchased from American Type Culture Collection (ATCC) (Manassas, VA, USA) and cultured in Dulbecco's modified Eagle medium (DMEM) containing 10% fetal bovine serum (FBS) (Life Technologies). The human lung adenocarcinoma cell line H1975 was kindly provided by Dr. James Chih-Hsin Yang (Graduate Institute of Oncology, Cancer Research Center, National Taiwan University) and cultured in RPMI1640 containing 10% FBS. All adherent cells were cultured at 37°C in a humidified atmosphere containing 5% $CO_2$ and 20% $O_2$. According to the manufacturer's recommendation, Expi293F cells were maintained in Expi293 expression medium with a shaking speed of 120 rpm at 37°C.

## Antibodies, immunoprecipitation, and immunoblot

Western blotting was performed as previously described (Huang *et al*, 2016). In brief, total cell lysates were prepared in IP lysis buffer (20 mM Tris, pH 7.5, 100 mM sodium chloride, 1% IGEPAL CA-630, 100 μM $Na_3VO_4$, 50 mM NaF, and 30 mM sodium pyrophosphate) containing complete protease inhibitor cocktail without EDTA (Roche Diagnostics, Basel, Switzerland). Protein concentrations were measured by the Bio-Rad Protein Assay (Bio-Rad, Richmond, CA). The primary antibodies used at a 1:1,000 to 1:10,000 dilutions were as follows: anti-flag (M2, Sigma, 1:10,000); anti-ACE2 (ab108209, Abcam, 1:1,000); anti-TPMRSS2 (sc-515727, Santa Cruz, 1:1,000); anti-Spike (GTX632604, GeneTex, 1:1,000); anti-ADAM17 (T735, ab182630, Abcam, 1:1,000); anti-angiotensin II Type 1 Receptor antibody (ab124734, Abcam, 1:1,000); anti-GAPDH (10494-1-AP, Proteintech, 1:5,000); human FC (l2136, Sigma, 1:10,000); NP: anti-nucleoprotein of SARS-CoV-2 (40143-R019, Sino biological, 1:5,000); and anti-PCNA (Millipore Corporation, 1:5,000). Horseradish peroxidase-conjugated anti-mouse (ab97023, Abcam, 1:5,000) and anti-rabbit (626520, Invitrogen, 1:5,000)

secondary antibodies at the 1:5,000 dilution were used in the analysis. Alexa Fluor® 594 goat anti-mouse IgG (A11032, Thermo Fisher Scientific, 1:500) was used as the secondary antibody for the immunofluorescence experiment. Protein signals were detected by chemiluminescent reagent (NEL105001EA, PerkinElmer).

## ACE2 peptidase activity assay

The enzyme activity of purified ACE2-Fc was measured using the fluorescent peptide substrate Mca-Tyr-Val-Ala-Asp-Ala-Pro-Lys (Dnp)-OH (Mca: (7-Methoxycoumarin-4-yl) acetyl, Dnp: 2,4-Dinitrophenyl) (ES007, R&D). In brief, ACE2-Fc was twofold serially diluted starting at 50 nM using the reaction buffer (50 mM MES, 300 mM NaCl, 10 μM $ZnCl_2$, 0.01% Brij-35, and pH 6.5). Five microliters of 1 mM peptide substrate was premixed with 45 μl of reaction buffer and co-incubated with variant concentrations of ACE2-Fc. The fluorescence (Ex/Em = 320/420 nm) was measured in kinetic mode for 30 min to 2 h at room temperature by a fluorescence reader (Molecular Devices).

## ACE2-Fc and Spike binding by ELISA assay

Fifty microliters of 50 ng/ml purified 1–674 or 319–591 Spike proteins was precoated onto a 96-well ELISA plate at 4°C overnight. The plate was first washed three times with PBST (phosphate-buffered saline (PBS) containing 0.05% Tween-20) and blocked with blocking buffer (1% BSA, 0.05% $NaN_3$, and 5% sucrose in PBS) at room temperature for 30 min. After that, the plate was washed three times with PBST. Serially diluted ACE2-Fc-Biotin with or without soluble unlabeled ACE2-Fc or Spike 1–674 was preincubated at 37°C for 1 h. After that, the mixtures were added to the 96-well plate and incubated at 37°C for 1 h. After that, the plate was washed three times with PBST and incubated with horseradish peroxidase (HRP)-conjugated streptavidin (1:500) at 37°C for 30 min. After three washes with PBST, tetramethylbenzidine substrate (TMB) (T8665, Sigma) was added for 30 min before stopping the reaction with 50 μl of 1 N $H_2SO_4$. HRP activity was measured at 450 nm using an ELISA plate reader (VERSAMAX).

## Cell fusion and syncytia formation

HEK293T cells were cotransfected with plasmid 5 μg of pCR3.1-Spike and 0.5 μg of pLKO AS2-GFP by lipofectamine 3000 (L3000015, Thermo Fisher Scientific) for 3 days before being used as effector cells (293T-S). H1975 lung adenocarcinoma cells and HEK293T cells were transduced with lentivirus encoding full-length ACE2 before being used as target cells (H1975-ACE2). H1975, H1975-ACE2, HEK293, and HEK293T ($7.5 × 10^5$ cells/well) cells were seeded in the 24-well plate at 37°C overnight. The 293T-S cells were detached with 0.48 mM EDTA for 5 min. The 293T-S ($1 × 10^5$/reaction) cells were preincubated with normal human IgG or ACE2-Fc at 37°C for 1 h. After that, the antibody and effector cell mixtures were added to target cells and incubated at 37°C for 4 h or 24 h. Cells were fixed with 4% paraformaldehyde at room temperature for 30 min. The 293T/Spike/EGFP cells fused or unfused with HEK293T-ACE2 or H1975-ACE2 cells were counted under an inverted fluorescence microscope (Leica DMI 6000B fluorescence microscope). The percent inhibition of syncytia formation

was calculated using the following formula: $(100 - (H\text{-}L)/(E\text{-}L) \times 100)$. H represents the total green fluorescent score in the individual picture. L represents the green fluorescent score in the negative control group in which target cells were replaced by HEK293 or H1975). E represents the green fluorescent score in each picture in the IgG or ACE2-Fc groups. Each image of the green fluorescent score was determined by the MetaMorph's extensive analysis tools.

### Flow cytometry

ACE2-Fc was conjugated with green fluorescence using a FITC Labeling Kit (ab102884, Abcam). H1975-Spike-overexpressing cells ($2 \times 10^5$/reaction) were detached by 0.48 mM EDTA and then incubated with FITC-conjugated ACE2-FC or isotype control (Thermo Fisher Scientific) on ice for 1 h. After that, the cells were washed twice and resuspended in cold PBS. The fluorescence levels were quantified by the FACSCanto flow cytometer (Becton Dickinson) and analyzed using the FlowJo software.

### Immunofluorescence staining

The H1975-Spike-expressing cells were fixed with 4% paraformaldehyde and blocked with 10% FBS. The cells were then stained with anti-Spike antibody (1:1,000) at 4°C overnight and incubated with Alexa Fluor® 594 conjugated secondary antibody (1:500) at 37°C for 1 h. Next, the cells were stained with FITC-conjugated ACE2-Fc at 4°C overnight and mounted with ProLong™ Diamond Antifade Mountant with DAPI (Thermo Fisher Scientific). Then, images were taken with an LSM 700 laser scanning confocal microscope (Carl Zeiss).

### Measurement of TNF-α levels in culture media

RAW264.7 macrophage cells ($1 \times 10^5$ cells/well) were seeded in the 12-well plate overnight. Angiotensin II (A9525, Sigma-Aldrich) was preincubated with or without ACE2-Fc at 37°C for 30 min. Then, the mixtures were added to RAW264.7 cells for another 12 h. The TNF-α concentrations in the culture supernatants were measured using the ELISA kit (DY410, R&D Systems) according to the manufacturer's protocols. The absorbance at 450 nm in each well was determined using a VersaMax microplate reader (Molecular Devices).

### Plasma stability

ACE2-Fc (2 μg/ml) was prepared in 50% normal human serum (Sigma, H4522) and incubated for 0, 1, 2, and up to 10 days at 37°C and then stored at −20°C. The ACE2-Fc binding activity was determined by the ELISA assay as described above.

### Cell viability assay

The cell viability assay was determined according to the manufacturer's instructions (CellTiter 96® AQueous MTS, G1111, Promega). Briefly, $5 \times 10^3$ cells per well were seeded into 96-well plates in complete culture media. After 24 h, cells were treated with various concentrations of ACE2-Fc or IgG for another 72 h in complete culture media at 37°C. Twenty microliters of the MTS stock solution was added to each well of the treated cells. After another 1 h of incubation, absorption was measured at 490 nm by the spectrophotometer (Molecular Devices).

### Production and purification of pseudotyped SARS-CoV-2 lentivirus

The pseudotyped lentivirus carrying SARS-CoV-2 Spike protein was generated according to a published method with minor modifications (Glowacka et al, 2011). In brief, HEK293T cells were transiently transfected with pLAS2w.Fluc.puro, pcDNA3.1-2019-nCoV-S, and pCMV-ΔR8.91 by using TransITR-LT1 transfection reagent (Mirus). The culture medium was refreshed at 16 h and harvested at 48 and 72 h post-transfection. Cell debris was removed by centrifugation at 4,000 g for 10 min, and the supernatant was filtered through a 0.45-μm syringe filter (Pall Corporation). For pseudovirus purification and concentration, the supernatant was mixed with 0.2 × volume of 50% PEG 8,000 (Sigma) and incubated at 4°C for 2 h. The pseudotyped lentivirus was then recovered by centrifugation at 5,000 g for 2 h, resolved in sterilized phosphate-buffered saline, aliquoted, and stored at −80°C.

### Estimation of lentiviral titer by using the luciferase assay

The standard VSV-G pseudotyped lentivirus was generated by transient transfection of HEK293T cells with pLAS2w.Fluc. puro, pMD-G, and pCMV-ΔR8.91 as described above. The transduction unit of VSV-G-pseudotyped lentivirus was estimated using the cell viability assay according to the National RNAi Core Facility, Academia Sinica, Taipei, Taiwan. The VSV-G pseudotyped lentivirus with a known transduction unit was used to estimate the lentiviral titer of the pseudotyped lentivirus with SARS-CoV-2 Spike protein. In brief, HEK293T cells stably expressing human ACE2 were plated onto 96-well plates 1 day before lentivirus transduction. For the titration of pseudotyped lentivirus, different amounts of lentivirus were added into the culture medium containing polybrene (final concentration of 8 μg/ml). Spin infection was carried out at 1,100 g in a 96-well plate for 15 min at 37°C. After incubating cells at 37°C for 16 h, the culture medium containing virus and polybrene was removed and replaced with fresh DMEM containing 10% FBS. The expression level of luciferase was determined at 72 h postinfection by the Bright-Glo™ Luciferase Assay System (Promega). The relative light unit (RLU) of VSV-G pseudovirus-transduced cells was used as a standard to determine the virus titer.

### Pseudotype SARS-CoV-2 virus neutralization assay

ACE2-Fc was serially diluted twofold in culture medium starting at 100 μg/ml or 200 μg/ml. ACE2-Fc and pseudovirus were preincubated at 37°C for 1 h. After that, the decoy antibody and virus mixtures were added to ACE2-expressing HEK293T cells, and spin infection was performed at 1,100 g for 15 min at 37°C before incubation at 37°C for an additional 4 h. The cells were washed once with PBS, refreshed with culture medium, and incubated at 37°C in a humidified atmosphere containing 5% $CO_2$ and 20% $O_2$ for another 48 h. Luciferase activity was determined according to the manufacturer's instructions (E1501, Promega).

## Human airway organoid culture

The human airway organoid culture method was slightly modified from a previous report (Sachs *et al*, 2019). In brief, human bronchial/tracheal epithelial cells (502-05a, Cell APPLICATIONS) were suspended in 10 mg/ml cold Corning Matrigel Growth Factor Reduced (GFR) Basement Membrane Matrix (356230, CORNING), and 50 µl drops of the cell suspension were solidified on prewarmed 24-well culture plates at 37°C with 5% $CO_2$ for 10–20 min. Five hundred microliters of airway organoid medium (composition is shown in Reagent Table) was added to each well, and the medium was changed every 2–3 days. Airway organoids were passaged every 2 weeks. For passaging, airway organoids were first mechanically sheared through a flamed glass Pasteur pipette and further dissociated by incubation with TrypLE select enzyme (12563011, Thermo Fisher Scientific). The dissociated organoid fragments were collected by centrifugation at 400 *g* for 5 min and reseeded as above at the ratios of 1:2 to 1:4.

## Pseudotype SARS-CoV-2 infection of human airway organoids

The airway organoids (organoid size approximately 100 µm) were extracted from the Matrigel by mechanical shearing through a flamed glass Pasteur pipette, and approximately 100 organoids were infected with $1.0 \times 10^5$ PFU of pseudotype SARS-CoV-2 in a 24-well plate containing 500 µl airway organoid medium. The virus and organoid-containing plates were first centrifuged at 1,100 *g* for 30 min and placed at 37°C with 5% $CO_2$ for an additional 30 min. After incubation, airway organoids were collected by centrifugation at 400 *g* for 5 min and re-embedded in Matrigel with the same airway organoid medium containing pseudotype SARS-CoV-2. At 72 h, organoids were harvested, and the luciferase activities were measured according to the manufacturer's instructions (E1501, Promega).

## Treatment of human airway organoids with ACE2-Fc

Different concentrations of ACE2-Fc were mixed with $1.0 \times 10^5$ PFU of pseudotype SARS-CoV-2 for 60 min at 37°C in a final volume of 500 µl of airway organoid medium. The ACE2-Fc and virus mixtures were added to the Airway organoids following the procedures described in the Section on pseudotype SARS-CoV-2 infection. At 72 h, organoids were harvested, and the luciferase activities were measured according to the manufacturer's instructions (E1501, Promega).

## Histological analysis of human airway organoids

Airway organoids were first washed with PBS and fixed with 4% paraformaldehyde for 20 min at room temperature. Next, the fixed airway organoids were processed and embedded in paraffin. Then, the blocks were cut into 3-µm thick sections. For hematoxylin and eosin (H&E) staining, the sections were deparaffinized, rehydrated, stained with H&E, and examined using an Olympus BX51 Microscope with a DP73 Olympus Color camera. For immunofluorescence staining, the sections were deparaffinized, rehydrated, and subjected to antigen retrieval by treatment with 0.1% trypsin in PBS at 37°C for 30 min. Then, the sections were blocked with 5% bovine serum albumin in PBS at room temperature for 30 min. The sections were incubated with primary antibodies overnight at 4°C (anti-p63, ab124762, Abcam, 1:50; anti-SCGB1A1, sc-365992, Santa Cruz, 1:50; anti-acetylated a-tubulin, sc-23950, Santa Cruz, 1:50; anti-mucin 5AC, MS-145, Thermo Fisher Scientific, 1:50; anti-ACE2, ab108209, Abcam, 1:100; anti-TMPRSS2, sc-515727, Santa Cruz, 1:50), washed three times with PBS, incubated with secondary antibodies (Alexa Fluor 488 goat anti-rabbit, A11034, Thermo Fisher Scientific, 1:500; Alexa Fluor 488 goat anti-mouse, A11001, Thermo Fisher Scientific, 1:500) for 1 h at room temperature, washed three times with PBS, incubated with fluorophore-conjugated Phalloidin (Phalloidin-TRITC, P1951, Sigma-Aldrich, 1:200), washed three times with PBS, and mounted with prolonged Diamond antifade mountant with DAPI (P36962, Thermo Fisher Scientific). The sections were imaged on a Zeiss LSM 510 Meta Inverted confocal microscope and processed using Zeiss LSM Image Browser software.

## Plaque reduction assay

Vero E6 cells were seeded into 24-well culture plates in DMEM with 10% FBS and antibiotics 1 day before infection. SARS-CoV-2 was incubated with antibodies for 1 h at 37°C before adding to the cell monolayer for another hour. Subsequently, virus-antibody mixtures were removed, and the cell monolayer was washed once with PBS before covering with media containing 1% methylcellulose for 5–7 days. The cells were fixed with 10% formaldehyde overnight. After removal of the overlay media, the cells were stained with 0.7% crystal violet, and the plaques were counted. The percentage of inhibition was calculated as $[1 - (VD/VC)] \times 100\%$, where VD and VC refer to the virus titers in the presence and absence of the compound, respectively.

## Yield reduction assay

Vero E6 cells were seeded into 24-well culture plates in DMEM with 10% FBS and antibiotics 1 day before infection. SARS-CoV-2 (multiplicity of infection, MOI = 1) was incubated with antibodies for 1 h at 37°C before adding to the cell monolayer for another hour. After removal of the virus inoculum, the cells were washed once with PBS and overlaid with 0.5 ml medium for 24 h at 37°C. The next day, the culture supernatants were harvested for RNA extraction, and the cells were retrieved for protein, RNA extraction, and immunofluorescence assays. The number of viruses in the supernatants and infected cells was determined by qPCR using the protocol provided by the WHO (https://virologie-ccm.charite.de/en). Quantitative PCR of the E gene was performed using the iTaq™ Universal Probes One-Step RT-PCR Kit (172-5140, Bio-Rad, USA) and Applied Biosystems 7500 Real-Time PCR software (version 7500SDS v1.5.1). A plasmid containing a partial E fragment was used as the standard to calculate the amount of viral load in the specimens. The percentage inhibition of virus yield was calculated as $[1 - (Vd/Vc)] \times 100\%$, where Vd and Vc refer to the virus copies in the presence and absence of the test compound, respectively.

## ACE2-Fc mediated degranulation assay

Human primary NK cells were thawed from cryogenic tubes and cultured in the NK MACS medium (Miltenyi Biotec) as described by

### The paper explained

#### Problem

The COVID-19 pandemic has been causing devastating damage worldwide. To date, there are no effective strategies to combat SARS-CoV-2 infection. Here, we report the development of a humanized decoy antibody that can block viral entry and prevent SARS-CoV-2 infection.

#### Results

We design a humanized decoy antibody (ACE2-FC fusion protein) that specifically binds to the SARS-CoV-2 Spike protein and blocks entry of six clinical isolates including the D614G variant strains with high infectivity, thus inhibiting SARS-CoV-2 infection of host cells. The preservation of the peptidase activity in the ACE2-Fc fusion protein was shown to reduce the angiotensin II-mediated cytokine cascade. This decoy antibody could also activate the degranulation of NK cells.

#### Impact

The humanized decoy antibody may have great potential to develop as an effective therapeutic agent to prevent SARS-CoV-2 infection and suppress the subsequent inflammatory cascade.

the manufacturer's protocol. Forty-eight hours before the assay, NK cells were activated by 1,000 U/ml recombinant human IL-2 (Peprotech). To initiate cytotoxicity, 50,000 NK cells were incubated with H1975-Spike cells at a 1:1 cell ratio in a U-bottom 96-well plate in the presence of ACE2-Fc, ACE2, or control (Dulbecco's Phosphate-Buffered Saline (DPBS, Corning®)). Two microliters of the anti-human CD107a antibody (BioLegend, H4A3) was mixed into each well. The plate was then centrifuged at 200 *g* for 5 min to facilitate the contact of NK cells and H1975-Spike cells. After 1 h of incubation at 37°C, Brefeldin A and Monensin mixtures (BioLegend) were added into each well, and the plate was incubated at 37°C for another 3 h. The cells were then fixed and permeabilized with Cyto-Fast Fix-Perm Buffer (BioLegend) and stained with anti-human IFNγ (BioLegend, B27) and anti-human TNF-α (BioLegend, MAb11) according to the manufacturer's protocol. The stained cells were resuspended in flow cytometry buffer and analyzed by the Cyto-FLEX flow cytometer (Beckman Coulter).

### SARS-CoV-2 isolation

Sputum or throat swab specimens obtained from SARS-CoV-2-infected patients were maintained in the viral-transport medium. The specimens were propagated in VeroE6 cells in DMEM supplemented with 2 μg/ml tosylsulfonyl phenylalantyl chloromethyl ketone (TPCK)-trypsin (Sigma-Aldrich). Culture supernatants were harvested when more than 70% of cells showed cytopathic effects. The full-length genomic sequences of the derived clinical isolates were determined and submitted, along with the patients' travel history and basic information, to the GISAID database. The virus strains used in this study include SARS-CoV-2/NTU03/TWN/human/2020 (Accession ID EPI_ISL_413592), SARS-CoV-2/NTU13/TWN/human/2020 (Accession ID EPI_ISL_422415), SARS-CoV-2/NTU14/TWN/human/2020 (Accession ID EPI_ISL_422416), SARS-CoV-2/NTU18/TWN/human/2020 (Accession ID EPI_ISL_447615), SARS-CoV-2/NTU25/TWN/human/2020 (Accession ID EPI_ISL_447619) and SARS-CoV-2/NTU27/TWN/human/2020 (Accession ID EPI_ISL_447621). The virus titers were determined by plaque assay

(Su *et al*, 2008) for subsequent analysis. The study was approved by the NTUH Research Ethics Committee (202002002RIND), and the participants gave written informed consent.

### Statistical analysis

Every figure legend contains information on the sample size, the statistical test used, and the *P* value. Data are expressed as the mean ± standard deviation (SD). An unpaired two-tailed Student's *t*-test was used for the comparison of continuous variables, with correction of unequal variances when appropriate. A $P < 0.05$ was considered statistically significant. $*P < 0.05$, $**P < 0.01$, $***P < 0.001$. Exact *P*-values are indicated for main and EV figures in Appendix Table S1.

## Data availability

This study includes no data deposited in external repositories.

**Expanded View** for this article is available online.

## Acknowledgements

This research was supported by Academia Sinica [AS-SUMMIT-108; AS-SUMMIT-109; AS-KPQ-109-BioMed] and the Ministry of Science and Technology [MOST 108-3114-Y-001-002, MOST 109-2314-B-002-254, MOST 109-2314-B-002-277 and MOST 109-2124-M-002-012-, MOST 107-2320-B-002-016-MY3, MOST 109-2327-B-002-009]. The study was approved by the NTUH Research Ethics Committee (202002002RIND by Dr. Jann-Tay Wang), and the participants gave written informed consent.

## Author contributions

K-YH, S-YC, and P-CY designed the research. K-YH and C-LC constructed the plasmids and produced and purified the soluble protein. K-YH, C-LC, M-SL, and C-CL performed the ELISA assay and Western blot. Y-CC produced and concentrated the pseudovirus. T-LC, Y-HP, H-CK, and S-YC isolated and performed SARS-CoV-2 relative experiments. T-CK created the lung organoid system. K-YH and M-SL performed cell–cell fusion assays and Ang II-induced inflammation experiments. R-SH and SL performed the degranulation assay. K-YH, S-YC, and P-CY analyzed the data and wrote the manuscript.

## Conflict of interest

The authors declare that they have no conflict of interest.

## For more information

www.ibms.sinica.edu.tw/pan-chyr-yang/

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
