## [Review Process File · EMBO Molecular Medicine]

Humanized COVID-19 decoy antibody effectively blocks viral entry and prevents SARS-CoV-2 infection

Kuo-Yen Huang, Ming-Shiu Lin, Ting-Chun Kuo, Ci-Ling Chen, Chung-Chih Lin, Yu-Chi Chou, Tai-Ling Chao, Yu-Hao Pang, Han-Chieh Kao, Rih-Sheng Huang, Steven Lin, Sui-Yuan Chang, and Pan-Chyr Yang

DOI: [10.15252/emmm.202012828](https://doi.org/10.15252/emmm.202012828)

Corresponding authors: Pan-Chyr Yang (pcyang@ntu.edu.tw), Sui-Yuan Chang (sychang@ntu.edu.tw)

Review Timeline:

Submission Date:	29th May 20
Editorial Decision:	16th Jun 20
Revision Received:	29th Sep 20
Editorial Decision:	22nd Oct 20
Revision Received:	3rd Nov 20
Accepted:	5th Nov 20

Editor: Celine Carret / Zeljko Durdevic

Transaction Report:

16th Jun 2020

Dear Prof. Yang,

Thank you for the submission of your manuscript to EMBO Molecular Medicine. We have now heard back from the two referees whom we asked to evaluate your manuscript. As you will see from the reports below, the referees find the topic of your study of potential interest although the novelty, at this stage is rather limited by existing published work. In order to strengthen the data and increase the novelty (indispensable for publication in EMBO Molecular Medicine), we would like to insist on the following items, which after cross-commenting from the referees, were unanimously agreed upon:

- incorporate ACE2-Fc neutralization of virus-cell and possibly cell-cell fusion in lung organoids,
- show a benefit of ACE2-Fc compared to the recombinant ACE2 protein,
- provide more details on the methodology,
- discuss potential limitations of using a therapeutic ACE2-Fc antibody on the RAS system
- demonstrating the effect of the ACE2-Fc on different clinical isolates

We therefore would be willing to consider a revised manuscript with the understanding that the referees' concerns itemised above must be fully addressed and that acceptance of the manuscript would entail a second round of review.

I should remind you that it is EMBO Molecular Medicine policy to allow a single round of revision only and that, therefore, acceptance or rejection of the manuscript will depend on the completeness of your responses included in the next, final version of the manuscript. I realize that addressing the referees' comments in full would involve a lot of additional experimental work and I am uncertain whether you will be able (or willing) to return a revised manuscript within 3 months and I would also understand your decision if you choose to rather seek rapid publication elsewhere at this stage.

I look forward to seeing a revised form of your manuscript as soon as possible.

Should you find that the requested revisions are not feasible within the constraints outlined here and choose, therefore, to submit your paper elsewhere, we would welcome a message to this effect.

Yours sincerely,

Celine Carret

***** Reviewer's comments *****

Referee #1 (Comments on Novelty/Model System for Author):

SARS-Coronavirus-2 (SARS-CoV-2) causes pandemic coronavirus disease-19 (COVID-19) with nearly 6 million people infected and there is currently no effective treatment. Similar to the disease caused by SARS-CoV and MERS-CoV, SARS-CoV-2 its major phenotype is severe acute respiratory distress syndrome (de Wit et al 2016; Huang et al 2020). SARS-CoV-2 uses Angiotensin Converting Enzyme II (ACE2) bind directly to SARS-CoV-2 Spike protein (Walls et al 2020;Wrapp et al 2020;Wan et al 2020) and is used as cell entry receptor (Zhou et al 2020;Hoffmann et al 2020).

ACE2 peptidase activity negatively regulates Angiotensin II (AngII). Treatment with recombinant ACE2 protein prevent Ang II-induced hypertension and cardiac dysfunction (Minato et al 2020), might control AngII-AT1R proinflammatory cytokine responses (Eguchi et al 2018), and protects from severe acute lung failure (Imai et al 2005). As impaired ACE2 expression was observed in mice receiving SARS-CoV Spike protein injection (Kuba et al, 2005), it is postulated that SARS-CoV-2 Spike proteins hijack ACE2 function driving COVID19-associated ARDS (Hirano et al 2020). Monteil et al 2020 demonstrated that administration of clinical grade human recombinant ACE2 protein significantly block early stages of SARS-CoV-2 infections in vero E6 cells. This protein has a low half-life, thereby limiting its therapeutical potential.

With this in mind, the authors generated recombinant ACE2 fused to Fc that is known to increase its half-life (Liu et al 2018). Using ELISA they demonstrate that recombinant ACE2-Fc binds recombinant SARS-CoV-2 Spike Fc protein. They show that pre-incubation of Spike protein pseudotyped lentivirus or clinically isolated SARS-CoV-2 with ACE2-Fc reduces infection of ACE2 overexpressing 293T cell line and Vero E6 cells, respectively. Interestingly, their ACE2-Fc has peptidase activity as demonstrated by cleaving fluorescent substrate.

The results are clear and I didn't find any obvious problems with the design and execution of the experiments. However, the concept of using recombinant ACE2 (Monteil et al 2020) and fusing it to Fc has previously been established (Lei et al 2020). Importantly, these authors' ACE2-Fc seem to be less efficient in virus neutralization with 25ug/ml resulting in 60% neutralization versus 0.1ug/ml described in Lei et al 2020. Although a direct comparison should not be made as different pseudotyped viruses, cells, and MOIs were used, there is no clear benefit of these authors ACE2-Fc. The idea that enzymatically active ACE2-Fc protects against lung injury in COVID19 is very interesting, but has not been demonstrated here or by others. Caution should be taken as administration of recombinant ACE2 is also able to inhibit myocardial remodelling, attenuate Ang II-induced cardiac hypertrophy, and cardiac dysfunction (Huentelman et al 2005).

In conclusion, this manuscript lacks novelty and there is no clear benefit above the previously reported ACE2-fc. Although the ACE2-Fc recombinant proteins seem promising in combating COVID19 and further analyses in their ability to neutralize SARS-CoV-2 infection in vitro and in animals should be stimulated, this is not enough to warrant publication.

Referee #1 (Remarks for Author):

Major comments:

- 1) Spike proteins form trimers in the viral and host cell membrane, but here the authors connect Spike to Fc forming presumably dimers. To show binding of ACE2-Fc to trimeric Spike protein demonstrate that ACE2-Fc stains only cells infected with Spike protein pseudotyped lentivirus and not cells infected by VSVG-lentivirus.
- 2) ACE2-Fc interfering with Spike-ACE2 interaction is dose-dependent. Therefore, neutralization of Spike protein pseudotyped viral infection should be examined in at least one more experiment with a higher viral dose.
- 3) It is crucial to extend the neutralizing activity of ACE2-Fc to lung cell infection with clinical isolated SARS-CoV-2.
- 4) A slightly larger recombinant ACE2-Fc (Lei et al 2020) has been demonstrated to neutralize SARS-CoV-2 Spike protein pseudotyped viral infection more efficient than these authors' ACE2-Fc. Are there any benefits of your ACE2-Fc in comparison to the previously published ACE2-Fc? For example, does NFkB activity or signs of injury reduce upon administration of enzymatically active ACE2-Fc during SARS-CoV-2 infection of lung cells (ideally lung organoids).
- 5) It should be determined whether passaging of SARS-CoV-2 in the presence of ACE2-Fc decreases viral infection of neighbouring cells and syncytia formation. Ideally, resistance development should be tested.

Minor comments:

- 1) Could the authors indicate what statistical method was used and what the "*" indicates in the figure legends.
- 2) Could the authors explain how IgG condition in Fig 2B could have a negative value?
- 3) Fig2D should have a logarithmic y-axis with fold change in comparison to the IgG treated condition as viral replication happens exponentially.
- 4) The authors themselves already indicate the importance of Spike protein glycosylation for Spike-ACE2 interaction. Could they show by PNGF-treatment that their engineered Spike proteins are indeed glycosylated.

Typo's:

- 1) SARA instead of SARS in start of line 16 in paragraph "ACE2-Fc blocks SARS-CoV-2 entry and replication".
- 2) Capital "I" in 12th line of methods paragraph "estimation of lentiviral titer by using luciferase assay".
- 3) Extra space in 4th line of methods paragraph "Plague reduction assay"

Referee #2 (Comments on Novelty/Model System for Author):

This study investigated the suitability of humanised ACE2-Fc fusion as decoy antibodies for preventing the entry of SARS-COV-2 into human cells invitro. The ACE2-Fc fusion protein can form dimer mimicking antibodies and specifically binds to SARS-CoV-2 Spike protein. By competitively binding to the viral Spike protein, the decoy antibodies prevent viral binding to the natural, membrane-bound ACE2, and thus blocks virus entry into host cells. The study utilised virus from clinical isolates and demonstrate that the ACE2-Fc fusion protein abrogated viral replication in Vero E6 cells. This study supports the recently published work by Monteil, and colleagues (Reference 9) who showed that recombinant purified human ACE2 (hrsACE2) - a decoy protein that has already been tested in phase I and II clinical trials can inhibit the interaction between SARS-CoV-2 and ACE2. Treatment of cells with hrsACE2 inhibited SARS-CoV-2 infection in a dose-dependent manner and attenuated propagation of the virus. Using two human organoid models the Monteil et al., showed that hrsACE2 indeed inhibited the virus from infecting the host cells.

This paper in review presents an improvement to Monteil et al.'s work by generating this antibody decoy. The ACE2-Fc fusion which was previously shown to significantly increase the half-life of the ACE2 protein (Liu et al. (12), which may likely be the case by extension. They tested this decoy on clinical isolates, although very little detail has been provided about the number of isolates that were tested. The Monteil paper tested clinical isolates from a single patient, so without clarifying how many isolates it is not clear how much extra knowledge can be learnt from this study from the point of view of heterogeneity of the isolates. There seem to be no direct evidence from this study of the benefit of adding the Fc domain to the ACE2. This is a main limitation of this work as the benefits of making this Fc-ACE2 fusion are based on previous publications with no effort to demonstrate directly the value of making this modification. This is very important as this new ACE2-Fc fusion will require rigorous testing and clinical trials before it can be used in therapy.

1. The author suggest that the fusion could trigger antibody -dependent cellular cytotoxicity and complement dependent cytotoxicity. However, this work does not address this added benefit of fusing this ACE-2 protein to make an antibody decoy. Based on published literature the authors also suggest that the peptidase activity may enable ACE2-Fc to reduce angiotensin II mediated cytokine cascade during SARS-CoV-2 infection (Hirano & Murakami, 2020). However, this work does not address any of these supposed added benefits of fusing the Fc region or the peptidase activity. Also, the discussion does not clearly articulate these benefits. It will be good to see at least one experiment which demonstrate the increased half-life and its benefit, as well as lack of toxicity of this decoy to the cells in particular due to its supposed longer half-life.
2. The design of the study focusses on the early stages of infection, as they preincubated the virus with the decoy fusion protein before adding to Vero-6 cells or to HEK293 cells This study cannot be used to predict the effect of the ACE2-Fc fusion in later stages of the disease process. Could the authors add the decoy to cells that are infected already and see how this decoy altered the course of infection when it has already established?
3. There was no attempt to study the effect in a disease model such as in lung organoids or in mice or at the very least discussing future work along these lines.
4. The RAS system represents a complex network of pathways that are influenced by many factors, hence it is not clear how the ACE2-Fc will impact on this system. All limitations listed above have not been discussed in this study.
5. The discussion should clarify how the authors see this decoy being employed as a therapeutic strategy. Such as to deliver the decoy product by inhaler given to healthcare workers or those people at risk as for preventing infection. This might be met with better outcomes as disease might be milder as this could be a 'protective shield' against Covid-19 by inhibiting and neutralising the virus' activity in naïve at-risk populations.
6. "The viral entry blocking effect of ACE2-Fc was further confirmed using real SARA-CoV-2 isolated from patients suffering from COVID-19 infection in National Taiwan University Hospital".

There are no further details in the methods regarding how the isolates were obtained including clarity in the methods as to how many isolates were tested. Were the isolates sequenced or identified by some method? Were they the same or different variants of SARS-COV-2?
7. Ethics relating to the collection of isolates have not been clarified if this is a consideration.
8. Statistical analysis should clarify number of repeats

Referee #2 (Remarks for Author):

The author suggest that the fusion could trigger antibody -dependent cellular cytotoxicity and complement dependent cytotoxicity. However, this work does not address this added benefit of fusing this ACE-2 protein to make an antibody decoy. Based on published literature the authors also suggest that the peptidase activity may enable ACE2-Fc to reduce angiotensin II mediated cytokine cascade during SARS-CoV-2 infection (Hirano & Murakami, 2020). However, this work does not address any of these supposed added benefits of fusing the Fc region or the peptidase activity. Also, the discussion does not clearly articulate these benefits. It will be good to see at least one experiment which demonstrate the increased half-life and its benefit, as well as lack of toxicity of this decoy to the cells in particular due to its supposed longer half-life.

2. The design of the study focusses on the early stages of infection, as they preincubated the virus with the decoy fusion protein before adding to Vero-6 cells or to HEK293 cells This study cannot be used to predict the effect of the ACE2-Fc fusion in later stages of the disease process. Could the authors add the decoy to cells that are infected already and see how this decoy altered the course of infection when it has already established?

3. There was no attempt to study the effect in a disease model such as in lung organoids or in mice or at the very least discussing future work along these lines.

4. The RAS system represents a complex network of pathways that are influenced by many factors hence it is not clear how the ACE2-Fc will impact on this system. All limitations listed above have not been discussed in this study.

5. The discussion should clarify how the authors see this decoy being employed as a therapeutic strategy. Such as to deliver the decoy product by inhaler given to healthcare workers or those people at risk as for preventing infection. This might be met with better outcomes as disease might be milder as this could be a 'protective shield' against Covid-19 by inhibiting and neutralising the virus' activity in naïve at-risk populations.

6. "The viral entry blocking effect of ACE2-Fc was further confirmed using real SARS-CoV-2 isolated from patients suffering from COVID-19 infection in National Taiwan University Hospital". There are no further details in the methods regarding how the isolates were obtained including clarity in the methods as to how many isolates were tested. Were the isolates sequenced or identified by some method? Were they the same or different variants of SARS-COV-2?

7. Ethics relating to the collection of isolates have not been clarified if this is a consideration.

8. Statistical analysis should clarify number of repeats

Spellings and to improve clarity I suggest rephrasing

In introduction

Base- Based

Whether ACE2-Fc fusion still owning the peptidase activity of ACE2 could block SARS-COV-2 entry and endow the ability of antibodies including ADCC and CDC remains to be investigated

Results

For clarity for non-specialist

Why was IL-2 signalling peptide included at the C-terminus of ACE2?

SARA-COV-2 should read SARS-COV-2

In discussion

Our study results demonstrated that even when we shortened the ACE protein to 597 amino acids...

The chimeric ACE2-Fc antibody had a much longer elimination phase or much longer half- life (not both in same sentence). I couldn't see this data; my assumption is that it is based on the cited publication (12) which clearly shows this effect.

Fragment need clarity...In addition to identify infected cells and provide the opportunity for the immune system to clean injured cells

Methods

At 1:1000 to 1:10000 dilution. This is not a sentence, needs revision

Eaction buffer should it read - reaction buffer?

The transduction unit of VSV-G-peudotyped lentivirus.... Typo

Figure 2 legend GAPDH is served as a control- fragment needs revision

Dear Editor:

We would like to thank the Editors/Reviewers for your time and kind consideration of our manuscript. We found that the comments from the Editor and the reviewers are very constructive. We have followed the comments from the expert reviewers, performed additional experiments, and carefully addressed all the critiques in the revised manuscript.

The associate editor suggests “In order to strengthen the data and increase the novelty (indispensable for publication in *EMBO Molecular Medicine*), we would like to insist on the following items, which after cross-commenting from the referees, were unanimously agreed upon”. These 5 specific items suggested were all carefully addressed and answered as follows:

Q1.- incorporate ACE2-Fc neutralization of virus-cell and possibly cell-cell fusion in lung organoids,

Response:

We thank the Editor for the suggestion. Please refer to our reply to Q3 of Reviewer 2 in which we have also addressed this point. In summary, we have performed the cell-cell fusion and the syncytia formation assay to investigate the neutralizing activities of ACE2-Fc. Our results indicated that ACE2-Fc could suppress cell-cell fusion and syncytia formation in two cell lines (Fig 3). In addition, we also established an airway organoid model, which is susceptible to SRS-CoV-2 infection, and showed that ACE2-Fc could block Spike-expressing pseudotyped virus entry into the airway organoids (Fig 4B-4D). The additional results have been incorporated in the revised manuscript as following:

(Page 5, line 21)

ACE2-Fc inhibits SARS-CoV-2 Spike-mediated cell-cell fusion and syncytia formation

To determine whether the decoy antibody is able to inhibit SARS-CoV-2 fusion with the target cells, we cotransfected SARS-CoV-2 Spike protein and EGFP into the HEK293T cells as the effector cells (293T-S) and used the ACE2-stable-expressing HEK293T and H1975 cells as the target cells (293T-ACE2 and H1975-ACE2) (Fig 3B). The target cells without ACE2 overexpression were used as controls. The effector cells (293T-S) were preincubated with ACE2-Fc or IgG at 37°C for one hour before mixing with the target cells or control cells for another 4 hours (cell-cell fusion assay) or 24 hours (syncytia formation assay) (Fig 3A). The area of EGFP was counted, and the inhibitory effect of ACE2-Fc on SARS-CoV-2 Spike-mediated cell-cell fusion and syncytia formation was quantified (n=6). As shown in Fig 3C and 3D, ACE2-Fc significantly impaired SARS-CoV-2 Spike-mediated cell-cell fusion and syncytia

formation compared to the normal human IgG control in both the HEK293T and the H1975 cell systems. These results demonstrated that ACE2-Fc could block SARS-CoV-2 infection via abrogating virus-mediated cell-cell fusion and syncytium formation.

(Page 6, line 25)

Since the lung is the primary site for SARS-CoV-2 infection, we established an airway organoid model following the methods described in a recent article (Sachs *et al*, 2019) to investigate the neutralization ability of ACE2-Fc against SARS-CoV-2 entry. The derived airway differentiation organoids were successfully established and composed of several airway epithelial cells with specific markers, including basal (P63), secretory (club cell marker secretoglobin family 1A member 1 (SCGB1A1) and secretory cell marker mucin 5AC (MUC5AC)), and multiciliated cells (cilia marker acetylated α -tubulin) (Fig 4B). Of note, unlike the A549 or the parental human normal bronchial epithelial cells (HBEpc), these airway organoids expressed a high-level of ACE2 in addition to TMPRSS2 (Fig 4B and 4C). We then examined the neutralization ability of ACE2-Fc for Spike-expressing pseudotyped virus in the airway organoid model. As shown in Fig 4D, the airway organoids are susceptible to virus entry and the ACE2-Fc could significantly block virus entry at the concentration of 100 μ g/mL. Our findings indicated that ACE2-Fc can inhibit entry of SARS-CoV-2 Spike-expressing pseudotyped virus entry into ACE2-expressing cells, including the airway organoids.

Figure 3. Inhibition of Spike-induced cell-cell fusion and syncytia formation by ACE2-Fc.

(A) The schematic diagram for cell-cell fusion and syncytia formation. (B) HEK293 and H1975 cells were transduced with full-length ACE2 by lentivirus. Protein extracts were immunoblotted with the indicated antibodies. O/E represents overexpression. (C) GFP and full-length Spike co-transfected HEK293T cells were used as effector cells (293-S). HEK293/ACE2 (HEK293T-overexpressing ACE2) (left panel) or H1975/ACE2 (H1975-overexpressing ACE2) (right panel) cells were used as target cells. The 293-S cells were preincubated with normal human IgG or ACE2-Fc at 37°C for 1 hr. After that, the mixtures were added to target cells for 4 hr (cell-cell fusion) or 24 hr (syncytia formation). The fluorescent areas were measured by inverted fluorescence microscopy and Metamorph software (Metamorphosis). Scale bars equal to 50 μ m. (D) The inhibition of cell-cell fusion and syncytia formation by ACE2-Fc in HEK293/ACE2 and H1975/ACE2 was determined with the formula described in the Methods section. Error bars represent the mean \pm SD, n=6. ** $P < 0.01$, *** $P < 0.001$. Experiments were performed at least three times with similar results.

B.

Figure 4. Blockage of Spike-expressing pseudovirus entry into ACE2-expressing cells by ACE2-Fc. (B) Confocal microscopy images of the airway organoids. The green fluorescence represents the specific staining of indicated monoclonal antibodies and Alexa Fluor 488-conjugated secondary antibodies. DAPI was used as the nuclear counterstain. Hematoxylin and eosin (H&E) stain: hematoxylin stained the nuclei in blue color; eosin stained the cytoplasm in pink color. Scale bars equal to 20 μ m. (C) Lung cancer A549 cells, human normal bronchial epithelial cells (HBEpc), and HBEpc-differentiated cells (airway organoids) were harvested for immunoblotting with the indicated antibodies. (D) Blockage of pseudovirus entry into airway organoids by ACE2-Fc. Mixtures of pseudotyped lentivirus with or without ACE2-Fc were cocultured with airway organoids for 72 hr. The virus entry was determined by quantifying the luciferase activity in the cell lysates. R.I.U: relative infection unit. Error bars represent the mean \pm SD, n=3. ** $P < 0.01$, *** $P < 0.001$. Experiments were performed at least three times with similar results.

Q2. - show a benefit of ACE2-Fc compared to the recombinant ACE2 protein,

Response:

We thank the Editor for the comment. Please refer to the reply to Q4 of Reviewer 1 in which we have also addressed this point.

The benefits of ACE2-Fc compared to the recombinant ACE2 protein have been summarized as follows. First, proteins fused to the Fc domain enable these molecules to interact with Fc receptors, which are critical for the induction of immune responses (Czajkowsky *et al*, 2012). After the specific interactions between antibodies or Fc fusion proteins with specific antigens on the cell surface of infected cells, cross-linking of the Fc domain could activate the CD16 (FcyRIII) receptor on the natural killer (NK) cells to trigger degranulation (CD107a on the cell membrane) and cytokine production (IFN- γ and TNF- α), which is essential to remove infected host cells (McDonald *et al*, 2015; Sun *et al*, 2019). Second, the antibody Fc domain, in addition to its ability to trigger antibody-dependent cellular cytotoxicity (ADCC) and

complement-dependent cytotoxicity (CDC), is known to endow the fusion protein with a longer half-life according to previous publications (Czajkowsky *et al.*, 2012). Third, based on previous findings in mouse ACE2 (Wysocki *et al.*, 2019), two shorten mouse ACE2 variants (1-605 and 1-619 A.A.) have been shown to exhibit higher ACE2 enzyme activity than that of the full-length ACE2 (1-740 A.A.). In addition, ACE2 enzyme activity could be detected in urine from ACE2 knockout mice after intravenous infusion of the mouse variant (1-619 A.A.), but not the full-length ACE2 (1-740 A.A.). Infusion of this short variant (ACE2 1-619 A.A.) also recovered the ACE2 activity of the kidney in the ACE2-deficient mice. These evidences supported that the shortened version of the ACE2-Fc (1-619 A.A.) is more stable in plasma and retains higher enzyme activity to convert Ang II to Ang 1-7 than the full-length ACE2. Based on the above reasons, an ACE2-Fc fusion protein with shortened ACE2 fragment (18-615 A.A.) were generated. Our study results first demonstrated that the decoy antibody ACE2-Fc exhibit potent neutralization activity against SARS-CoV-2 entry into cell lines as well as the airway organoids. Second, the decoy antibody ACE2-Fc is very stable since nearly 100 % of the spike-binding ability of ACE2-Fc was retained after being incubated with 50 % normal human serum for up to 10 days (Fig EV3C). Third, the ACE2-Fc decoy antibody preserves the peptidase activity of ACE2-Fc to reduce the Ang II-mediated cytokine cascade. The purified ACE2-Fc retained peptidase activity as compared to the human normal IgG and buffer controls (Fig 2D). After coincubation of Ang II with ACE2-Fc, we observed that ACE2-Fc could significantly suppress Ang II-induced TNF- α production (Fig 2E) and phosphorylation of ADAM17 (a disintegrin and metalloprotease 17) (Fig 2F). Finally, ACE2-Fc was shown to induce NK cell activation and antibody-dependent cellular cytotoxicity (ADCC), which may help to remove the infected cells *in vivo* (Fig 7). The additional results have been incorporated into the revised manuscript as following:

(Page 6, line 5)

The stability of ACE2-Fc in serum was subsequently determined. We incubated 2 μ g/mL ACE2-Fc in 50% normal human serum at 37°C for 0, 1, 2, and up to 10 days. The stability of ACE2-Fc was determined by assaying its binding ability to the Spike proteins in the ELISA assay. As shown in Fig EV3C, not a significant reduction of ACE2-Fc/Spike binding was observed up to ten days. These results suggest that ACE2-Fc has no toxicity to epithelial cells and may be stable in serum for ten days, which may facilitate its future clinical application.

(Page 7, line 28)

ACE2-Fc induced degranulation of natural killer (NK) cells.

Proteins fused to the Fc domain enable these molecules to interact with Fc receptors, which are

critical for the induction of immune responses (Czajkowsky *et al.*, 2012). Among these, antibody-dependent cellular cytotoxicity (ADCC) is an adaptive immune response, mainly mediated by the natural killer (NK) cells. After the specific interactions between antibodies or Fc fusion proteins with specific antigens on the cell surface of infected cells, cross-linking of the Fc domain will activate the CD16 (Fc γ RIII) receptor to trigger degranulation (CD107a on the cell membrane) and cytokine production (IFN- γ and TNF- α) of NK cells (McDonald *et al.*, 2015; Sun *et al.*, 2019). A recent report showed that the RBD-specific antibodies from an individual infected with SARS-CoV in 2003 could induce nearly 10% ADCC against SARS-CoV-2 (Pinto *et al.*, 2020). Therefore, experiments were conducted to examine whether ACE2-Fc could induce primary human NK cell activation. H1975 cells, transduced with full-length Spike by the lentiviral vector (H1975-Spike), were used as target cells (Fig 7A). The NK cell degranulation assay was performed to determine the CD107a, IFN- γ , and TNF- α expression levels after the co-incubation of the NK cells with H1975-Spike cells in the presence of ACE2-Fc or recombinant ACE2 (1-740 A.A. without an Fc tag). Induction of the expression levels of these three degranulation markers was observed when serially diluted ACE2-Fc was added to the co-culture of NK and H1975-Spike cells (Fig 7B-7D). In contrast, the degranulation of NK cells was not observed in the presence of recombinant ACE2. Taken together, our results suggest that ACE2-Fc could not only block SARS-CoV-2 infection but also induce NK cell activation, which may help to remove the infected cells *in vivo*.

Figure EV3C

Figure EV3 *in vitro* cytotoxicity and plasma stability and of ACE2-Fc. (C) *In vitro* serum stability of ACE2-Fc. ACE2-Fc was incubated with 50% normal human serum at 37°C for up to ten days. At the indicated time points, samples were collected to quantify the binding ability of ACE2-Fc to Spike proteins by ELISA. Error bars represent the mean \pm SD; n=3. Experiments were performed at least three times with similar results.

Figure 2. Functional characterization of the decoy antibody. (D) Preservation of ACE2-Fc peptidase activity. The peptidase activity of ACE2-Fc was measured by cleavage of the fluorescent peptide substrates. (E) Inhibition of angiotensin II (Ang II)-induced TNF- α production by ACE2-Fc. Ang II was preincubated with indicated amounts of ACE2-Fc or IgG for 30 minutes. After that, the mixtures were added to RAW264.7 cells for 12 hr. The concentration of TNF- α in the culture medium was determined by ELISA. (F) Inhibition of Ang II-induced ADAM17 (a disintegrin and metalloprotease 17) phosphorylation by ACE2-Fc. The protein extracts from ϵ were immunoblotted with the indicated antibodies (left panel). β -actin served as the loading control. ADAM17 represented. The signal intensity was normalized to cells only (right panel). Data are representative of three independent experiments, and the values are expressed as the mean \pm SD (lower panel). Error bars represent the mean \pm SD, $n=3$. * $P < 0.05$. IgG represents the human normal IgG control. Experiments were performed at least three times with similar results.

Figure 7. Effects of ACE2-Fc on NK cell degranulation. IL-2 activated NK cells were incubated with H1975-Spike cells at a 1:1 cell ratio in the presence of ACE2-Fc or ACE2. The activation of NK cells were determined by the CD107a, IFN- γ , and TNF- α expression levels. (A) H1975 cells were transduced with full-length Spike by Lentiviral vector. Protein extracts were immunoblotted with the indicated antibodies. O/E represents overexpress. (B) The effect of ACE2-Fc activation on degranulative capacity of NK cells. The experiments were performed with the primary human NK cells derived from three independent donors. Error bars represent mean \pm SD, n= 3. * P<0.05, *** P< 0.001. (C) The CD107a, IFN- γ , and TNF- α expression levels were determined by flow cytometry after the NK cells and H1975-Spike cells co-incubation in the

presence or absence of ACE2-Fc or ACE2 control. Error bars represent the mean \pm SD. The percentage of positive cells in both groups were normalized to the NK/H1975-Spike only group. (D) The differential expression of degranulation markers after the treatment of ACE2-Fc or ACE2 by the flow cytometry. The numbers in each plot indicate the percentages of positive cells.

Q3. - provide more details on the methodology,

Response:

We thank the Editor for the comment. We have added the detailed information in the **Material and Method** section in the revised manuscript. (Page 11, line 1 to Page 18, line 32)

Q4. - discuss potential limitations of using a therapeutic ACE2-Fc antibody on the RAS system

Response:

We thank the Editor for raising this important issue. Please refer to the response to Q4 of Reviewer 2. We have added the potential limitations of using a therapeutic ACE2-Fc antibody on the RAS system in the Discussion section of the revised manuscript.

(Page 9, line 10)

The renin-angiotensin system (RAS) is a hormone system that regulates vascular function, including the regulation of blood pressure, natriuresis, and blood volume control (Tikellis & Thomas, 2012). As a key regulator in the RAS, ACE2 acts by converting Ang II into Ang 1–7 (Gheblawi *et al*, 2020). Ang II can also downregulate ACE2 expression through AT1R when the RAS is overactivated in cardiovascular disease through the ERK1/2 and p38 MAPK signaling pathways (Koka *et al*, 2008). Three strategies have been evolved to reduce Ang II levels upon RAS overactivation, including ACE inhibitors (ACEis), angiotensin receptor blockers (ARBs), and mineralocorticoid receptor blockers (Gheblawi *et al.*, 2020). Most of these compounds increase the protein and mRNA levels of ACE2 or peptidase activity in animal models (Kai & Kai, 2020). Being known as the receptor for SARS-CoV-2, we speculate that ARBs- or ACEis-induced ACE2 upregulation may facilitate SARS-CoV-2 infection. In addition, the administration of recombinant ACE2 or ACE2-Fc could augment ACE2 enzymatic activity and reduce systemic inflammation. A lower plasma Ang II level after taking recombinant ACE2 may be safe in healthy volunteers and patients with acute respiratory distress syndrome or pulmonary arterial hypertension as demonstrated in the previous pilot studies (Haschke *et al*, 2013; Hemnes *et al*, 2018; Khan *et al*, 2017). Therefore, for RAS disorder patients with COVID-19 treatment, administration of ACE2-Fc in circulation may block SARS-CoV-2 infection even in those subjects with induced ACE2 after receipt of ARBs or ACEis treatment. Nevertheless, it should be

noted that similar to ARBs and ACEis, administration of ACE2-Fc may be harmful to the developing fetus during pregnancy since it could lower blood pressure by converting Ang II to Ang 1-7.

Q5- demonstrating the effect of the ACE2-Fc on different clinical isolates

Response:

We thank the Editor for the suggestion to explore the potential application of our ACE2-Fc decoy antibody to variant SARS-CoV-2 clinical strains. We have examined the inhibitory effects of the ACE2-Fc on six clinical isolated strains of SARS-CoV-2. As shown in Fig 5 and 6, the study results clearly demonstrated that ACE2-Fc could block these six strains SARS-CoV-2 infection, including the variant strains bearing the D614G mutation, which has been reported to increase viral infectivity after adaption during human-human transmission. These additional results have been incorporated in the revised manuscript.

(Page 7, line 3)

ACE2-Fc blocks SARS-CoV-2 entry and replication

The inhibitory effects of ACE2-Fc on viral entry was further confirmed using real SARS-CoV-2 isolates from patients suffering from COVID-19 infection at National Taiwan University Hospital. As expected, the preincubation of SARS-CoV-2 with ACE2-Fc blocked plaque formation in Vero E6 cells (Fig 5A and Fig EV5). The half-maximal effective concentrations (EC50) value for ACE2-Fc was 23.8 ± 5.94 $\mu\text{g/mL}$. The inhibitory effect was subsequently verified by the yield reduction assay. Pretreatment of SARS-CoV-2 with ACE2-Fc reduced the SARS-CoV-2 RNA copies in the culture supernatant (Fig 5B) and the SARS-CoV-2 nucleoprotein expression in the infected cells (Fig 5C). In addition, we extended the incubation period of virus-ACE2-Fc from 1 hr to 48 hr to examine whether resistant viruses would emerge (Fig 5D). Notably, comparable inhibitory effects on viral protein expression and supernatant viral RNA were observed when the ACE2-Fc was present in the culture medium for 48 hours as compared to the pretreatment group (Fig 5E and 5F).

To determine whether ACE2-Fc also exhibit inhibitory effects on other circulating SARS-CoV-2 strains, five other clinical SARS-CoV-2 strains, NTU3, NTU13, NTU14, NTU25, and NTU27, were included for analysis, and their genetic characteristics were summarized in Table 1. Notably, NTU3, NTU14 and NTU25 strains harbor the D614G mutation, which has been reported to increase the viral infectivity (Korber *et al*, 2020; Yurkovetskiy *et al*, 2020). As shown in Fig. 6, ACE2-Fc exhibited a potent ability to block SARS-CoV-2 protein expression (Fig 6A) and viral RNA in the supernatants and infected cells (Fig 6B and 6C). In this study, we

further showed that our decoy antibody could block entry of various SARS-CoV-2 strains and no resistant virus could be selected after prolonged co-incubation of ACE2-Fc and SARS-CoV-2.

Figure 5. Blockage of SARS-CoV-2 entry into host cells by ACE2-Fc. (A) Inhibition of SARS-CoV-2 infection by ACE2-Fc in the plaque assay. Mixtures of ACE2-Fc and SARS-CoV-2 were incubated for one hour before adding to Vero E6 cells for another 1 hr at 37°C. The ACE2-Fc and SARS-CoV-2 premixtures were removed, and the cells were washed once with

PBS and overlaid with methylcellulose with 2% FBS for 5-7 days before being stained with crystal violet. TPCCK-treated trypsin: N-tosyl-L-phenylalanine chloromethyl ketone-treated trypsin. (B-C) Yield reduction assay was performed to determine the inhibitory effects of ACE2-Fc on virus titers and protein expression. The culture medium and cell extracts were harvest 24 hours post-infection for real-time PCR (B) and western blot (C). The NP/PCNA represents the relative NP expression as compared to that of PCNA, which served as a loading control. The NP/PCNA numbers below the panel are the ratios of NP/PCNA normalized to that of the human IgG control group. PCNA: Proliferating cell nuclear antigen. (D) Schematic illustration of the ACE2-Fc pretreatment and full-time experiment procedure, delineating the stages where the ACE2-Fc was present during the experiment. (E) The cell lysates from the pretreatment and full-time experiments were harvested and immunoblotted with the indicated antibodies. (F) The SARS-CoV-2 titer in the pretreatment and full-time experiments was analyzed by real-time PCR. Error bars represent the mean \pm SD, n=3. * $P < 0.05$, ** $P < 0.01$.

Figure 6. Neutralization activity of ACE2-Fc on different SARS-CoV-2 strains. Yield reduction assay was performed to determine the inhibitory effects of ACE2-Fc on entry of 5

different SARS-CoV-2 strains into Vero E6 cells. The (A) NP proteins in the cell lysates were determined by Western blot analysis. The virus RNA in the (B) culture medium and (C) cell lysates was quantified by real time RT-PCR. The genetic information and sequence reference numbers for the 5 SARS-CoV-2 strains were summarized in Table 1. Error bars represent the mean \pm SD. ** $P < 0.01$, *** $P < 0.001$. Experiments were performed at least three times with similar results.

Referee #1 (Comments on Novelty/Model System for Author):

SARS-Coronavirus-2 (SARS-CoV-2) causes pandemic coronavirus disease-19 (COVID-19) with nearly 6 million people infected and there is currently no effective treatment. Similar to the disease caused by SARS-CoV and MERS-CoV, SARS-CoV-2 its major phenotype is severe acute respiratory distress syndrome (de Wit et al 2016; Huang et al 2020). SARS-CoV-2 uses Angiotensin Converting Enzyme II (ACE2) bind directly to SARS-CoV-2 Spike protein (Walls et al 2020; Wrapp et al 2020; Wan et al 2020) and is used as cell entry receptor (Zhou et al 2020; Hoffmann et al 2020).

ACE2 peptidase activity negatively regulates Angiotensin II (AngII). Treatment with recombinant ACE2 protein prevent Ang II-induced hypertension and cardiac dysfunction (Minato et al 2020), might control AngII-AT1R proinflammatory cytokine responses (Eguchi et al 2018), and protects from severe acute lung failure (Imai et al 2005). As impaired ACE2 expression was observed in mice receiving SARS-CoV Spike protein injection (Kuba et al, 2005), it is postulated that SARS-CoV-2 Spike proteins hijack ACE2 function driving COVID19-associated ARDS (Hirano et al 2020). Monteil et al 2020 demonstrated that administration of clinical grade human recombinant ACE2 protein significantly block early stages of SARS-CoV-2 infections in vero E6 cells. This protein has a low half-life, thereby limiting its therapeutical potential.

With this in mind, the authors generated recombinant ACE2 fused to Fc that is known to increase its half-life (Liu et al 2018). Using ELISA they demonstrate that recombinant ACE2-Fc binds recombinant SARS-CoV-2 Spike Fc protein. They show that pre-incubation of Spike protein pseudotyped lentivirus or clinically isolated SARS-CoV-2 with ACE2-Fc reduces infection of ACE2 overexpressing 293T cell line and Vero E6 cells, respectively. Interestingly, their ACE2-Fc has peptidase activity as demonstrated by cleaving fluorescent substrate.

The results are clear and I didn't find any obvious problems with the design and execution of the experiments. However, the concept of using recombinant ACE2 (Monteil et al 2020) and fusing it to Fc has previously been established (Lei et al 2020). Importantly, these authors' ACE2-Fc seem to be less efficient in virus neutralization with 25ug/ml resulting in 60% neutralization versus 0.1ug/ml described in Lei et al 2020. Although a direct comparison should not be made as different pseudotyped viruses, cells, and MOIs were used, there is no clear benefit of these authors ACE2-Fc.

The idea that enzymatically active ACE2-Fc protects against lung injury in COVID19 is very interesting, but has not been demonstrated here or by others. Caution should be taken as administration of recombinant ACE2 is also able to inhibit myocardial remodelling, attenuate Ang II-induced cardiac hypertrophy, and cardiac dysfunction (Huentelman et al 2005).

In conclusion, this manuscript lacks novelty and there is no clear benefit above the previously reported ACE2-fc. Although the ACE2-Fc recombinant proteins seem promising in combating COVID19 and further analyses in their ability to neutralize SARS-CoV-2 infection in vitro and in animals should be stimulated, this is not enough to warrant publication.

Response:

We very appreciate the Reviewer's comments. Compared to the recent publication by Monteil et al., who described the inhibitory effects of soluble human ACE2 against SARS-CoV-2 infection, our decoy antibody ACE2-Fc is a fusion protein with Fc, which was previously shown to significantly increase the half-life of ACE2 protein (Lei *et al*, 2020). In addition, based on previous findings in mouse ACE2 (Wysocki *et al.*, 2019), two shorten mouse ACE2 variants (1-605 and 1-619 A.A.) have been shown to exhibit higher ACE2 enzyme activity than that of the full-length ACE2 (1-740 A.A.). In addition, ACE2 enzyme activity could be detected in urine from *Ace2* knockout mice after intravenous infusion of the mouse variant (1-619 A.A.), but not ACE2 (1-740 A.A.). Infusion of this short variant (ACE2 1-619 A.A.) also recovered the ACE2 activity of the kidney in the *Ace2-deficient* mice. These evidences supported that the shortened version of the ACE2-Fc (1-619 A.A.) is more stable in plasma and retains higher enzyme activity to convert Ang II to Ang 1-7. Our data also demonstrated that the ACE2-Fc is stable and retain binding activity to the Spike proteins after being incubated with plasma for up to ten days (Fig EV3C). Secondly, we increased the numbers of SARS-CoV-2 clinical isolates to demonstrate that ACE2-Fc can efficiently inhibit virus variants, including the D614G variants which have been shown to exhibit increased infectivity (Fig 6). The study results further extend our understanding that the ACE2-Fc can still inhibit SARS-CoV-2 efficiently in terms of the heterogeneity of the clinical isolates. Finally, the benefits of adding the Fc domain to the ACE2 have been demonstrated. The ACE2-Fc decoy antibody was shown to preserve the peptidase activity of ACE2-Fc to reduce the Ang II-mediated cytokine cascade. After co-incubation of Ang II with ACE2-Fc, we observed that ACE2-Fc could significantly suppress Ang II-induced TNF- α production (Fig 2E) and phosphorylation of ADAM17 (a disintegrin and metalloprotease 17) (Fig 2F). Finally, ACE2-Fc was shown to induce NK cell activation and antibody-dependent cellular cytotoxicity (ADCC), which may help to remove the infected cells *in vivo* (Fig 7). These

additional experiment results have been provided in the revised manuscript to strengthen the novelty of our study.

Figure EV3C

Figure EV3 *in vitro* cytotoxicity and plasma stability and of ACE2-Fc. (C) *In vitro* serum stability of ACE2-Fc. ACE2-Fc was incubated with 50% normal human serum at 37°C for up to ten days. At the indicated time points, samples were collected to quantify the binding ability of ACE2-Fc to Spike proteins by ELISA. Error bars represent the mean \pm SD; n=3. Experiments were performed at least three times with similar results.

Figure 6. Neutralization activity of ACE2-Fc on different SARS-CoV-2 strains. Yield reduction assay was performed to determine the inhibitory effects of ACE2-Fc on the entry of 5 different SARS-CoV-2 strains into Vero E6 cells. The (A) NP proteins in the cell lysates were determined by Western blot analysis. The virus RNA in the (B) culture medium and (C) cell lysates was quantified by real-time RT-PCR. The genetic information and sequence reference numbers for the 5 SARS-CoV-2 strains were summarized in Table 1. Error bars represent the mean \pm SD. ** $P < 0.01$, *** $P < 0.001$. Experiments were performed at least three times with similar results.

Figure 2. Functional characterization of the decoy antibody. (E) Inhibition of angiotensin II (Ang II)-induced TNF- α production by ACE2-Fc. Ang II was preincubated with indicated amounts of ACE2-Fc or IgG for 30 minutes. After that, the mixtures were added to RAW264.7 cells for 12 hr. The concentration of TNF- α in the culture medium was determined by ELISA. (F) Inhibition of Ang II-induced ADAM17 (a disintegrin and metalloprotease 17) phosphorylation by ACE2-Fc. The protein extracts from ϵ were immunoblotted with the indicated antibodies (left panel). β -actin served as the loading control. ADAM17 represented. The signal intensity was normalized to cells only (right panel). Data are representative of three independent experiments, and the values are expressed as the mean \pm SD (lower panel). Error bars represent the mean \pm SD, $n=3$. * $P < 0.05$. IgG represents the human normal IgG control. Experiments were performed at least three times with similar results.

Figure 7. Effects of ACE2-Fc on NK cell degranulation. IL-2 activated NK cells were incubated with H1975-Spike cells at a 1:1 cell ratio in the presence of ACE2-Fc or ACE2. The activation of NK cells were determined by the CD107a, IFN- γ , and TNF- α expression levels. (A) H1975 cells were transduced with full-length Spike by Lentiviral vector. Protein extracts were immunoblotted with the indicated antibodies. O/E represents overexpress. (B) The effect of ACE2-Fc activation on degranulative capacity of NK cells. The experiments were performed with the primary human NK cells derived from three independent donors. Error bars represent mean \pm SD, n = 3. * P < 0.05, *** P < 0.001. (C) The CD107a, IFN- γ , and TNF- α expression levels were determined by flow cytometry after the NK cells and H1975-Spike cells co-cultivation in the

presence or absence of ACE2-Fc or ACE2 control. Error bars represent the mean \pm SD. The percentage of positive cells in both groups were normalized to the NK/H1975-Spike only group. (D) The differential expression of degranulation markers after the treatment of ACE2-Fc or ACE2 by the flow cytometry. The numbers in each plot indicate the percentages of positive cells.

Referee #1 (Remarks for Author):

Major comments:

Q1. Spike proteins form trimers in the viral and host cell membrane, but here the authors connect Spike to Fc forming presumably dimers. To show binding of ACE2-Fc to trimeric Spike protein demonstrate that ACE2-Fc stains only cells infected with Spike protein pseudotyped lentivirus and not cells infected by VSVG-lentivirus.

Response:

We completely agree with the Reviewer on this point. Experiments were conducted to provide more evidence to support that ACE2-Fc can target trimeric spike proteins on the cell surface. First, we showed that ACE2-Fc could only bind to the cell surface of spike expressing H1975 cells in a dose-dependent manner by the flow cytometry, but not to that of the parental H1975 cells (Fig 2B). In addition, the confocal microscopy data revealed substantial colocalization of the spike (red) and ACE2-Fc (green) (Fig 2C). After transduction of the full-length spike into H1975 cells using the lentivirus system, we confirmed that these cells could induce the cell-cell fusion or syncytia formation, which can be efficiently blocked by the decoy antibody ACE2-Fc (Fig 3). These additional evidences have been incorporated in the Results section as following:

(Page 5, line 5)

In addition, ACE2-Fc could bind to the cell surface of human lung adenocarcinoma H1975 cells expressing full-length Spike protein in a dose-dependent manner (Fig 2B). The colocalization of the FITC-conjugated ACE2-Fc and anti-Spike antibody by confocal microscopy further confirm the specific recognition of the Spike proteins by the ACE2-Fc (Fig 2C).

(Page 5, line 21)

ACE2-Fc inhibits SARS-CoV-2 Spike-mediated cell-cell fusion and syncytia formation

To determine whether the decoy antibody is able to inhibit SARS-CoV-2 fusion with the target cells, we cotransfected SARS-CoV-2 Spike protein and EGFP into the HEK293T cells as the effector cells (293T-S) and used the ACE2-stable-expressing HEK293T and H1975 cells as the target cells (293T-ACE2 and H1975-ACE2) (Fig 3B). The target cells without ACE2 overexpression were used as controls. The effector cells (293T-S) were preincubated with ACE2-Fc or IgG at 37°C for one hour before mixing with the target cells or control cells for

another 4 hours (cell-cell fusion assay) or 24 hours (syncytia formation assay) (Fig 3A). The area of EGFP was counted, and the inhibitory effect of ACE2-Fc on SARS-CoV-2 Spike-mediated cell-cell fusion and syncytia formation was quantified (n=6). As shown in Fig 3C and 3D, ACE2-Fc significantly impaired SARS-CoV-2 Spike-mediated cell-cell fusion and syncytia formation compared to the normal human IgG control in both the HEK293T and the H1975 cell systems. These results demonstrated that ACE2-Fc could block SARS-CoV-2 infection via abrogating virus-mediated cell-cell fusion and syncytium formation.

Figure 2. Functional characterization of the decoy antibody. (B) Recognition of full-length Spike on H1975 cells by ACE2-Fc using flow cytometry analysis. Isotype control: 40 µg/mL mouse IgG-FITC. (C) Confocal microscopy of H1975-Spike-overexpressing cells. Spikes on H1975 cells were stained with anti-Spike antibody and Alexa Fluor® 594-conjugated secondary antibody on ice for 1 hr. After that, ACE2-Fc-FITC was incubated for another 1 hr. Scale bars equal to 20 µm. DAPI was used as a nuclear counterstain.

Figure 3. Inhibition of Spike-induced cell-cell fusion and syncytia formation by ACE2-Fc.

(A) The schematic diagram for cell-cell fusion and syncytia formation. (B) HEK293 and H1975 cells were transduced with full-length ACE2 by lentivirus. Protein extracts were immunoblotted with the indicated antibodies. O/E represents overexpression. (C) GFP and full-length Spike co-transfected HEK293T cells were used as effector cells (293-S). HEK293/ACE2 (HEK293T-overexpressing ACE2) (left panel) or H1975/ACE2 (H1975-overexpressing ACE2) (right panel) cells were used as target cells. The 293-S cells were preincubated with normal human IgG or ACE2-Fc at 37°C for 1 hr. After that, the mixtures were added to target cells for 4 hr (cell-cell fusion) or 24 hr (syncytia formation). The fluorescent areas were measured by inverted fluorescence microscopy and Metamorph software (Metamorphosis). Scale bars equal to 50 µm. (D) The inhibition of cell-cell fusion and syncytia formation by ACE2-Fc in HEK293/ACE2 and H1975/ACE2 was determined with the formula described in the Methods

section. Error bars represent the mean \pm SD, n=6. ** $P < 0.01$, *** $P < 0.001$. Experiments were performed at least three times with similar results.

Q2. ACE2-Fc interfering with Spike-ACE2 interaction is dose-dependent. Therefore, neutralization of Spike protein pseudotyped viral infection should be examined in at least one more experiment with a higher viral dose.

Response:

We thank the Reviewer for the suggestion. The neutralization activities of ACE2-Fc against different virus input (MOI=0.1 and MOI=1) in ACE2-Fc-overexpressed HEK293 or ACE2-Fc-overexpressed H1975 have been added into the section of “**ACE2-Fc blocks pseudotyped lentivirus entry into ACE2-expressing cells and lung organoids**”. As shown in Fig EV4, dose-dependent inhibition of viral entry was also observed at the virus input of 1 MOI. These additional data have been described in the Results section, page 6, line 14 with Fig 4A and Fig EV4:

(Page6, line13)

ACE2-Fc blocks entry of pseudotyped lentivirus into ACE2-expressing cells and lung organoids

The ability of ACE2-Fc to block SARS-CoV-2 entry was first examined by using the Spike-expressing pseudotyped lentivirus whose backbone G protein of vesicular stomatitis virus (VSVG) was replaced with the SARS-CoV-2 Spike protein. The pseudotyped virus was preincubated with either ACE2-Fc or human IgG1 for one hour at 37°C before adding to the ACE2-expressing HEK293T cells (293T-ACE2) or parental HEK293T cells for another hour. The entry of Spike-expressing pseudotyped virus into host cells was specifically mediated by ACE2 expression (Fig 4A). The blockage of viral entry by ACE2-Fc was not only observed in HEK293T cells but also in another ACE2-expressing H1975 cell (H1975-ACE2) (Fig 4A). Similar neutralization effects were also observed at a 10-fold higher virus input (Fig EV4).

Figure 4A

Figure EV 4

Figure 4. Blockage of Spike-expressing pseudovirus entry into ACE2-expressing cells by ACE2-Fc. (A) ACE2-Fc blocked the entry of Spike-expressing pseudotyped lentivirus into HEK293T-ACE2 and H1975-ACE2 cells. The relative luciferase activities, normalized to the only virus group, represent the efficiency of virus entry. MOI: Multiplicity of infection. The virus entry was determined by quantifying the luciferase activity in the cell lysates. Error bars represent the mean \pm SD, n=3. ** $P < 0.01$, *** $P < 0.001$. Experiments were performed at least three times with similar results.

Figure EV4 Inhibition of pseudovirus entry by ACE2-Fc at higher virus input. The same experimental protocol was conducted as in Fig 4A, except with a higher virus input. ACE2-Fc blocked Spike-expressing pseudotyped lentivirus entry into HEK293T-ACE2 and H1975-ACE2 cells. MOI (Multiplicity of infection) =1. Virus entry was determined by measuring luciferase activity. R.I. U: (R.I. U=relative infection unit. Error bars represent the mean \pm SD, n=3. ** $P < 0.01$, *** $P < 0.001$.

Q3. It is crucial to extend the neutralizing activity of ACE2-Fc to lung cell infection with clinical isolated SARS-CoV-2.

Response:

We thank the Reviewer for noticing this detail. As requested, we have generated ACE2-expressing H1975 lung adenocarcinoma cells (Fig 3B). Experiments were performed to determine the neutralizing activity of ACE2-Fc against clinical SARS-CoV-2 viruses at the MOI of 0.1 and 0.5 in ACE2-expressing H1975 lung cells. A significant reduction of viral RNA expression was observed when ACE2-Fc was pre-incubated with the SARS-CoV-2 viruses as compared to that of the IgG control.

Figure Blockage of SARS-CoV-2 entry into ACE2 expressing H1975 cells by the decoy antibody. ACE2-Fc and SARS-CoV-2 premixtures were incubated for one hour before being added to ACE2 expressing H1975 cells for another 1 hr at 37°C. After that, the cells were washed once with PBS before refreshing with fresh medium for 24 hr. The RNA was extracted from the cell lysates and the viral RNA expression was determined by the real-time PCR.

Q4. A slightly larger recombinant ACE2-Fc (Lei et al 2020) has been demonstrated to neutralize SARS-CoV-2 Spike protein pseudotyped viral infection more efficient than these authors' ACE2-Fc. Are there any benefits of your ACE2-Fc in comparison to the previously published ACE2-Fc? For example, does NFkB activity or signs of injury reduce upon administration of enzymatically active ACE2-Fc during SARS-CoV-2 infection of lung cells (ideally lung organoids).

Response:

We appreciate the Reviewer's frank comments here. Based on previous findings in mouse ACE2 (Wysocki *et al.*, 2019), two shorten mouse ACE2 variants (1-605 and 1-619 A.A.) have been shown to exhibit higher ACE2 enzyme activity than that of the full-length ACE2 (1-740 A.A.). In addition, ACE2 enzyme activity could be detected in urine from *Ace2* knockout mice after intravenous infusion of the mouse variant (1-619 A.A.), but not the full-length ACE2 (1-740 A.A.). Infusion of this short variant (ACE2 1-619 A.A.) also recovered the ACE2 activity of the kidney in the ACE2-deficient mice. These evidences supported that the shortened version of the ACE2-Fc (1-619 A.A.) is more stable in plasma and retains higher enzyme activity to convert Ang II to Ang 1-7 than the full-length ACE2. In our study, we first confirmed that the shorten ACE2-Fc (18-615 A.A.) **retains the strong binding ability to the spike** proteins after coincubation with 50% normal human serum for up to 10 days (Fig EV3C). The shorter ACE2-Fc was **not toxic to two independent normal bronchial epithelial cells** (Fig EV3A and EV3B). Third, this shorter ACE2-Fc still maintains the peptidase activity which could convert Ang II to Ang 1-7 (Fig 2D). Subsequently, a functional assay was performed to confirm that ACE2-Fc could **suppress Ang II inducing inflammatory response** (Fig 2E and 2F). Finally, we confirmed that this shorter ACE2-Fc could still activate the **degranulation of the NK cells** whereas the recombinant ACE2 could not (Fig 7).

In this revised manuscript, we have provided additional study results to characterize the functional role of ACE2-Fc in human epithelial cells as well as the human airway organoids. Please see the below Table for the comparison between experimental data provided in our

revised article and those in the previous publications. We have added these results into our revised manuscript, and hope that these evidences can increase the significance of our manuscript.

Evidences		EMBO MM		Lei et al, 2020	Monteil et al, 2020
1	Cytotoxicity in normal cells	Yes	Fig EV3A-B	No	No
2	Half-life in serum	Yes	Fig EV3C	No	No
3	ACE2-Fc : Spike interaction on the cell surface	Yes	Fig 2B-C	No	No
4	ACE2-Fc peptidase activity for inflammatory response	Yes	Fig 2D-F	No	No
5	Cell-cell fusion and Syncytia formation	Yes	Fig 3	Yes	No
6	Airway organoid system	Yes	Fig 4B-D	No	* other organoids
7	Real virus of SARS-CoV-2	Yes	Fig 5	No	Yes
8	SARS-CoV-2 with D614G mutation and others strain	Yes	Fig 6	No	No
9	NK cell activation	Yes	Fig 7	No	No

Q5. *It should be determined whether passaging of SARS-CoV-2 in the presence of ACE2-Fc decreases viral infection of neighbouring cells and syncytia formation. Ideally, resistance development should be tested.*

Response:

We appreciate the Reviewer’s suggestion. As requested, an experiment was performed to determine whether resistant SARS-CoV-2 would emerge in the presence of ACE2-Fc for a longer period of incubation, and the results were added into the section of “**ACE2-Fc blocks SARS-CoV-2 entry and replication**”. We extended the incubation period of virus-ACE2-Fc from 1 hr to 48 hr to examine whether resistant viruses would emerge (Fig 5D). Notably, comparable inhibitory effects on viral protein expression and supernatant viral RNA were observed when the ACE2-Fc was present in the culture medium for 48 hours as compared to the pretreatment group (Fig 5E and 5F).

(Page 7, line 12)

ACE2-Fc blocks SARS-CoV-2 entry and replication

In addition, we extended the incubation period of virus-ACE2-Fc from 1 hr to 48 hr to examine whether resistant viruses would emerge (Fig 5D). Notably, comparable inhibitory effects on viral protein expression and supernatant viral RNA were observed when the ACE2-Fc was present in the culture medium for 48 hours as compared to the pretreatment group (Fig 5E and 5F).

D.

E.

F.

Figure 5. Blockage of SARS-CoV-2 entry into host cells by ACE2-Fc. (D) Schematic illustration of the ACE2-Fc pretreatment and full-time experiment procedure, delineating the stages where the ACE2-Fc was present during the experiment. (E) The cell lysates from the pretreatment and full-time experiments were harvested and immunoblotted with the indicated antibodies. (F) The SARS-CoV-2 titer in the pretreatment and full-time experiments was analyzed by real-time PCR. Error bars represent the mean \pm SD, n=3. * $P < 0.05$, ** $P < 0.01$.

Minor comments:

Q1. Could the authors indicate what statistical method was used and what the "*" indicates in the figure legends.

Response:

We appreciate the Reviewer's suggestion. We have added the statistical method into the Section of "Statistical analysis" and indicated "*" represents $P < 0.05$ in the figure legend.

Q2. Could the authors explain how IgG condition in Fig 2B could have a negative value?

Response:

We thank the Reviewer for noticing this detail. In Fig 5A (the original Fig 2B), the percentage of inhibition was calculated as $[1 - (VD/VC)] \times 100\%$, where VD and VC refer to the virus titers in the presence and absence of the ACE2-Fc or IgG. Due to each condition contained at least three repeats, the variation sometimes might lead to negative values (the raw data as shown in Fig EV5). If the Reviewer prefer to prevent confusion, we could convert the negative values into zero by adding a sentence of description, "Those results showed increase of plaque formation was regarded as no inhibition of plaque formation".

Figure 5A.

Figure EV5

Figure 5. Blockage of SARS-CoV-2 entry into host cells by ACE2-Fc. (A) Inhibition of SARS-CoV-2 infection by ACE2-Fc in the plaque assay. Mixtures of ACE2-Fc and SARS-CoV-2 were incubated for one hour before adding to Vero E6 cells for another 1 hr at 37°C. The ACE2-Fc and SARS-CoV-2 premixtures were removed, and the cells were washed once with PBS and overlaid with methylcellulose with 2% FBS for 5-7 days before being stained with crystal violet. TPCK-treated trypsin: N-tosyl-L-phenylalanine chloromethyl ketone-treated trypsin.

Figure EV5 Plaque reduction assay. SARS-CoV-2 (4000 plaque-forming units, PFUs) was incubated with antibodies at the indicated amounts for 1 hr at 37°C before adding to the Vero E6 cell monolayer for another hour. After the mixtures were removed, the cells were washed and replaced with an overlay medium for 5 days. Plaque formation was determined by crystal violet staining. NTU01: SARA-CoV-2 that was isolated from a female patient suffering from COVID-19 infection at National Taiwan University Hospital.

Q3. Fig2D should have a logarithmic y-axis with fold change in comparison to the IgG treated condition as viral replication happens exponentially.

Response:

We thank for the Reviewer’s comment. A logarithmic y-axis with fold change in comparison to the IgG treated condition has been updated in the revised Fig 5B.

Figure 5. Blockage of SARS-CoV-2 entry into host cells by ACE2-Fc. (B-C) Yield reduction

assay was performed to determine the inhibitory effects of ACE2-Fc on virus titers and protein expression. The culture medium and cell extracts were harvest 24 hours post-infection for real-time PCR (B).

Q4. The authors themselves already indicate the importance of Spike protein glycosylation for Spike-ACE2 interaction. Could they show by PNGF-treatment that their engineered Spike proteins are indeed glycosylated.

Response:

We thank the Reviewer for giving this constructive comment. Experiments have been performed to determine the effects of PNGF-treatment on the Spike proteins by SAS-PAGE and Western blotting. The experimental results have been incorporated into the section of “Production and functional assay of the ACE2-Fc decoy antibody” and Fig 1E and 1F.

(Page 4, line 33)

The ACE2-Fc and Spike 1-674-Fc protein are likely to be heavily N-glycosylated since size reduction was observed in SDS-PAGE after PNGase F (Peptide: N-glycosidase F) treatment (Fig 1E and 1F).

Figure 1. Production of the decoy antibody (chimeric ACE2-Fc). (D-E) Deglycosylation of ACE2-Fc and Spike 1-674-Fc by PNGase F. PNGase F digested ACE2-Fc (500 ng) and Spike 1-674-Fc (500 ng) were subjected to Coomassie Brilliant Blue staining (E) and Western blot analysis by anti-human IgG Fc antibody (F). NTD: N-terminal domain; RBD: receptor-binding domain; SD: connector domain; TM: transmembrane domain; CT: cytoplasmic tail; FP: fusion peptide. IB, immunoblotted with the indicated antibodies. GAPDH served as a loading control. Experiments were performed at least three times with similar results.

Q5.

- 1) *SARA instead of SARS in start of line 16 in paragraph "ACE2-Fc blocks SARS-CoV-2 entry and replication".*
- 2) *Capital "I" in 12th line of methods paragraph "estimation of lentiviral titer by using luciferase assay".*
- 3) *Extra space in 4th line of methods paragraph "Plague reduction assay"*

Response:

We thank the Reviewer for pointing out these typos. We have carefully revised our manuscript accordingly. English editing and proof-reading of the revised manuscript have been performed.

Referee #2 (Comments on Novelty/Model System for Author):

This study investigated the suitability of humanised ACE2-Fc fusion as decoy antibodies for preventing the entry of SARS-COV-2 into human cells invitro. The ACE2-Fc fusion protein can form dimer mimicking antibodies and specifically binds to SARS-CoV-2 Spike protein. By competitively binding to the viral Spike protein, the decoy antibodies prevent viral binding to the natural, membrane-bound ACE2, and thus blocks virus entry into host cells. The study utilised virus from clinical isolates and demonstrate that the ACE2-Fc fusion protein abrogated viral replication in Vero E6 cells. This study supports the recently published work by Monteil, and colleagues (Reference 9) who showed that recombinant purified human ACE2 (hrsACE2) - a decoy protein that has already been tested in phase I and II clinical trials can inhibit the interaction between SARS-CoV-2 and ACE2. Treatment of cells with hrsACE2 inhibited SARS-CoV-2 infection in a dose-dependent manner and attenuated propagation of the virus. Using two human organoid models the Monteil et al., showed that hrsACE2 indeed inhibited the virus from infecting the host cells.

This paper in review presents an improvement to Monteil et al.'s work by generating this antibody decoy. The ACE2-Fc fusion which was previously shown to significantly increase the half-life of the ACE2 protein (Liu et al. (12), which may likely be the case by extension. They tested this decoy on clinical isolates, although very little detail has been provided about the number of isolates that were tested. The Monteil paper tested clinical isolates from a single patient, so without clarifying how many isolates it is not clear how much extra knowledge can be learnt from this study from the point of view of heterogeneity of the isolates. There seem to be no direct evidence from this study of the benefit of adding the Fc domain to the ACE2. This is a main limitation of this work as the benefits of making this Fc-ACE2 fusion are based on previous publications with no effort to demonstrate directly the value of making this modification. This is very important as this new ACE2-Fc fusion will require rigorous testing and clinical trials before it can be used in therapy.

1. The author suggest that the fusion could trigger antibody -dependent cellular cytotoxicity and complement dependent cytotoxicity. However, this work does not address this added benefit of fusing this ACE-2 protein to make an antibody decoy. Based on published literature the authors also suggest that the peptidase activity may enable ACE2-Fc to reduce angiotensin II mediated cytokine cascade during SARS-CoV-2 infection (Hirano & Murakami, 2020). However, this work does not address any of these supposed added benefits of fusing the Fc region or the peptidase activity. Also, the discussion does not clearly articulate these benefits. It will be good to see at least one experiment which demonstrate the increased half-life and its benefit, as well as lack of toxicity of this decoy to

the cells in particular due to its supposed longer half-life.

2. The design of the study focusses on the early stages of infection, as they preincubated the virus with the decoy fusion protein before adding to Vero-6 cells or to HEK293 cells. This study cannot be used to predict the effect of the ACE2-Fc fusion in later stages of the disease process. Could the authors add the decoy to cells that are infected already and see how this decoy altered the course of infection when it has already established?

3. There was no attempt to study the effect in a disease model such as in lung organoids or in mice or at the very least discussing future work along these lines.

4. The RAS system represents a complex network of pathways that are influenced by many factors, hence it is not clear how the ACE2-Fc will impact on this system. All limitations listed above have not been discussed in this study.

5. The discussion should clarify how the authors see this decoy being employed as a therapeutic strategy. Such as to deliver the decoy product by inhaler given to healthcare workers or those people at risk as for preventing infection. This might be met with better outcomes as disease might be milder as this could be a 'protective shield' against Covid-19 by inhibiting and neutralising the virus' activity in naïve at-risk populations.

6. "The viral entry blocking effect of ACE2-Fc was further confirmed using real SARA-CoV-2 isolated from patients suffering from COVID-19 infection in National Taiwan University Hospital". There are no further details in the methods regarding how the isolates were obtained including clarity in the methods as to how many isolates were tested. Were the isolates sequenced or identified by some method? Were they the same or different variants of SARS-COV-2?

7. Ethics relating to the collection of isolates have not been clarified if this is a consideration.

8. Statistical analysis should clarify number of repeats

Response:

We really appreciate the Reviewer's comments, which allowed us to further improve our manuscript. Compared to the recent publication by Monteil et al. who described the inhibitory effects of soluble human ACE2 against SARS-CoV-2 infection, our decoy antibody ACE2-Fc is a fusion protein with Fc, which was previously shown to significantly increase the half-life of the ACE2 protein (Lei *et al.*, 2020). In addition, based on previous findings in mouse ACE2 (Wysocki *et al.*, 2019), two shorten mouse ACE2 variants (1-605 and 1-619 A.A.) were shown to exhibit higher ACE2 enzyme activity than that of the full-length ACE2 (1-740 A.A.). In addition, ACE2 enzyme activity could be detected in urine from *Ace2* knockout mice after intravenous infusion of the mouse variant (1-619 A.A.), but not the full-length ACE2 (1-740 A.A.). Infusion of this short variant (ACE2 1-619 A.A.) also recovered the ACE2 activity of the kidney in the

Ace2-deficient mice. These evidences supported that the shortened version of the ACE2-Fc (1-619 A.A.) is more stable in plasma and retains higher enzyme activity to convert Ang II to Ang 1-7 than the full-length ACE2-Fc. Our data also demonstrated that the ACE2-Fc is stable and retain binding activity to the Spike proteins after being incubated with plasma for up to ten days (Fig EV3C). Secondly, we increased the numbers of SARS-CoV-2 clinical isolates to demonstrate that ACE2-Fc can efficiently inhibit virus variants, including the D614G variants which have been shown to exhibit increased infectivity. The study results further extend our understanding that the ACE2-Fc can still inhibit SARS-CoV-2 efficiently in terms of the heterogeneity of the clinical isolates (Fig 6). Finally, the benefits of adding the Fc domain to the ACE2 have been demonstrated. The ACE2-Fc decoy antibody was shown to preserve the peptidase activity of ACE2-Fc to reduce the Ang II-mediated cytokine cascade. After the coincubation of Ang II with ACE2-Fc, we observed that ACE2-Fc could significantly suppress Ang II-induced TNF- α production (Fig 2E) and phosphorylation of ADAM17 (a disintegrin and metalloprotease 17) (Fig 2F). Finally, ACE2-Fc was shown to induce NK cell activation and antibody-dependent cellular cytotoxicity (ADCC), which may help to remove the infected cells *in vivo* (Fig 7). These additional experiment results have been incorporated into the revised manuscript to strengthen the novelty and significance of our study.

Figure EV3C

Figure EV3 *in vitro* cytotoxicity and plasma stability and of ACE2-Fc. (C) *In vitro* serum stability of ACE2-Fc. ACE2-Fc was incubated with 50% normal human serum at 37°C for up to ten days. At the indicated time points, samples were collected to quantify the binding ability of ACE2-Fc to Spike proteins by ELISA. Error bars represent the mean \pm SD; n=3. Experiments were performed at least three times with similar results.

Figure 6. Neutralization activity of ACE2-Fc on different SARS-CoV-2 strains. Yield reduction assay was performed to determine the inhibitory effects of ACE2-Fc on the entry of 5 different SARS-CoV-2 strains into Vero E6 cells. The (A) NP proteins in the cell lysates were determined by Western blot analysis. The virus RNA in the (B) culture medium and (C) cell lysates was quantified by real-time RT-PCR. The genetic information and sequence reference numbers for the 5 SARS-CoV-2 strains were summarized in Table 1. Error bars represent the mean \pm SD. ** $P < 0.01$, *** $P < 0.001$. Experiments were performed at least three times with similar results.

Figure 2. Functional characterization of the decoy antibody. (E) Inhibition of angiotensin II (Ang II)-induced TNF- α production by ACE2-Fc. Ang II was preincubated with indicated

amounts of ACE2-Fc or IgG for 30 minutes. After that, the mixtures were added to RAW264.7 cells for 12 hr. The concentration of TNF- α in the culture medium was determined by ELISA. (F) Inhibition of Ang II-induced ADAM17 (a disintegrin and metalloprotease 17) phosphorylation by ACE2-Fc. The protein extracts from ϵ were immunoblotted with the indicated antibodies (left panel). β -actin served as the loading control. ADAM17 represented. The signal intensity was normalized to cells only (right panel). Data are representative of three independent experiments, and the values are expressed as the mean \pm SD (lower panel). Error bars represent the mean \pm SD, n=3. * $P < 0.05$. IgG represents the human normal IgG control. Experiments were performed at least three times with similar results.

Figure 7. Effects of ACE2-Fc on NK cell degranulation. IL-2 activated NK cells were incubated with H1975-Spike cells at a 1:1 cell ratio in the presence of ACE2-Fc or ACE2. The activation of NK cell were determined by the CD107a, IFN- γ , and TNF- α expression levels. (A) H1975 cells were transduced with full-length Spike by Lentiviral vector. Protein extracts were immunoblotted with the indicated antibodies. O/E represents overexpress. (B) The effect of

ACE2-Fc activation on degranulative capacity of NK cells. The experiments were performed with the primary human NK cells derived from three independent donors. Error bars represent mean \pm SD, n= 3. * P<0.05, *** P< 0.001. (C) The CD107a, IFN- γ , and TNF- α expression levels were determined by flow cytometry after the NK cells and H1975-Spike cells coincubation in the presence or absence of ACE2-Fc or ACE2 control. Error bars represent the mean \pm SD. The percentage of positive cells in both groups were normalized to the NK/H1975-Spike only group. (D) The differential expression of degranulation markers after the treatment of ACE2-Fc or ACE2 by the flow cytometry. The numbers in each plot indicate the percentages of positive cells.

Referee 2

Q1. The author suggest that the fusion could trigger antibody -dependent cellular cytotoxicity and complement dependent cytotoxicity. However, this work does not address this added benefit of fusing this ACE-2 protein to make an antibody decoy. Based on published literature the authors also suggest that the peptidase activity may enable ACE2-Fc to reduce angiotensin II mediated cytokine cascade during SARS-CoV-2 infection (Hirano & Murakami, 2020). However, this work does not address any of these supposed added benefits of fusing the Fc region or the peptidase activity. Also, the discussion does not clearly articulate these benefits. It will be good to see at least one experiment which demonstrate the increased half-life and its benefit, as well as lack of toxicity of this decoy to the cells in particular due to its supposed longer half-life.

Response:

We thank the Reviewer for the critical suggestions. We have performed the suggested experiments to demonstrate the benefits of fusing this ACE-2 protein to make an antibody decoy. First, the decoy antibody ACE2-Fc is very stable since nearly 100 % of the spike-binding ability of ACE2-Fc was retained after incubated with 50 % normal human serum for up to 10 days (Fig EV3C). Second, the ACE2-Fc decoy antibody preserves the peptidase activity of ACE2-Fc to reduce the Ang II-mediated cytokine cascade. The purified ACE2-Fc retained peptidase activity as compared with the human normal IgG and buffer controls (Fig 2D). After the coincubation of Ang II with ACE2-Fc, we observed that ACE2-Fc could significantly suppress Ang II-induced TNF- α production (Fig 2E) and phosphorylation of ADAM17 (a disintegrin and metalloprotease 17) (Fig 2F). Third, ACE2-Fc was shown to induce NK cell activation and antibody-dependent cellular cytotoxicity (ADCC), which may help to remove the infected cells *in vivo* (Fig 7). We have incorporated these study results to our revised manuscript. The expected benefit as well as the potential limitations of our study design has also been added to the Discussion section. We hope that these modifications can increase the significance of our manuscript.

(Page 5, line 38)

Cytotoxicity and stability of ACE2-Fc.

To examine the potential cytotoxicity of ACE2-Fc on normal cells, two different normal human bronchial epithelial (NBE) cells were treated with various concentrations of ACE2-Fc or IgG for 3 days before the cell viability assay. As shown in Fig EV3A and EV3B, no cell toxicity was observed in these two normal cells at the concentration up to 200 $\mu\text{g/mL}$ of ACE2-Fc. The stability of ACE2-Fc in serum was subsequently determined. We incubated 2 $\mu\text{g/mL}$ ACE2-Fc in 50% normal human serum at 37°C for 0, 1, 2, and up to 10 days. The stability of ACE2-Fc was determined by assaying its binding ability to the Spike proteins in the ELISA assay. As shown in Fig EV3C, not a significant reduction of ACE2-Fc/Spike binding was observed up to ten days. These results suggest that ACE2-Fc has no toxicity to epithelial cells and may be stable in serum for ten days, which may facilitate its future clinical application.

(Page 7, line 28)

ACE2-Fc induced degranulation of natural killer (NK) cells.

Proteins fused to the Fc domain enable these molecules to interact with Fc receptors, which are critical for the induction of immune responses (Czajkowsky *et al.*, 2012). Among these, antibody-dependent cellular cytotoxicity (ADCC) is an adaptive immune response, mainly mediated by the natural killer (NK) cells. After the specific interactions between antibodies or Fc fusion proteins with specific antigens on the cell surface of infected cells, cross-linking of the Fc domain will activate the CD16 (Fc γ RIII) receptor to trigger degranulation (CD107a on the cell membrane) and cytokine production (IFN- γ and TNF- α) of NK cells (McDonald *et al.*, 2015; Sun *et al.*, 2019). A recent report showed that the RBD-specific antibodies from an individual infected with SARS-CoV in 2003 could induce nearly 10% ADCC against SARS-CoV-2 (Pinto *et al.*, 2020). Therefore, experiments were conducted to examine whether ACE2-Fc could induce primary human NK cell activation. H1975 cells, transduced with full-length Spike by the lentiviral vector (H1975-Spike), were used as target cells (Fig 7A). The NK cell degranulation assay was performed to determine the CD107a, IFN- γ , and TNF- α expression levels after the co-incubation of the NK cells with H1975-Spike cells in the presence of ACE2-Fc or recombinant ACE2 (1-740 A.A. without an Fc tag). Induction of the expression levels of these three degranulation markers was observed when serially diluted ACE2-Fc was added to the co-culture of NK and H1975-Spike cells (Fig 7B-7D). In contrast, the degranulation of NK cells was not observed in the presence of recombinant ACE2. Taken together, our results suggest that ACE2-Fc could not only block SARS-CoV-2 infection but also induce NK cell activation, which may help to remove the infected cells *in vivo*.

(Page 9, line 10)

The renin-angiotensin system (RAS) is a hormone system that regulates vascular function, including the regulation of blood pressure, natriuresis, and blood volume control (Tikellis & Thomas, 2012). As a key regulator in the RAS, ACE2 acts by converting Ang II into Ang 1–7 (Gheblawi *et al.*, 2020). Ang II can also downregulate ACE2 expression through AT1R when the RAS is overactivated in cardiovascular disease through the ERK1/2 and p38 MAPK signaling pathways (Koka *et al.*, 2008). Three strategies have been evolved to reduce Ang II levels upon RAS overactivation, including ACE inhibitors (ACEis), angiotensin receptor blockers (ARBs), and mineralocorticoid receptor blockers (Gheblawi *et al.*, 2020). Most of these compounds increase the protein and mRNA levels of ACE2 or peptidase activity in animal models (Kai & Kai, 2020). Being known as the receptor for SARS-CoV-2, we speculate that ARBs- or ACEis-induced ACE2 upregulation may facilitate SARS-CoV-2 infection. In addition, the administration of recombinant ACE2 or ACE2-Fc could augment ACE2 enzymatic activity and reduce systemic inflammation. A lower plasma Ang II level after taking recombinant ACE2 may be safe in healthy volunteers and patients with acute respiratory distress syndrome or pulmonary arterial hypertension as demonstrated in the previous pilot studies (Haschke *et al.*, 2013; Hemnes *et al.*, 2018; Khan *et al.*, 2017). Therefore, for RAS disorder patients with COVID-19 treatment, administration of ACE2-Fc in circulation may block SARS-CoV-2 infection even in those subjects with induced ACE2 after receipt of ARBs or ACEis treatment. Nevertheless, it should be noted that similar to ARBs and ACEis, administration of ACE2-Fc may be harmful to the developing fetus during pregnancy since it could lower blood pressure by converting Ang II to Ang 1-7.

Figure EV3 *in vitro* cytotoxicity and plasma stability and of ACE2-Fc. (A and B) Two normal human bronchial epithelial cells were incubated with ACE2-Fc and normal human IgG at the indicated concentrations for 72 hr, and cell viability was analyzed by MTS assay. Error bars represent the mean \pm SD; $n=3$. (C) *In vitro* serum stability of ACE2-Fc. ACE2-Fc was incubated with 50% normal human serum at 37°C for up to ten days. At the indicated time points, samples

were collected to quantify the binding ability of ACE2-Fc to Spike proteins by ELISA. Error bars represent the mean \pm SD; n=3. Experiments were performed at least three times with similar results.

Figure 2. Functional characterization of the decoy antibody. (D) Preservation of ACE2-Fc peptidase activity. The peptidase activity of ACE2-Fc was measured by cleavage of the fluorescent peptide substrates. (E) Inhibition of angiotensin II (Ang II)-induced TNF- α production by ACE2-Fc. Ang II was preincubated with indicated amounts of ACE2-Fc or IgG for 30 minutes. After that, the mixtures were added to RAW264.7 cells for 12 hr. The concentration of TNF- α in the culture medium was determined by ELISA. (F) Inhibition of Ang II-induced ADAM17 (a disintegrin and metalloprotease 17) phosphorylation by ACE2-Fc. The protein extracts from ϵ were immunoblotted with the indicated antibodies (left panel). β -actin served as the loading control. ADAM17 represented. The signal intensity was normalized to cells only (right panel). Data are representative of three independent experiments, and the values are expressed as the mean \pm SD (lower panel). Error bars represent the mean \pm SD, n=3. * $P < 0.05$. IgG represents the human normal IgG control. Experiments were performed at least three times with similar results.

Figure 7. Effects of ACE2-Fc on NK cell degranulation. IL-2 activated NK cells were incubated with H1975-Spike cells at a 1:1 cell ratio in the presence of ACE2-Fc or ACE2. The activation of NK cell were determined by the CD107a, IFN- γ , and TNF- α expression levels. (A) H1975 cells were transduced with full-length Spike by Lentiviral vector. Protein extracts were immunoblotted with the indicated antibodies. O/E represents overexpress. (B) The effect of ACE2-Fc activation on degranulative capacity of NK cells. The experiments were performed with the primary human NK cells derived from three independent donors. Error bars represent mean \pm SD, n= 3. * P<0.05, *** P< 0.001. (C) The CD107a, IFN- γ , and TNF- α expression levels were determined by flow cytometry after the NK cells and H1975-Spike cells coincubation in the presence or absence of ACE2-Fc or ACE2 control. Error bars represent the mean \pm SD. The percentage of positive cells in both groups were normalized to the NK/H1975-Spike only group. (D) The differential expression of degranulation markers after the treatment of ACE2-Fc or ACE2 by the flow cytometry. The numbers in each plot indicate the percentages of positive cells.

Q2. The design of the study focusses on the early stages of infection, as they preincubated the virus with the decoy fusion protein before adding to Vero-6 cells or to HEK293 cells This study cannot be used to predict the effect of the ACE2-Fc fusion in later stages of the disease process. Could the authors add the decoy to cells that are infected already and see how this decoy altered the course of infection when it has already established?

Response:

We thank the Reviewer for this constructive comment. In order to address this question, we added the decoy antibody to the culture medium after SARS-CoV-2 infection of Vero E 6 cells. Based on the experiment results (below), it demonstrated that the decoy antibody was functioning well when added at the pre-treatment stage, but less effective when added in the post-treatment stage. It implicates that the decoy antibody might exert more inhibitory effects as a prophylactic strategy. Base on the experiment result, it demonstrated that the ACE2-Fc may partially block (30-40%) the virus infection and replication at the post-treatment stage. However, further experiment in human lung epithelial cells is required to confirm this finding since Vero E6 is interferon-deficient, which might not be a good target cell to examine the functional consequences of ACE2-Fc's peptidase activity and capability to induce NK cell activation at the post-infection stage.

Figure Blockage of SARS-CoV-2 entry into Vero E6 cells by ACE2-Fc. Pre-treat group: ACE2-Fc and SARS-CoV-2 premixtures were incubated for one hour before being added to ACE2 expressing H1975 cells for another 1 hr at 37°C. Post-treat group: After SARS-CoV-2 infection, Vero E6 with 0.1 MOI for 1 hr, replaced the culture medium with 200 $\mu\text{g/mL}$ ACE2-Fc or IgG for another 2 days. After that, the RNA was extracted from the cell lysates and the viral

RNA expression was determined by the real-time PCR (A) and western blot (B). (C) Inhibition of SARS-CoV-2 infection by ACE2-Fc in the plaque assay. Mixtures of ACE2-Fc and SARS-CoV-2 were incubated for one hour before adding to Vero E6 cells for another 1 hr at 37°C. The ACE2-Fc and SARS-CoV-2 premixtures were removed, and the cells were washed once with PBS and overlaid with methylcellulose with 2% FBS for 6 days before being stained with crystal violet. The NP/PCNA represents the relative NP expression as compared to that of PCNA, which served as a loading control. The NP/PCNA numbers below the panel are the ratios of NP/PCNA normalized to that of the human IgG control group. PCNA: Proliferating cell nuclear antigen. Error bars represent the mean \pm SD, n=3. *** $P < 0.001$.

Q3. There was no attempt to study the effect in a disease model such as in lung organoids or in mice or at the very least discussing future work along these lines.

Response:

We thank the Reviewer for this constructive comment.

The airway organoid model was established to examine the neutralizing ability of ACE2-Fc on spike mediated pseudotyped entry. The airway organoids are carefully characterized and shown to exhibit similar features as described in the previous report (Sachs *et al.*, 2019) (Fig 4B and 4C). The airway organoids are shown to be susceptible to SARS-CoV-2 infection. As expected, the ACE2-Fc can significantly block virus entry in the airway organoids (Fig 4D). We have incorporated the experimental results to the section of **ACE2-Fc blocks pseudotyped lentivirus entry into ACE2-expressing cells and lung organoids**. The additional results have been incorporated in the Results section as following:

(Page6, line 25)

Since the lung is the primary site for SARS-CoV-2 infection, we established an airway organoid model following the methods described in a recent article (Sachs *et al.*, 2019) to investigate the neutralization ability of ACE2-Fc against SARS-CoV-2 entry. The derived airway differentiation organoids were successfully established and composed of several airway epithelial cells with specific markers, including basal (P63), secretory (club cell marker secretoglobin family 1A member 1 (SCGB1A1) and secretory cell marker mucin 5AC (MUC5AC)), and multiciliated cells (cilia marker acetylated α -tubulin) (Fig 4B). Of note, unlike the A549 or the parental human normal bronchial epithelial cells (HBEpc), these airway organoids expressed a high-level of ACE2 in addition to TMPRSS2 (Fig 4B and 4C). We then examined the neutralization ability of ACE2-Fc for Spike-expressing pseudotyped virus in the airway organoid model. As shown in Fig 4D, the airway organoids are susceptible to virus entry and the ACE2-Fc could significantly block virus entry at the concentration of 100 μ g/mL. Our findings indicated that ACE2-Fc can

inhibit entry of SARS-CoV-2 Spike-expressing pseudotyped virus entry into ACE2-expressing cells, including the airway organoids.

Figure 4. Blockage of Spike-expressing pseudovirus entry into ACE2-expressing cells by ACE2-Fc. (B) Confocal microscopy images of the airway organoids. The green fluorescence represents the specific staining of indicated monoclonal antibodies and Alexa Fluor 488-conjugated secondary antibodies. DAPI was used as a nuclear counterstain. Hematoxylin and eosin (H&E) stain: hematoxylin stained the nuclei in blue color; eosin stained the cytoplasm in pink color. Scale bars equal to 20 μm . (C) Lung cancer A549 cells, human normal bronchial epithelial cells (HBEpc), and HBEpc-differentiated cells (airway organoids) were harvested for immunoblotting with the indicated antibodies. (D) Blockage of pseudovirus entry into airway organoids by ACE2-Fc. Mixtures of pseudotyped lentivirus with or without ACE2-Fc were cocultured with airway organoids for 72 hr. The virus entry was determined by quantifying the luciferase activity in the cell lysates. R.I.U: relative infection unit. Error bars represent the mean \pm SD, n=3. ** $P < 0.01$, *** $P < 0.001$. Experiments were performed at least three times with similar results.

Q4. The RAS system represents a complex network of pathways that are influenced by many factors hence it is not clear how the ACE2-Fc will impact on this system. All limitations listed above have not been discussed in this study.

Response:

We thank the Reviewer’s critical suggestions. We have added the potential impacts of ACE2-Fc on the RAS system in the *Discussion* section.

(Page 9, line 10)

The renin-angiotensin system (RAS) is a hormone system that regulates vascular function, including the regulation of blood pressure, natriuresis, and blood volume control (Tikellis & Thomas, 2012). As a key regulator in the RAS, ACE2 acts by converting Ang II into Ang 1–7 (Gheblawi *et al.*, 2020). Ang II can also downregulate ACE2 expression through AT1R when the

RAS is overactivated in cardiovascular disease through the ERK1/2 and p38 MAPK signaling pathways (Koka *et al.*, 2008). Three strategies have been evolved to reduce Ang II levels upon RAS overactivation, including ACE inhibitors (ACEis), angiotensin receptor blockers (ARBs), and mineralocorticoid receptor blockers (Gheblawi *et al.*, 2020). Most of these compounds increase the protein and mRNA levels of ACE2 or peptidase activity in animal models (Kai & Kai, 2020). Being known as the receptor for SARS-CoV-2, we speculate that ARBs- or ACEis-induced ACE2 upregulation may facilitate SARS-CoV-2 infection. In addition, the administration of recombinant ACE2 or ACE2-Fc could augment ACE2 enzymatic activity and reduce systemic inflammation. A lower plasma Ang II level after taking recombinant ACE2 may be safe in healthy volunteers and patients with acute respiratory distress syndrome or pulmonary arterial hypertension as demonstrated in the previous pilot studies (Haschke *et al.*, 2013; Hemnes *et al.*, 2018; Khan *et al.*, 2017). Therefore, for RAS disorder patients with COVID-19 treatment, administration of ACE2-Fc in circulation may block SARS-CoV-2 infection even in those subjects with induced ACE2 after receipt of ARBs or ACEis treatment. Nevertheless, it should be noted that similar to ARBs and ACEis, administration of ACE2-Fc may be harmful to the developing fetus during pregnancy since it could lower blood pressure by converting Ang II to Ang 1-7.

Q5. The discussion should clarify how the authors see this decoy being employed as a therapeutic strategy. Such as to deliver the decoy product by inhaler given to healthcare workers or those people at risk as for preventing infection. This might be met with better outcomes as disease might be milder as this could be a 'protective shield' against Covid-19 by inhibiting and neutralising the virus' activity in naïve at-risk populations.

Response:

We thank the Reviewer for this constructive comment. We have added descriptions about the employment of the decoy antibody as a therapeutic strategy in the *Discussion* section.

(Page 10, line 24)

Taken together, we demonstrated that ACE2-Fc (18-615A.A.) has three important characteristics: neutralization of SARS-CoV-2 infection, conversion of Ang II to Ang 1-7, and activation of NK cells. Based on the efficient blocking of SARS-CoV-2 entry, including the D614G variant strains, in the human epithelial cells and the airway organoids, we believe that the prophylactic use of ACE2-Fc could prevent healthcare workers or people at high risk from SARS-CoV-2 infection. Nevertheless, the potential benefits of Ang II conversion and NK cell activation activities of ACE2-Fc in the COVID-19 disease process require further evaluation in the animal model. Our *in vitro* results suggest that ACE2-Fc has the potential to develop as an

effective therapeutic against SARS-CoV-2 infection.

Q6. "The viral entry blocking effect of ACE2-Fc was further confirmed using real SARA-CoV-2 isolated from patients suffering from COVID-19 infection in National Taiwan University Hospital". There are no further details in the methods regarding how the isolates were obtained including clarity in the methods as to how many isolates were tested. Were the isolates sequenced or identified by some method? Were they the same or different variants of SARS-COV-2?

Response:

We thank the Reviewer for pointing out this question. We have added the detail information at “**SARS-CoV2 isolation**” of the Materials and Methods section. The genetic characteristics of the clinical isolates used were summarized in Table 1, and Page 18, line 9 in the revised manuscript,

SARS-CoV2 isolation

Sputum or throat swab specimens obtained from SARS-CoV-2-infected patients were maintained in the viral-transport medium. The specimens were propagated in VeroE6 cells in DMEM supplemented with 2 µg/mL tosylsulfonyl phenylalanyl chloromethyl ketone (TPCK)-trypsin (Sigma-Aldrich). Culture supernatants were harvested when more than 70% of cells showed cytopathic effects. The full-length genomic sequences of the derived clinical isolates were determined and submitted, along with the patients’ travel history and basic information, to the GISAID database. The virus strains used in this study include SARS-CoV-2/NTU03/TWN/human/2020 (Accession ID EPI_ISL_413592), SARS-CoV-2/NTU13/TWN/human/2020 (Accession ID EPI_ISL_422415), SARS-CoV-2/NTU14/TWN/human/2020 (Accession ID EPI_ISL_422416), SARS-CoV-2/NTU18/TWN/human/2020 (Accession ID EPI_ISL_447615), SARS-CoV-2/NTU25/TWN/human/2020 (Accession ID EPI_ISL_447619) and SARS-CoV-2/NTU27/TWN/human/2020 (Accession ID EPI_ISL_447621). The virus titers were determined by plaque assay (Su *et al*, 2008) for subsequent analysis. The study was approved by the NTUH Research Ethics Committee (202002002RIND), and the participants gave written informed consent.

Table 1. Genetic characteristics of SARS-CoV-2 strains used in this study.

*SARS-CoV-2 viruses were isolated from sputum or throat swab specimens obtained from SARS-CoV-2-infected patients using VeroE6 cells in DMEM supplemented with 2 µg/mL tosylsulfonyl phenylalanyl chloromethyl ketone (TPCK)-trypsin (Sigma-Aldrich). The full-length genomic sequences of the derived clinical isolates were determined and submitted,

along with the patients’ travel history and basic information, to the GISAID database. The nucleotide numbers where the variance were observed in these 6 clinical isolates were indicated, along with the corresponding amino acid changes, if any, in the indicated viral genes.

	PANGOLIN lineages	GISAID clade	Accession ID	S-		NS3-	NS8-	N-		
				D614G	Q57H	L84S	G204R			
				241	3037	8782	23403	25563	28144	28882
NTU01	A	S	EPI_ISL_408489			T			C	
NTU03	B.1	GH	EPI_ISL_413592	T	T		G	T		
NTU13	A.3	S	EPI_ISL_422415			T			C	
NTU14	B.1.1	GR	EPI_ISL_422416	T	T		G			A
NTU25	B.1.5.3	G	EPI_ISL_447619	T	T		G			
NTU27	B	L	EPI_ISL_447621							

Q7. Ethics relating to the collection of isolates have not been clarified if this is a consideration.

Response:

We thank the Reviewer for reminding us of the critical issue. We have added the related information at “SARS-CoV2 isolation” of the Materials and Methods section as following:

(Page 18, line 24)

The study was approved by the NTUH Research Ethics Committee (202002002RIND), and the participants gave written informed consent.

Q8. Statistical analysis should clarify number of repeats

Response:

We appreciate the Reviewer’s suggestion. The numbers of repeats for the statistical analysis have been provided.

Q9. Spellings and to improve clarity I suggest rephrasing

In introduction

Base- Based

Whether ACE2-Fc fusion still owning the peptidase activity of ACE2 could block SARS-COV-2 entry and endow the ability of antibodies including ADCC and CDC remains to be investigated

Results

For clarity for non-specialist

*Why was IL-2 signalling peptide included at the C-terminus of ACE2?
SARA-COV-2 should read SARS-COV-2*

In discussion

Our study results demonstrated that even when we shortened the ACE protein to 597 amino acids....

The chimeric ACE2-Fc antibody had a much longer elimination phase or much longer half-life (not both in same sentence). I couldn't see this data; my assumption is that it is based on the cited publication (12) which clearly shows this effect.

Fragment need clarity...In addition to identify infected cells and provide the opportunity for the immune system to clean injured cells

Methods

At 1:1000 to 1:10000 dilution. This is not a sentence, needs revision

Eaction buffer should it read - reaction buffer?

The transduction unit of VSV-G-peudotyped lentivirus.... Typo

Figure 2 legend GAPDH is served as a control- fragment needs revision

Response:

We thank the Reviewer for pointing out the typos and we have carefully revised our manuscript accordingly.

Reference

- Czajkowsky DM, Hu J, Shao Z, Pleass RJ (2012) Fc-fusion proteins: new developments and future perspectives. *EMBO Mol Med* 4: 1015-1028
- Gheblawi M, Wang K, Viveiros A, Nguyen Q, Zhong JC, Turner AJ, Raizada MK, Grant MB, Oudit GY (2020) Angiotensin-Converting Enzyme 2: SARS-CoV-2 Receptor and Regulator of the Renin-Angiotensin System: Celebrating the 20th Anniversary of the Discovery of ACE2. *Circ Res* 126: 1456-1474
- Haschke M, Schuster M, Poglitsch M, Loibner H, Salzberg M, Bruggisser M, Penninger J, Krahenbuhl S (2013) Pharmacokinetics and pharmacodynamics of recombinant human angiotensin-converting enzyme 2 in healthy human subjects. *Clin Pharmacokinet* 52: 783-792
- Hemnes AR, Rathinasabapathy A, Austin EA, Brittain EL, Carrier EJ, Chen X, Fessel JP, Fike CD, Fong P, Fortune N *et al* (2018) A potential therapeutic role for angiotensin-converting enzyme 2 in human pulmonary arterial hypertension. *Eur Respir J* 51
- Kai H, Kai M (2020) Interactions of coronaviruses with ACE2, angiotensin II, and RAS inhibitors-lessons from available evidence and insights into COVID-19. *Hypertens Res* 43: 648-654
- Khan A, Benthin C, Zeno B, Albertson TE, Boyd J, Christie JD, Hall R, Poirier G, Ronco JJ, Tidswell M *et al* (2017) A pilot clinical trial of recombinant human angiotensin-converting enzyme 2 in acute respiratory distress syndrome. *Crit Care* 21: 234
- Koka V, Huang XR, Chung AC, Wang W, Truong LD, Lan HY (2008) Angiotensin II up-regulates angiotensin I-converting enzyme (ACE), but down-regulates ACE2 via the AT1-ERK/p38 MAP kinase pathway. *Am J Pathol* 172: 1174-1183
- Korber B, Fischer WM, Gnanakaran S, Yoon H, Theiler J, Abfalterer W, Hengartner N, Giorgi EE, Bhattacharya T, Foley B *et al* (2020) Tracking Changes in SARS-CoV-2 Spike: Evidence that D614G Increases Infectivity of the COVID-19 Virus. *Cell* 182: 812-827 e819
- Lei C, Qian K, Li T, Zhang S, Fu W, Ding M, Hu S (2020) Neutralization of SARS-CoV-2 spike pseudotyped virus by recombinant ACE2-Ig. *Nat Commun* 11: 2070
- McDonald R, Boaden R, Roland M, Kristensen SR, Meacock R, Lau YS, Mason T, Turner AJ, Sutton M (2015) In: *A qualitative and quantitative evaluation of the Advancing Quality pay-for-performance programme in the NHS North West*, Southampton (UK)
- Pinto D, Park YJ, Beltramello M, Walls AC, Tortorici MA, Bianchi S, Jaconi S, Culap K, Zatta F, De Marco A *et al* (2020) Cross-neutralization of SARS-CoV-2 by a human monoclonal SARS-CoV antibody. *Nature* 583: 290-295

Sachs N, Papaspyropoulos A, Zomer-van Ommen DD, Heo I, Bottinger L, Klay D, Weeber F, Huelsz-Prince G, Iakobachvili N, Amatngalim GD *et al* (2019) Long-term expanding human airway organoids for disease modeling. *EMBO J* 38

Su CT, Hsu JT, Hsieh HP, Lin PH, Chen TC, Kao CL, Lee CN, Chang SY (2008) Anti-HSV activity of digitoxin and its possible mechanisms. *Antiviral Res* 79: 62-70

Sun P, Williams M, Nagabhushana N, Jani V, Defang G, Morrison BJ (2019) NK Cells Activated through Antibody-Dependent Cell Cytotoxicity and Armed with Degranulation/IFN-gamma Production Suppress Antibody-dependent Enhancement of Dengue Viral Infection. *Sci Rep* 9: 1109

Tikellis C, Thomas MC (2012) Angiotensin-Converting Enzyme 2 (ACE2) Is a Key Modulator of the Renin Angiotensin System in Health and Disease. *Int J Pept* 2012: 256294

Wysocki J, Schulze A, Batlle D (2019) Novel Variants of Angiotensin Converting Enzyme-2 of Shorter Molecular Size to Target the Kidney Renin Angiotensin System. *Biomolecules* 9

Yurkovetskiy L, Pascal KE, Tompkins-Tinch C, Nyalile T, Wang Y, Baum A, Diehl WE, Dauphin A, Carbone C, Veinotte K *et al* (2020) SARS-CoV-2 Spike protein variant D614G increases infectivity and retains sensitivity to antibodies that target the receptor binding domain. *bioRxiv*

22nd Oct 2020

Dear Prof. Yang,

Thank you for the submission of your revised manuscript to EMBO Molecular Medicine. I am pleased to inform you that we will be able to accept your manuscript pending the following final amendments:

Please implement all adjustments suggested by the referee #2

Yours sincerely,

Zeljko Durdevic

***** Reviewer's comments *****

Referee #1 (Comments on Novelty/Model System for Author):

Incorporation of the airway organoids, clinical isolate CoV2, and assessing ACE2-Fc activity in neutralization assay and cell-cell fusion has significantly improved the technical quality and model systems used. Adding the effect of ACE2-Fc on NK cell activity, ADCC, and its stability in plasma has improved the manuscript's translational impact. The readability and reproducibility has been improved by including the statistics methodology.

Referee #2 (Remarks for Author):

SARS-Coronavirus-2 (SARS-CoV-2) is the cause of the COVID-19 pandemic (with over 38 million people infected and over a million death) and there is currently no effective treatment. SARS-CoV-2, just like SARS-CoV and MERS-CoV causes severe acute respiratory distress syndrome in susceptible hosts. SARS-CoV-2 uses Angiotensin Converting Enzyme II (ACE2) bind directly to SARS-CoV-2 Spike protein and is used as cell entry receptor, hence this ACE-2 receptor is a key drug target; a lot of research is now centered at investigating the therapeutic potential of ACE-2. ACE2 has peptidase activity which negatively regulates Angiotensin II (AngII).

This study investigated the suitability of humanised ACE2-Fc fusion as decoy antibodies for preventing the entry of SARS-COV-2 into human cells in vitro. The ACE2-Fc fusion protein can form dimer mimicking antibodies and specifically binds to SARS-CoV-2 Spike protein. By competitively binding to the viral Spike protein, the decoy prevents viral binding to the natural, membrane bound ACE2, and thus blocks virus entry into host cells. Using ELISA, the authors demonstrate that recombinant ACE2-Fc binds recombinant SARS-CoV-2 Spike Fc protein. Pre-incubation of Spike protein pseudotyped lentivirus or clinically isolated SARS-CoV-2 with ACE2-Fc reduces infection of ACE2 overexpressing 293T cell line and Vero E6 cells, respectively. The study utilised virus from six clinical isolates including the six SARS-CoV-2 clinical strains, including the D614G variants which have been shown to exhibit increased infectivity and demonstrated that the ACE2-Fc fusion protein abrogated viral replication. This study supports the recently published work by Monteil, and colleagues who showed that recombinant purified human ACE2 (hrsACE2) - a decoy protein that has already been tested in phase I and II clinical trials can inhibit the interaction between SARS-

CoV-2 and ACE2. However, the use of different clinical isolates including the D614 isolate is a novel finding that informs the suitability of the ACE2 decoy for generalized therapy for SARS-CoV-2 infections.

Previous studies showed that treatment with recombinant ACE2 protein prevents Ang II-induced hypertension and cardiac dysfunction and might control AngII-AT1R proinflammatory cytokine responses, thereby protecting from severe acute lung failure. In this study, the authors generated recombinant ACE2 fused to Fc that is known to increase its half-life. They show that ACE2-Fc has increased half-life of up to 10 days and is also not cytotoxic. Interestingly, their ACE2-Fc has peptidase activity as demonstrated by cleaving fluorescent substrate. The effects of ACE2-Fc on Ang II-mediated inflammatory cascade was investigated by coincubation of Ang II with ACE2-Fc. In this case ACE2-Fc significantly suppressed Ang II-induced TNF- α production and phosphorylation of ADAM17 (a disintegrin and metalloprotease 17). These observations suggest that the ACE2-Fc decoy antibodies preserves the peptidase activity of ACE2-Fc to reduce the Ang II-mediated cytokine cascade which might be useful in reducing the so-called cytokine storm. The results also suggest this ACE2-Fc decoy can induce degranulation of NK cells and this was not observed in the presence of recombinant ACE2, suggesting the ACE2-Fc decoy specifically induces NK cell activation which could have the added benefit of promoting clearance of the virus. They also showed that ACE2-Fc could inhibit SARS-CoV-2 Spike-mediated cell-cell fusion and syncytia formation.

The author then established an airway organoid model and confirmed the expression of the key markers (including basal32 (P63), secretory (club cell marker secretoglobin family 1A member 1 (SCGB1A1) and 33 secretory cell marker mucin 5AC (MUC5AC). The organoids expressed a high-level of ACE2 but lower levels of TMPRSS2 when compared to the A549 cell line or the parental human normal bronchial epithelial cells (HBEpc). In these organoids ACE2-Fc decoy also reduced the infectivity of the SARS-CoV-2 virus.

Overall, the results are clear, and the design and execution of the experiments is sound. The authors have taken the reviewers and editors comments onboard and revised their manuscript to include new data on:

1. The benefit of their ACE-Fc decoy with the peptidase activity.
2. Used airway organoids to demonstrate application of this ACE2-Fc in a more physiologically relevant model.
3. Used at least 6 different clinical isolates to show that this ACE-2 can be applicable to SARS-CoV-2 infections in general.
4. They have also performed experiments on Spike-mediated cell-cell fusion and syncytia formation to investigate how the decoy affects this.
5. They have revised the discussion highlight the limitations of the study
6. The details of obtaining clinical isolates and ethics have been included.
7. They have also tried to find out if this decoy could work on established infection. Here they found very little benefit suggesting the decoy could be useful more for prevention instead of as a treatment option.

I am satisfied with the work that has been done for improving this manuscript.

Minor Improvements

1. Figure 3C- Please show by arrows the syncytia so that it is clear what the readers are looking at. Also, the figures C and D are showing different concentrations of the ACE2-Fc; 25 and 50 ug/ml) in

D but 25 and 100 ug /ml in C. Is Figure D not a quantification of C? If not please shows the results of C and D which correlate. This is important as there is a dose effect on this experiment. 25ug/ml of ACE2-Fc still show some syncytia when compared to 100.

2. Also, by using different concentrations are you investigating dose dependency? Please clarify the dose effect in your description of results, this applies for other figures as well.

3. The cytotoxicity was investigated with increasing concentrations up to 200mg/ml, yet the organoid assay went up to 300mg/ml. It will be good to see these experiments done in a way that the concentrations are harmonized across experiments. Is 300mg/ml cytotoxic or not?

Dear Editor:

We would like to thank the Editors/Reviewers for your time and kind consideration of our manuscript. We have followed the comments from the expert reviewers, performed additional experiments, and carefully addressed all the critiques in the revised manuscript. Please see the point-to-point response below,

***** Editor's comments *****

1) Please implement all adjustments suggested by the referee #2.

Response:

We thank the Editor for the comment. We thank the Reviewer for pointing out the “Minor Improvements” and we have carefully revised our manuscript accordingly.

***** Reviewer's comments *****

Referee #1 (Comments on Novelty/Model System for Author):

Incorporation of the airway organoids, clinical isolate CoV2, and assessing ACE2-Fc activity in neutralization assay and cell-cell fusion has significantly improved the technical quality and model systems used. Adding the effect of ACE2-Fc on NK cell activity, ADCC, and its stability in plasma has improved the manuscripts translational impact. The readability and reproducibility has been improved by including the statistics methodology.

Response:

We thank the Reviewer for the kind comment.

Referee #2 (Remarks for Author):

SARS-Coronavirus-2 (SARS-CoV-2) is the cause of the COVID-19 pandemic (with over 38 million people infected and over a million death) and there is currently no effective treatment. SARS-CoV-2, just like SARS-CoV and MERS-CoV causes severe acute respiratory distress syndrome in susceptible hosts. SARS-CoV-2 uses Angiotensin Converting Enzyme II (ACE2) bind directly to SARS-CoV-2 Spike protein and is used as cell entry receptor, hence this ACE-2 receptor is a key drug target; a lot of research is now centered at investigating the therapeutic potential of ACE-2. ACE2 has peptidase activity which negatively regulates Angiotensin II (AngII).

This study investigated the suitability of humanised ACE2-Fc fusion as decoy antibodies for preventing the entry of SARS-CoV-2 into human cells *in vitro*. The ACE2-Fc fusion protein can form dimer mimicking antibodies and specifically binds to SARS-CoV-2 Spike protein. By competitively binding to the viral Spike protein, the decoy prevents viral binding to the natural, membrane bound ACE2, and thus blocks virus entry into host cells. Using ELISA, the authors demonstrate that recombinant ACE2-Fc binds recombinant SARS-CoV-2 Spike Fc protein. Pre-incubation of Spike protein pseudotyped lentivirus or clinically isolated SARS-CoV-2 with ACE2-Fc reduces infection of ACE2 overexpressing 293T cell line and Vero E6 cells, respectively. The study utilised virus from six clinical isolates including the six SARS-CoV-2 clinical strains, including the D614G variants which have been shown to exhibit increased infectivity and demonstrated that the ACE2-Fc fusion protein abrogated viral replication. This study supports the recently published work by Monteil, and colleagues who showed that recombinant purified human ACE2 (hrsACE2) - a decoy protein that has already been tested in phase I and II clinical trials can inhibit the interaction between SARS-CoV-2 and ACE2. However, the use of different clinical isolates including the D614 isolate is a novel finding that informs the suitability of the ACE2 decoy for generalized therapy for SARS-CoV-2 infections.

Previous studies showed that treatment with recombinant ACE2 protein prevents Ang II-induced hypertension and cardiac dysfunction and might control AngII-AT1R proinflammatory cytokine responses, thereby protecting from severe acute lung failure. In this study, the authors generated recombinant ACE2 fused to Fc that is known to increase its half-life. They show that ACE2-Fc has increased half-life of up to 10 days and is also not cytotoxic. Interestingly, their ACE2-Fc has peptidase activity as demonstrated by cleaving fluorescent substrate. The effects of ACE2-Fc on Ang II-mediated inflammatory cascade was investigated by coincubation of Ang II with ACE2-Fc. In this case ACE2-Fc significantly suppressed Ang II-induced TNF- α production and phosphorylation of ADAM17 (a disintegrin and metalloprotease 17). These observations suggest that the ACE2-Fc decoy antibodies preserves the peptidase activity of ACE2-Fc to reduce the Ang II-mediated cytokine cascade which might be useful in reducing the so-called cytokine storm. The results also suggest this ACE2-Fc decoy can induce degranulation of NK cells and this was not observed in the presence of recombinant ACE2, suggesting the ACE2-Fc decoy specifically induces NK cell activation which could have the added benefit of promoting clearance of the virus. They also showed that ACE2-Fc could inhibit SARS-CoV-2 Spike-mediated cell-cell fusion and syncytia formation.

The author then established an airway organoid model and confirmed the expression of the key markers (including basal32 (P63), secretory (club cell marker secretoglobin family 1A member 1 (SCGB1A1) and 33 secretory cell marker mucin 5AC (MUC5AC). The organoids expressed a high-level of ACE2 but lower levels of TMPRSS2 when compared to the A549 cell line or the parental human normal bronchial epithelial cells (HBEpc). In these organoids ACE2-Fc decoy also reduced the infectivity of the SARS-CoV-2 virus.

Overall, the results are clear, and the design and execution of the experiments is sound.

The authors have taken the reviewers and editors comments onboard and revised their manuscript to include new data on:

1. The benefit of their ACE-Fc decoy with the peptidase activity.
2. Used airway organoids to demonstrate application of this ACE2-Fc in a more physiologically relevant model.
3. Used at least 6 different clinical isolates to show that this ACE-2 can be applicable to SARS-CoV-2 infections in general.
4. They have also performed experiments on Spike-mediated cell-cell fusion and syncytia formation to investigate how the decoy affects this.
5. They have revised the discussion highlight the limitations of the study
6. The details of obtaining clinical isolates and ethics have been included.
7. They have also tried to find out if this decoy could work on established infection. Here they found very little benefit suggesting the decoy could be useful more for prevention instead of as a treatment option.

I am satisfied with the work that has been done for improving this manuscript.

Response:

We thank the Reviewer for the kind comment. We will try to adjust the concentration of the decoy antibody or reduce the virus load for further confirming the treated benefit on the patients who had established infection in our future publications.

Minor Improvements

1. Figure 3C- Please show by arrows the syncytia so that it is clear what the readers are looking at. Also, the figures C and D are showing different concentrations of the ACE2-Fc; 25 and 50 ug/ml) in D but 25 and 100 ug /ml in C. Is Figure D not a quantification of C? If not please shows the results of C and D which correlate. This is important as there is a dose effect on this experiment. 25ug/ml of ACE2-Fc still show some syncytia when compared to 100.

Response:

Thank the Reviewer for point out this mistake. Indeed, Figure 3D is a quantification of Figure 3C. A corrected panel has been updated in the revised Fig 3D. We also carefully checked other panels in our revised manuscript. In figure 5A, two IgG controls were used 200 µg/mL. We have also revised the y-axis in figure 6C. As reviewer point out, there is a dose effect in syncytia formation.

Original figure 3C and 3D,

Revised figure 3C and 3D,

Figure 3. Inhibition of Spike-induced cell-cell fusion and syncytia formation by ACE2-Fc.

(C) GFP and full-length Spike co-transfected HEK293T cells were used as effector cells (293-S). HEK293/ACE2 (HEK293T-overexpressing ACE2) (left panel) or H1975/ACE2 (H1975-overexpressing ACE2) (right panel) cells were used as target cells. The 293-S cells were preincubated with normal human IgG or ACE2-Fc at 37°C for 1 hr. After that, the mixtures were added to target cells for 4 hr (cell-cell fusion) or 24 hr (syncytia formation). The fluorescent areas were measured by inverted fluorescence microscopy and Metamorph software (Metamorphosis). Scale bars equal to 50 µm. The white arrows indicate the cell-cell fusion or syncytia formation.

(D) The inhibition of cell-cell fusion and syncytia formation by ACE2-Fc in HEK293/ACE2 and H1975/ACE2 was determined with the formula described in the Methods section. Error bars represent the mean ± SD, n=6. Statistical analysis was performed by unpaired two tail t-test. ** P < 0.01, *** P < 0.001. Experiments were performed at least three times with similar results.

Original figure 5A,

Revised figure 5A,

Figure 5. Blockage of SARS-CoV-2 entry into host cells by ACE2-Fc.

(A) Inhibition of SARS-CoV-2 infection by ACE2-Fc in the plaque assay. Mixtures of ACE2-Fc and SARS-CoV-2 were incubated for one hour before adding to Vero E6 cells for another 1 hr at 37°C. The ACE2-Fc and SARS-CoV-2 premixtures were removed, and the cells were washed once with PBS and overlaid with methylcellulose with 2% FBS for 5-7 days before being stained with crystal violet. Those results showed increase of plaque formation was regarded as no inhibition of plaque formation. TPCK-treated trypsin: N-tosyl-L-phenylalanine chloromethyl ketone-treated trypsin.

Original figure 6C,

Revised figure 6C,

Figure 6. Neutralization activity of ACE2-Fc on different SARS-CoV-2 strains. Yield reduction assay was performed to determine the inhibitory effects of ACE2-Fc on the entry of 5 different SARS-CoV-2 strains into Vero E6 cells.

(B, C) The virus RNA in the (B) culture medium and (C) cell lysates was quantified by real-time RT-PCR. The genetic information and sequence reference numbers for the 5 SARS-CoV-2 strains

were summarized in Table 1. Error bars represent the mean \pm SD, n=3. Statistical analysis was performed by unpaired two tail t-test. ** $P < 0.01$, *** $P < 0.001$. Experiments were performed at least three times with similar results.

2. Also, by using different concentrations are you investigating dose dependency? Please clarify the dose effect in your description of results, this applies for other figures as well.

Response:

We appreciate the Reviewer's suggestion. Yes, we used different concentrations for investigating the dose dependency. We also added the dose effect in our revised manuscript in the result section and also uploaded the exact p value between each dose in Appendix table S1.

3. The cytotoxicity was investigated with increasing concentrations up to 200 mg/ml, yet the organoid assay went up to 300mg/ml. It will be good to see these experiments done in a way that the concentrations are harmonized across experiments. Is 300mg/ml cytotoxic or not?

Response:

We appreciate the Reviewer's comment. We have elevated the concentration of IgG and ACE2-Fc to 400 $\mu\text{g/mL}$. The MTS assay shown no cell toxicity was observed in these two normal cells at the concentration up to 400 $\mu\text{g/mL}$ of ACE2-Fc.

(Page 6, Line 3-14)

Cytotoxicity and stability of ACE2-Fc.

To examine the potential cytotoxicity of ACE2-Fc on normal cells, two different normal human bronchial epithelial (NBE) cells were treated with various concentrations of ACE2-Fc or IgG for 3 days before the cell viability assay. As shown in Fig EV3A and EV3B, no cell toxicity was observed in these two normal cells at the concentration up to 400 $\mu\text{g/mL}$ of ACE2-Fc. The stability of ACE2-Fc in serum was subsequently determined. We incubated 2 $\mu\text{g/mL}$ ACE2-Fc in 50% normal human serum at 37°C for 0, 1, 2, and up to 10 days. The stability of ACE2-Fc was determined by assaying its binding ability to the Spike proteins in the ELISA assay. As shown in Fig EV3C, not a significant reduction of ACE2-Fc/Spike binding was observed up to ten days. These results suggest that ACE2-Fc has no toxicity to epithelial cells and may be stable in serum for ten days, which may facilitate its future clinical application.

Original figure EV3A and EV3B,

Revised figure EV3A and EV3B,

Figure EV3 *in vitro* cytotoxicity and plasma stability and of ACE2-Fc.

(A and B) Two normal human bronchial epithelial cells were incubated with ACE2-Fc and normal human IgG at the indicated concentrations for 72 hr, and cell viability was analyzed by MTS assay. Error bars represent the mean \pm SD; n=2 in left; n=3 in right. The dotted line represents the 50 % of cell viability.

The authors performed the requested changes.

Corresponding Author Name: Sui-Yuan Chang and Pan-Chyr Yang

Manuscript Number: EMM-2020-12828